# Impact of sea ice floe size distribution on seasonal fragmentation and melt of Arctic sea ice

Adam W. Bateson[1], Daniel L. Feltham[1], David Schröder[1], Lucia Hosekova[1], Jeff K. Ridley[2], Yevgeny Aksenov[3]

[1]Department of Meteorology, University of Reading, Reading, RG2 7PS, United Kingdom
[2]Hadley Centre for Climate Prediction and Research, Met Office, Exeter, EX1 3PB, United Kingdom
[3]National Oceanography Centre Southampton, Southampton, SO14 3ZH, United Kingdom

*Correspondence to*: Adam W. Bateson (a.w.bateson@pgr.reading.ac.uk)

**Abstract.** Recent years have seen a rapid reduction in the summer Arctic sea ice extent. To both understand this trend and project the future evolution of the summer Arctic sea ice, a better understanding of the physical processes that drive the seasonal loss of sea ice is required. The marginal ice zone, here defined as regions with between 15 and 80% sea ice cover, is the region separating pack ice from the open ocean. Accurate modelling of this region is important to understand the dominant mechanisms involved in seasonal sea ice loss. Evolution of the marginal ice zone is determined by complex interactions between the atmosphere, sea ice, ocean, and ocean surface waves. Therefore, this region presents a significant modelling challenge. Sea ice floes span a range of sizes but sea ice models within climate models assume they adopt a constant size. Floe size influences the lateral melt rate of sea ice and momentum transfer between atmosphere, sea ice, and ocean, all important processes within the marginal ice zone. In this study, the floe size distribution is represented as a power law defined by an upper floe size cut-off, lower floe size cut-off, and power law exponent. This distribution is also defined by a new tracer that varies in response to lateral melting, wave induced break-up, freezing conditions and advection. This distribution is implemented within a sea ice model coupled to a prognostic ocean mixed layer model. We present results to show that the use of a power law floe size distribution has a spatially and temporally dependent impact on the sea ice, in particular increasing the role of the marginal ice zone in seasonal sea ice loss. This feature is important in correcting existing biases within sea ice models. In addition, we show a much stronger model sensitivity to floe size distribution parameters than other parameters used to calculate lateral melt, justifying the focus on floe size distribution in model development. We also find that the attenuation rate of waves propagating under the sea ice cover modulates the impact of wave breakup on the floe size distribution. It is finally concluded that the model approach presented here is a flexible tool for assessing the importance of a floe size distribution in the evolution of sea ice and is a useful stepping-stone for future development of floe size modelling.

## 1 Introduction

Arctic sea ice is an important component of the climate system. The sea ice cover moderates high latitude energy transfers between the ocean and atmosphere (Screen et al., 2013) and generates a positive feedback response to global warming via the albedo feedback mechanism (Dickinson et al., 1987; Winton, 2006, 2013). Accurate representation of the sea ice within climate models can contribute to improved projections of the climate response to present and future forcings (Vihma, 2014). On a more local scale sea ice modelling is necessary to understand how environments within and around the Arctic are likely to develop. This is important for Arctic communities to plan for the future (Laidler et al., 2009), to enable ecologists to identify practical responses to protect vulnerable species that live in the Arctic or seasonally migrate into the region (Hauser et al., 2017; Post et al., 2009; Regehr et al., 2010), and shipping companies to understand the potential viability of new routes in the next few decades (Aksenov et al., 2017; Ho, 2010; Smith and Stephenson, 2013).

The Arctic is currently in a state of transition (Notz and Stroeve, 2018; Stroeve and Notz, 2018). Multiyear sea ice fraction has decreased by more than 50% with an increasing proportion of the ice cover now seasonal first year ice (Kwok, 2018; Maslanik

et al., 2007). First year ice does not have the same surface roughness or the same mechanical or thermophysical (salinity, conductivity, permeability) properties as ice that has developed over multiple years. In particular, first year ice is thinner and weaker (Stroeve et al., 2018) and hence more vulnerable to fracture in response to external stress (Zhang et al., 2012). Similarly the region of the Arctic identified as the marginal ice zone (MIZ), generally defined as the region where ocean waves are able

to significantly influence the dynamics of the sea ice (Strong et al., 2017), is projected to increase in extent (Aksenov et al., 2017). An alternative definition of the MIZ, and the one that will be used in the present study, is the region where the concentration of the sea ice extends between 15 and 80 %. This definition of the MIZ is often more practical for modelling and observational studies where sea ice concentration data is more readily available than information about wave behaviour in sea ice.

Modelling the MIZ is a significant challenge due to its complexity; it is a region in which there is strong coupling between the sea ice, ocean and atmosphere (Lee et al., 2012; McPhee et al., 1987). The sea ice cover in this region is significantly broken up and fragmented by the waves that define the MIZ (Liu et al., 1992). Wave intensity and storm frequency are projected to increase, which will strengthen wave-sea ice interactions (Casas-Prat et al., 2018; Day and Hodges, 2018). This continues a trend already observed over the past few decades (Stopa et al., 2016). Such interactions are even more prominent around

Antarctica due to the dominance of seasonal sea ice in the region (Parkinson and Cavalieri, 2012) and large and increasing wave fetch (Young et al., 2011).

Floe size is a key parameter in describing the evolution of the MIZ (Rothrock and Thorndike, 1984). As sea ice floes become smaller the available perimeter per unit area of sea ice cover increases, enhancing the lateral melt rate (Steele, 1992). Increased lateral ice melt increases the area of exposed ocean, allowing the input of more heat into the ocean mixed layer from solar

insolation.  Warming of the upper mixed layer also re-stratifies the ocean. These two processes increase heat available for ice melt through basal and lateral ice melting mechanisms. The former is a well-known mechanism, the albedo feedback (Curry et al., 1995). As the MIZ expands, the lateral ice melting is expected to become an increasingly significant driver of seasonal ice loss.

Currently climate models either assume a fixed and constant characteristic floe size across the Arctic cover, for all types of sea

ice (Hunke et al., 2015), or they ignore floe size entirely. This approach does not allow for regional or temporal variations in floe size. Multiple sea ice processes depend on floe size. Lateral melt rate is a function of floe size; the melt rate is proportional to the perimeter per unit area of sea ice. A recent study has found that the basal melt rate may also be influenced by floe size (Horvat and Tziperman, 2018). Floe size can also impact the propagation of waves under the sea ice (Boutin et al., 2018; Meylan and Squire, 1994; Squire, 2007). The assumption of a fixed floe size also prevents sea ice models from accurately

representing the impact of processes on the sea ice evolution that act via the perturbation of floe size such as lateral melting and wave induced fragmentation of floes. Whilst these assumptions are significant, the use of a variable floe size within models will need to be justified against the increased computational cost. The most suitable modelling approach will be context dependent; for example, high resolution regional sea ice models would be expected to require a higher complexity of floe size treatment than large scale climate models.

There have been several observational studies aiming to characterise the floe size distribution (FSD) using techniques including satellite imagery and in-situ studies (Stern et al., 2018a). FSD data is generally fitted to a power law (Rothrock and Thorndike, 1984). Values have been reported for the magnitude of the exponent of this power law ranging from 1.5 to over 3.5 between different datasets (Stern et al., 2018a). Comparing these observations is complicated by the fact that some studies report a value for the probability distribution of floe size, and some for the cumulative floe size distribution. It has been recently pointed

out that if a distribution adopts a power law for a probability distribution, it will have a tailing off for larger floes when plotted as a cumulative distribution (Stern et al., 2018a). Furthermore, a recent study (Stern et al., 2018b) found evidence to suggest that the exponent of the power law FSD evolves throughout the year and is not fixed. This same study was also able to use two satellite data sets with different resolutions but operating over the same region to show floes from as small as 10 m and as

large as 30,000 m follow power laws. Other studies find different values for these limits, for example Toyota et al. (2016) showed a power law extending to 1 m (using data collected *in situ* from a ship), whereas Hwang et al. (2017) found a tailing off from the power law around 300 – 400 m. As each study operates over a different spatial extent, with a different resolution and different algorithms used to extract the FSD, it is not trivial to identify whether the cut-offs in each scenario are physical

or a product of limited resolution or spatial extent. Alternative approaches to a single power law have been proposed including the use of two power laws over different size ranges, with smaller floes found to have a smaller exponent (Steer et al., 2008). The Pareto distribution has also been discussed (Herman, 2010); it is analogous to a power law but with a non-constant exponent. To fully understand and characterise the FSD across the Arctic sea ice good spatial and temporal coverage is required. Novel techniques, particularly those using autonomous platforms and robotic instruments, are enabling increased

high-resolution data capture of sea ice and ocean conditions that can be used alongside time series of up to 1m-resolution FSD data obtained through remote sensing to better understand the factors driving FSD evolution (Thomson and Lee, 2017). This data could be applied within an approach analogous to Perovich and Jones (2014), who used aerial photography alongside simple parameterisations for lateral melting and floe fragmentation by waves, assuming the floe size cumulative distribution adopts a power law, to explore whether these processes could result in the observed changes to the FSD. There are also efforts

to characterise the floe size distribution resulting from individual processes, such as laboratory analogues to the wave break-up of ice (Herman et al., 2018). Future Arctic expeditions including "Multidisciplinary drifting Observatory for the Study of Arctic Climate" (MOSAiC, Dethloff et al., 2016), planned to last one year within the central Arctic, should contribute to the existing FSD datasets.

Modelling studies have used contrasting approaches to represent floes as a distribution. A very simple approach is the use of

a semi-empirical relationship between floe size and sea ice concentration (Lüpkes et al., 2012; Tsamados et al., 2015). Although this approach involves a simple amendment to the code and has a negligible computational cost, it is unable to respond to fragmentation processes. It will not capture the desired feedbacks during events such as storms that are expected to produce significant fragmentation of the sea ice cover. Furthermore, the parameters used within the relationship were constrained by a set of observations from a specific region and season and might not be applicable across the whole sea ice extent and full

seasonal cycle.

Extending beyond using this simple dependency of floe size on sea ice concentration, Zhang et al. (2015) introduced a thickness, floe size and enthalpy distribution. This model aims to represent the impacts on floe size of advection, thermodynamic growth, lateral melting, ice ridging and ice fragmentation. However, the impacts of wind, current and wave forcing are represented by an empirically parameterised floe size distribution factor. Bennetts et al. (2017) focus on the

incorporation of a physically realistic wave-induced break-up model (Williams et al., 2013a, 2013b). Bennetts et al. (2017) assume that the FSD follows a split power law, with a change in exponent at some critical diameter. The wave component of this model assumes steady-state conditions over a timestep and uses a Bretschneider spectrum defined by a significant wave height and a peak period for computational efficiency and propagates it in the mean wave direction. The propagation directions are calculated from averages of the wave directions entering the neighbouring cells and weighted according to the respective

wave energy. The model implementation also assumes floe sizes to be assigned to a minimum representative diameter if ice is too thin and compliant to be broken by waves. A recent study by Boutin et al. (2019) also considers the interactions between floe size and waves within the MIZ. This study includes a fully coupled ocean surface wave model and is unique in considering the impact of momentum transfer to the sea ice from the waves via the wave radiative stress.

There has also been a significant drive to develop a physically derived prognostic floe size-thickness distribution (Horvat and

Tziperman, 2015, 2017; Roach et al., 2018a). A recent approach by Roach et al. (2018a) includes the representation of five processes: new ice formation; welding of floes; lateral growth; lateral melt; and fracture by ocean surface waves. This model has the advantage that it does not involve any assumptions about the form of the distribution. Provided the model incorporates good physical representations of the processes which impact floe size, the model should respond accurately to localised

extremes in behaviour (such as the large waves associated with storms), or future changes (e.g. changing wind speeds). It is also possible to model floe evolution at the floe by floe scale, for example Herman (2018) uses a discrete-element model to investigate the wave-induced behaviour of floes.

For this study, a single power law will be applied to describe the FSD within a standalone sea ice model coupled to a prognostic mixed layer model, hereafter referred to as the WIPoFSD model (Waves-in-Ice module and Power law Floe Size Distribution model). The distribution is defined by three parameters: $d_{min}$, lower floe size cut-off for the distribution; $d_{max}$, upper floe size cut-off; and $\alpha$, the power law exponent. $\alpha$, $d_{min}$ and $d_{max}$ are set to fixed values. We also introduce a new floe size tracer, $l_{var}$, which evolves between fixed limits in response to four key processes: wave induced break-up; lateral melting; advection; and a restoring mechanism in freezing conditions. The WIPoFSD model has been selected as it is able to respond to processes that influence floe size without the computational expense of a full prognostic FSD model. The model allows an assessment of how a power law distribution of floes will impact the sea ice cover and by what mechanisms these changes occur. Furthermore, it provides a simple framework to explore the model sensitivity to the three parameters used to define the WIPoFSD. A series of additional experiments are also possible within this framework including imposing a variable exponent, changing the parameters that define the impact of waves on sea ice, and comparing the model sensitivity of the floe size parameters to other parameters that influence the lateral melt rate. A standalone sea ice model has been selected over a coupled approach to limit model complexity so that the physical impacts and feedbacks of imposing the WIPoFSD model can be more easily identified and to permit more sensitivity studies. The WIPoFSD model is coupled to a prognostic mixed layer so that mixed layer feedbacks can also be considered.

In this study we present results to understand the thermodynamic response of the sea ice to a power-law derived FSD and the individual impacts of wave-floe size and lateral melting-floe size interactions. Our focus will be on the impact of this FSD on the seasonal sea-ice retreat and variability rather than on longer term changes and trends.

This paper will proceed as follows: Section 2 describes the sea ice model used; section 2.1 describes standard model physics, and sections 2.2 – 2.4 outline the new WIPoFSD model. Section 3 describes the modelling methodology used including the forcing data and model domain. Section 4 describes the results of the simulations in three sections: section 4.1 looks at the general impacts of the FSD on the sea ice; section 4.2 explores the model sensitivity to the different FSD parameters; and section 4.3 looks at the model response to a series of perturbations to the model including the wave-in-ice setup, floe shape parameter, lateral melt constants and a variable $\alpha$. Sections 5 and 6 are the discussion and conclusion sections respectively.

## 2 Model description

For this study a CPOM (Centre for Polar Observation and Modelling) version of the Los Alamos Sea Ice model v 5.1.2, hereafter referred to as CICE, was used (Hunke et al., 2015). This is a dynamic and thermodynamic sea ice model designed for inclusion within a climate model. CICE includes a large choice of different physical parameterisations, see Hunke et al. (2015) for details. Section 2.1 outlines the features pertinent to this study. Our local version also includes some state-of-the-art parameterisations not included within the general CICE distribution, also described in section 2.1. The WIPoFSD model that we have implemented into standalone CICE is adapted from an implementation developed at the National Oceanography Centre of the UK within a coupled sea ice-ocean framework, called the NEMO-CICE-Waves-in-Ice (WIM) model (Hosekova et al., 2015; NERSC, 2016). This approach was originally developed to understand the impact of waves on the MIZ and the upper ocean via the thermodynamic and dynamic response with applications for the operational forecasting of the MIZ and large-scale coupled sea ice-ocean global modelling, where assuming a power law is particularly practical. The model includes the wave attenuation and floe break-up model based on the Waves-in-Ice Model from the Nansen Environmental and Remote Sensing Center (NERSC) Norway (Williams et al., 2013a, 2013b). An overview of this scheme is given in section 2.2. Floe size is assumed to follow a single power law within the WIPoFSD model. Three new global parameters and one tracer are required to define this power law. The global parameters are $d_{min}$, lower floe size cut-off for the distribution; $d_{max}$, upper floe

size cut-off, and $\alpha$, the power law exponent. The introduced variable FSD tracer, $l_{var}$, is a function of several processes that change floe sizes: lateral melting, wave break-up of sea ice, advection, and freeze-up. We also introduce a new floe size metric $l_{eff}$ to characterise the FSD, the effective floe size. Section 2.3 outlines how the imposed FSD is defined and describes amendments made to model thermodynamics to account for the change in floe size treatment. This section also provides a
definition of $l_{eff}$. Further details about the treatment of floe size and how $l_{var}$ evolves are given in section 2.4.

## 2.1 Description of Standard Model Physics

Within the CICE v 5.1.2 model we use the incremental remapping advection scheme (Lipscomb and Hunke, 2004), an ice thickness redistribution scheme (Lipscomb et al., 2007), along with 5 ice thickness categories (Hunke et al., 2015). The default elastic-viscous-plastic (EVP) rheology is used (Hunke and Dukowicz, 2002) along with an ice strength formulation (Rothrock,
1975). The frictional energy dissipation parameter is set to 12. A topological based melt pond scheme is used (Flocco et al., 2012) in conjunction with a Delta-Eddington radiation scheme (Briegleb and Light, 2007). The atmospheric and oceanic neutral drag coefficients are assumed constant in time and space. An ocean heat flux formulation is used at the ice-ocean interface (Maykut and McPhee, 1995).

The rate of thermodynamic ice loss is calculated as follows (Maykut and Perovich, 1987; Steele, 1992),

$$\frac{d}{dt}(AH) = A\left[ w_{top} + w_{bas} + \frac{\pi H}{\alpha_{shape}L} w_{lat} \right],\qquad(1)$$

where $A$ refers to the sea ice concentration, $H$ to the ice thickness, $L$ refers to the floe diameter (300 m in the default set up) and $\alpha_{shape}$ is a geometrical parameter to represent the deviation of floes from having a circular profile (0.66 in the default set up). The terms $w_{top}$, $w_{bas}$ and $w_{lat}$ refer to the melt rate at the floe upper surface (top melt), base (basal melt) and sides (lateral melt). The lateral melt rate is calculated as follows:

$$w_{lat} = m_1 \Delta T^{m_2}.\qquad(2)$$

Here $m_1 = 1.6 \times 10^{-6}\ m\ s^{-1}\ K^{-m_2}$ and $m_2 = 1.36$ (Perovich, 1983). $\Delta T$ is the elevation of the surface water temperature above freezing. The basal and top melt rates are not explicitly calculated, but instead expressed as changes in height derived from a consideration of fluxes over the top and bottom floe surfaces (Hunke et al., 2015). Both lateral and basal melting are reliant on there being sufficient heat flux from the ocean to the sea ice to produce the predicted melting. The model calculates a melting
potential term, $F_{frzmlt}$, for the upper ocean layer. If $F_{frzmlt} < 0$ in a grid cell where sea ice is present, lateral and basal melting will occur. $F_{frzmlt}$ is proportional to the difference between the sea surface temperature and sea ice freezing temperature (up to a maximum limit of 1000 Wm$^{-2}$). If the total heat flux required to produce the calculated basal and lateral melt exceeds the value permitted by the melting potential, then both values will be reduced proportionally such that the total heat flux required equals $F_{frzmlt}$. Note that $H$ stays constant with respect to lateral melt, so discarding the $w_{top}$ and $w_{bas}$ terms in Eq. (1) we have
an expression for the rate of sea ice concentration loss via lateral melt,

$$\frac{1}{A}\frac{dA}{dt} = \frac{\pi}{\alpha_{shape}L} w_{lat.}\qquad(3)$$

In these simulations, the default CICE fixed slab ocean mixed layer (ML) is not used, and instead a prognostic mixed layer model is used wherein the temperature, salinity and depth of the layer are all able to evolve with time (Petty et al., 2014). These variables evolve based on surface fluxes and entrainment/detrainment at the base of the ML. The ML entrainment rate is
calculated based on the mechanical energy input by wind forcing and surface buoyancy fluxes and profiles of water properties beneath the mixed layer (Kraus and Turner, 1967). This implementation also includes a minimum ML depth, set to 10 m. The prognostic mixed layer model used here cannot capture the full extent of ocean variability, however it is sufficient to represent sea ice-mixed layer feedbacks via the mixed layer properties. Tsamados et al. (2015) have previously compared the performance of the prognostic ML model used here to observations (Peralta-Ferriz and Woodgate, 2015). The mixed layer was
found to be generally realistic, though shows a bias towards too shallow mixed layer depths through the melting season.

A number of amendments are made to CICE version 5.1.2 based on recent work by Schröder et al. (2019). The maximum melt water added to melt ponds is reduced from 100 % to 50 %. This produces a more realistic distribution of melt ponds (Rösel et al., 2012). Snow erosion, to account for a redistribution of snow based on wind fields, snow density and surface topography, is parameterised based on Lecomte et al. (2015) with the additional assumptions described by Schröder et al. (2019). The 'bubbly' conductivity formulation of Pringle et al. (2007) is also included, which results in larger thermal conductivities for cooler ice.

## 2.2 Waves-in-ice module

The full details of this module are described in Williams et al. (2013a, 2013b), to which the reader is referred for details; here we provide an overview of the elements pertinent to our study alongside developments unique to the WIPoFSD model. The waves-in-ice module described here reproduces wave conditions near the sea ice edge within the MIZ. Local wind direction determines the direction of wave propagation with adjustments made for attenuation imposed by the sea ice cover. This is a compromise dictated by availability of forcing data, lack of observational studies and the course resolution of the CICE model. The module operates using its own internal time-step defined by:

$$t_{wav} = \frac{c \Delta x_{min}}{c_{g,max}},$$ 
(4)

where $c$ is the Courant-Friedrichs-Lewy (CFL) condition, here set to 0.7, $\Delta x_{min}$ is the size of the smallest grid cell, and $c_{g,max}$ is the highest available group velocity. This is necessary due to the high wave-speeds observed in the Arctic. Over each module time step, the wave field is advected, attenuation of waves is calculated and any ice breaking events are identified. Note also the forcing fields within each module time-step are interpolated between the prior reading and the subsequent reading to ensure smooth variations in the field (note this only applies if the grid cell remains ice-free over this period).

We construct the wave energy spectra using $H_s$, the significant wave height (m), and $T_p$ the peak wave period (s). These parameters are obtained from the ERA-interim reanalysis dataset (Dee et al., 2011). The forcings are updated at 6 hour intervals, but only for locations where the sea ice is at less than 1 % coverage, i.e. grid cells where there will be negligible wave-ice interactions. The ocean surface wave spectra, $S$ (m$^2$ s$^{-1}$), is then constructed using the 2-parameter Bretschneider formula,

$$S_B(\omega, T_p, H_s) = \frac{1.25}{8\pi} \frac{T^5}{T_p^4} H_s^2 e^{-1.25 \left(\frac{T}{T_p}\right)^4}.$$
(5)

Here $\omega$ is the frequency (rad s$^{-1}$). $H_s$ and $T_p$ are used rather than the full wave energy spectra for consistency with Williams et al. (2013a, 2013b).

Once the wave field $S$ is defined, it needs to be advected into the ice-covered regions. In the first instance this involves defining the directional space of advection. A principal direction is defined as that of the boundary surface stress component of the ocean. This is generally close to the atmospheric wind direction; however, sea ice also contributes to the boundary surface stress. The waves are advected in 5 directions spaced equally around the principal direction, with the total angular size of the surface wave spread equal to 90°. The energy is distributed amongst the bins according to $\frac{2}{\pi}(\cos \Delta\theta)^2$ where $\Delta\theta$ is the deviation from the principal wave direction. The wave energy spectra is then discretised into 25 individual frequencies from a minimum wave period of 2.5 s and a maximum of 23 s. The wave energy spectra is then advected in each defined direction using an upwind advection scheme with each individual spectrum advected separately using its group velocity $c_g(\omega)$. This advection process is necessary because the wave forcing, derived from the ERA reanalysis data, does not cover areas with a sea ice cover. Furthermore, due to differences between the modelled sea ice edge and observations, there can exist ice-free regions within the model for which no wave forcing data is available.

The decision to use the ocean surface stress to define the primary direction of wave propagation rather than the Stokes drift direction was made because the Stokes drift direction data was not available within the sea ice field at the time of model development. The use of ocean surface stress will be sensible for wind-driven seas, but not for swell-driven seas where the Stokes drift is a more appropriate choice. Stopa et al. (2016) discuss wave climate in the Arctic between 1992 and 2014 and they find that regions exposed to the North Atlantic wave climate will be strongly influenced by swells generated within the North Atlantic Ocean. Semi-enclosed and isolated seas e.g. Laptev, Kara are more event driven and have an equal mix of wind driven and swell driven waves. The results presented in this study should therefore be considered in the context that the direction of wave propagation is a significant approximation. Furthermore, we are only able to represent the impacts of waves generated externally to the sea ice cover within this setup. The choice of surface wave spread is also non-trivial. Wadhams et al. (2002) showed that a wave propagating into the MIZ could experience significant wave spreading until it was essentially isotropic. However, a distinction was found between wind seas where the isotropic state could be achieved within a few km and swell seas where spreading occurs much more slowly, if at all. Wave spreading has been shown to be dependent on the wavelength. Montiel et al. (2016) found that shorter wavelengths experienced spreading and longer wavelengths did not with a transition between these two regimes defined by the maximum floe size. This is consistent with the observed behaviour of wave driven regimens and swell driven regimes. Using a fixed surface wave spread across a limited number of categories is a significant simplification of the rather complex spreading behaviour of waves, however it represents a balance between short wave periods that quickly achieve an isotropic state and longer wave periods that propagate much further into the MIZ before they experience significant spreading.

After advection, the attenuation of waves over each wave timestep is calculated. This will be calculated for each individual wave energy spectrum:

$$S_{at}(\omega) = S(\omega)e^{-\alpha_{dim}c_g(\omega)t_{wav}}, \tag{6}$$

where $S_{at}$ is the wave spectrum after attenuation (m$^2$ s$^{-1}$), $\alpha_{dim}$ is the dimensional attenuation coefficient (m$^{-1}$), $t_{wav}$ is the module timestep (s), and other variables are as previously defined. $\alpha_{dim}$ can also be described as the rate of exponential attenuation per metre. It is here modelled as a sum of the linear wave scattering at floe edges in addition to a viscosity term. It is also updated discontinuously when the wave energy is large enough to cause ice breakage. $\alpha_{dim}$ effectively becomes a function of mean floe size, sea ice concentration, ice thickness and wave period (see Williams et al., 2013a, for further details). After attenuation, the wave energy spectra within each grid cell is reconstructed as a discretised function of $\omega$ by summing the advected spectra from each of the 5 incident directions. The final spectra, $S(\omega)$, can then be advected using the process described above for subsequent time steps. If we assume that the sea surface elevation follows a Gaussian distribution i.e. non-linear affects that can cause asymmetry are neglected, we can calculate the following properties of interest from the wave energy spectra: the mean square surface elevation of the ocean, $\langle\eta^2\rangle$; the mean square surface elevation of the sea ice, $\langle\eta^2_{ice}\rangle$; the mean square strain for the sea ice (modelled as a thin elastic plate), $\langle\varepsilon^2\rangle$; and the representative wave period, $T_W$. Each of these metrics requires the computation of integrals over frequency, here approximated using Simpson's rule (see Williams et al., 2013a, for further details). $H_s$, a model output, can be calculated as $4\sqrt{\langle\eta^2\rangle}$ (World Meteorological Organization, 1998).

The floe fragmentation scheme used is identical to Williams et al. (2013a), which should be referred to for a detailed description of the scheme. An overview of this scheme is presented here. Ice breaking events occur when the probability that the breaking strain amplitude, $E_s$, exceeds the breaking strain, $\varepsilon_c$, becomes larger than a critical probability, $P_{crit}$:

$$P(E_s > \varepsilon_c) = e^{\frac{-2\varepsilon_c^2}{E_s^2}} > P_{crit}. \tag{7}$$

We assume that the spectrum is narrow enough to be considered monochromatic. In this case $P_{crit} = e^{-1}$ and the criterion reduces to $E_s > \varepsilon_c\sqrt{2}$. $E_s$ is defined as $2\sqrt{\langle\varepsilon^2\rangle}$, i.e. twice the standard deviation in strain. $\varepsilon_c$ is calculated as $\frac{\sigma_c}{Y^*}$, where $\sigma_c$ is the flexural strength and $Y^*$ the effective Young's modulus for the sea ice. $\sigma_c$ and $Y^*$ are calculated using empirically derived expressions, where both are dependent on the brine volume fraction.

$T_W$ is used to calculate the representative wavelength, $\lambda_W$, required to update the FSD after a wave fragmentation event (see section 2.4 for details on how the FSD is changed). $\lambda_W$ is calculated as $\frac{2\pi}{k_W}$, where $k_W = k_{ice}\left(\frac{2\pi}{T_W}\right)$. Here $k_{ice}(\omega)$ is the positive real root of the dispersion equation for a section of ice-covered ocean.

**2.3 Floe size distribution model**

We employ a number-weighted FSD, $N(x)$, where x is the floe diameter. $N(x)$ is fitted to a power law as shown in fig. 1. It is described by the following equation:

$$N(x \mid d_{min} \le x \le l_{var}) = Cx^{-\alpha}. \tag{8}$$

Where $N$ has units $m^{-1}$, $d_{min}$ and $l_{var}$ have units $m$, and $\alpha$ is unitless. $l_{var}$ is the variable FSD tracer, also in m. $l_{var}$ evolves independently in each grid cell as a function of physical processes between the upper and lower floe size cut-offs of the

distribution, $d_{max}$ and $d_{min}$ respectively. $d_{max}$ also has units of m. $d_{min}$, $d_{max}$ and $\alpha$ can all be defined independently for each grid cell, however in this study they will be fixed across the sea ice cover within an individual simulation. $l_{var}$ can be considered to represent the history of a given area of sea ice in terms of physical processes that effect the FSD.

The model is initiated with $l_{var}$ set to $d_{max}$ in all grid cells where sea ice is present. The floe size number distribution factor, $C$, is determined such that the total area of individual floes, $N\alpha_{shape}x^2$, sum to equal the total sea ice area, $Al^2$ (where $l^2$ is

the total grid cell area):

$$\frac{\alpha_{shape}}{Al^2} \int_{d_{min}}^{l_{var}} Nx^2 \, dx = 1. \tag{9}$$

It should be noted this treatment of $N$ means that in this model the sea ice cover consists only of floes between the limits of $d_{min}$ and $d_{max}$ in diameter. There are no floes with sizes outside these limits.

It is useful here to define an additional floe size parameter, $l_{eff}$, the effective floe size. $l_{eff}$ is defined as the floe size of a

distribution of identical floes that would produce the same lateral melt rate in a given instant to a distribution of non-uniform floes, when under the same conditions with the same total sea ice cover. Equation (3), used to calculate the lateral melt rate, can be adapted for use within the WIPoFSD model:

$$\frac{1}{A}\frac{dA}{dt} = \frac{\pi}{\alpha_{shape}l_{eff}}w_{lat}. \tag{10}$$

The lateral melt rate of a given area of sea ice is proportional to the total perimeter of that sea ice. It is therefore also useful to

introduce a second parameter called perimeter density, $\rho_P$, which is the length of the ice edge per unit area of sea ice cover. $l_{eff}$ is hence the constant floe size which produces the same $\rho_P$ as an FSD.

First, Eq. (8) and Eq. (9) can be used to give an expression for the total sea ice area, $Al^2$:

$$Al^2 = \int_{d_{min}}^{l_{var}} \alpha_{shape}x^2 * Cx^{-\alpha}dx. \tag{11}$$

The total ice edge length, $P_{fsd}$, within a grid cell, can also be expressed in terms of the WIPoFSD parameters:

$$P_{fsd} = \int_{d_{min}}^{l_{var}} \pi x * Cx^{-\alpha}dx. \tag{12}$$

We can then divide the second expression by the first to give $\rho_P^{fsd}$, which is $P_{fsd}$ divided by the total ice area in the grid cell, $Al^2$:

$$\rho_P^{fsd} = \frac{P_{fsd}}{Al^2} = \frac{\pi(3-\alpha)[l_{var}^{2-\alpha} - d_{min}^{2-\alpha}]}{\alpha_{shape}(2-\alpha)[l_{var}^{3-\alpha} - d_{min}^{3-\alpha}]}. \tag{13}$$

Whilst perimeter density has not been a standard parameter to report from observations, it can be easily calculated from

available FSD data. A similar value has been reported by Perovich (2002), though this was reported per unit area of domain

size (i.e. ocean plus sea ice area). We can then also define $\rho_P^{con}$, the perimeter density for a distribution of floes of constant size, using an analogous approach:

$$\rho_P^{con} = \frac{P_{con}}{Al^2} = \frac{\pi}{\alpha_{shape}L}; \tag{14}$$

$L$ corresponds to the constant floe size, hence for the 300 m case we would get a perimeter density of 0.0159 m$^{-1}$. Setting the perimeter density expressions for both a constant floe size and power law FSD to be equal, and noting that this defines $L = l_{eff}$, we obtain:

$$l_{eff} = \frac{(2 - \alpha)[l_{var}^{3 - \alpha} - d_{min}^{3 - \alpha}]}{(3 - \alpha)\,[l_{var}^{2 - \alpha} - d_{min}^{2 - \alpha}]}. \tag{15}$$

Note that equations (13) and (15) are not valid where $\alpha = 2$ or 3. For these cases, $\alpha$ is taken to be 2.001 and 3.001 to maintain code simplicity with only a negligible cost to accuracy.

## 2.4 Processes that impact $l_{var}$

In our model there are four ways in which the floe size distribution can be perturbed: lateral melt; break-up of floes by ocean waves; advection of floes; and restoring due to freezing. Changes in $l_{var}$ impact the entire FSD via the floe number distribution factor, $C$, which is also a function of $l_{var}$, as defined in Eq. (9). Note that $C$ is also a function of sea ice concentration, and therefore for processes such as lateral melting changes in both $l_{var}$ and $A$ will contribute to changes in the floe number distribution. It should be noted here that the WIPoFSD model is not intended to represent the impact of physical processes on the details of the floe size distribution; it is indeed not possible to do so in a framework where a power law is imposed. Instead the impact of the different processes considered here are represented via parameterisations, here expressed in terms of the model variable, $l_{var}$.

As lateral melt involves the loss of ice volume from the sides of floes, it can be expected to reduce floe size. To represent this in the model, we set the reduction in $l_{var}^2$ from lateral melting to be proportional to the reduction in $A$, the sea ice concentration, from lateral melting:

$$\left(\frac{l_{var,final}}{l_{var,initial}}\right)^2 = \frac{A_{final}}{A_{initial}}. \tag{16}$$

If we then express $A_{final}$ in terms of $A_{initial}$ and $\Delta A_{lm}$, the reduction in sea ice concentration from lateral melting, we obtain:

$$l_{var,final} = l_{var,initial} \sqrt{1 - \frac{\Delta A_{lm}}{A}}. \tag{17}$$

The act of reducing $l_{var}$ alone acts to redistribute sea ice area attributed to floes larger than $l_{var}$ to floes smaller than $l_{var}$. However, the change in $A$ also independently acts to reduce $C$, as described above. The combined effect is to decrease the number of floes across the whole distribution. Previous studies, such as Horvat and Tziperman (2017), have shown that lateral melting causes stronger deviation from the power law for smaller floes than larger floes. However lateral melting also results in floes smaller than $d_{min}$ that will contribute to an even higher lateral melt relative to the floe size. Hence the behaviour of this lateral melt scheme compensates between these two expected changes to the distribution.

Section 2.4 outlines the conditions necessary to trigger the break-up of floes by waves. If these conditions are fulfilled, $l_{var}$ is updated according to the following expression:

$$l_{var} = \max\left(d_{min}, \frac{\lambda_W}{2}\right), \tag{18}$$

where $\lambda_W$ is the representative wavelength, as defined in section 2.2. Here $l_{var}$ can be considered a fragmentation length-scale, defining the transition from a regime where floes are broken up by waves to a regime where the number of floes is increasing due to this breakup of larger floes.

There are three processes thought to be the main drivers of floe formation and growth during freezing conditions: lateral growth; welding of floes; and formation of new floes (Roach et al., 2018b). The focus of this study is on the seasonal melt and fragmentation of sea ice rather than the winter evolution, hence a simple floe growth restoring scheme is used. During conditions when the model identifies frazil ice growth, $l_{var}$ is restored to its maximum value according to the following expression:

$$l_{var,final} = \min\left(d_{max}, l_{var,initial} + \frac{d_{max}\Delta t}{T_{rel}}\right),$$ (19)

where $T_{rel}$ is a relaxation time which relates to how quickly the ice floes would be expected to grow to cover the entire grid cell area. It is set to 10 days as standard. In grid cells that transition from being ice free to having a sea ice cover, $l_{var}$ is initiated with its minimum value i.e. $d_{min}$. The behaviour of the full floe number distribution depends not only on $l_{var}$ but on A, the sea ice concentration. During periods of freezing when the sea ice concentration increases significantly, both $C$ and $l_{var}$ will increase in value, leading to increases in the number density across all sizes of floes. This is consistent with a scenario where lots of new floes are being formed. During periods of freezing where the sea ice concentration does not increase significantly (e.g. where the sea ice area fraction is already close to 1), then $l_{var}$ will increase and $C$ will decrease. This represents a shift in the distribution from smaller floe to larger floes. It corresponds physically to a scenario where floe welding is the dominant process driving changes in the FSD.

$l_{var}$ is transported using the horizontal remapping scheme with a conservative transport equation, the standard within CICE for ice area tracers (Hunke et al., 2015). An amendment to the usual scheme involves calculating a weighted average of the $l_{var}$ over ice thickness categories after advection and the subsequent mechanical redistribution. This is necessary as the tracer is not defined independently for each thickness category unlike other tracer fields. It is useful here to comment on the choice of advection scheme. Firstly, properties that scale to the root of the sea ice area, such as the floe diameter, cannot be advected as an area tracer. Secondly, it has been shown that normalised or mean properties relating to the FSD also do not advect as an area-conserved property (Horvat and Tziperman, 2017). Here, $l_{var}$ is a parameter assigned to areas of sea ice to represent the prior history of that sea ice area in terms of processes that can affect the FSD. $l_{var}$ is not a parameter attributed to individual floes and it is calculated independently to the FSD and is not a diagnostic property calculated from the distribution. Hence, it is appropriate to treat $l_{var}$ as an ice area tracer.

It is worth commenting here on the limitations of the modelling approach to floe size used in this study. The use of a power law distribution with a fixed exponent to describe the FSD is a valuable simplification to explore the impact of floe size on the Arctic sea ice. The tracer $l_{var}$ is an internal model tool used to enable parameterisations of how individual processes impact the FSD within this constrained framework. The parameterisations described in this section are necessarily approximations of how these processes might impact the FSD and should not be considered exact physical descriptions.

**3 Methodology**

Our modified version of CICE is run over a pan-Arctic domain with a 1° tripolar (129 x 104) grid. The surface forcing is derived from the 6 hourly NCEP-2 reanalysis fields (Kanamitsu et al., 2002). The mixed layer properties are restored over a timescale of 5 days to a monthly climatology reanalysis at 10 m depth taken from a global ocean physical reanalysis product (Ferry et al., 2011). This restoring is needed to effectively represent advection within the mixed layer. The deep ocean post detrainment retains the mixed layer properties, however it is restored over a timescale of 90 days to the winter climatology (herein meaning the mean of January 1st conditions from 1993-2010) from the MYO reanalysis.

All simulations are spun-up between 1st January 1990 and 31st December 2004 using the standard setup described in section 2.1 with a constant floe size of 300 m (without the WIPoFSD model included). Simulations are initiated on the 1st January 2005 using the output of the spin-up and evaluated for 12 years until 31st December 2016. Results are all taken from the period 2007 – 2016 to allow 2 years for the model to adjust to the addition of the WIPoFSD model. A reference run is also evaluated

over this period using the standard setup and a 300 m constant floe size. Figure 2 shows this model simulates the climatological monthly sea ice extent realistically for this period. All further simulations are evaluated over the same time period using the same initial model state, however with the WIPoFSD model imposed. Some simulations have additional modifications made to the model as described.

## 4. Results

Results are presented for the pan-Arctic domain with a focus on the melting season. All plots compare the mean behaviour over 10 years from 2007 to 2016 against the reference simulation, referred to as *ref*, which uses a constant floe size of 300 m. The results for 2005 and 2006 are discarded to allow two years for the model to adjust to the imposed FSD. In this study we are trying to understand the impact of the FSD and associated processes on the seasonal sea ice loss. 2007 – 2016 has been selected as the baseline for these simulations as it will capture the current climatology of the Arctic, including the record September minimum sea ice extent observed in 2012.

### 4.1 General impact of an imposed distribution

The WIPoFSD model introduces new parameters that can be constrained through observations. Stern et al. (2018b) were recently able to show a region of floe sizes could be described by power laws over a size range from 10 to 30,000 m. This is the largest range of floe sizes that a power law has produced a good fit to, hence these are set as the standard values for $d_{min}$ and $d_{max}$ in this study. A collated analysis of observations (Stern et al., 2018a) shows that $\alpha$ can adopt values generally ranging from 1.6 to 3.5 (when the FSD is reported as a probability distribution). A standard exponent value of $\alpha = 2.5$ is adopted as an intermediate value over this range, noting in addition that this value is consistent with the ranges reported by Stern et al. (2018b). The simulation using these standard FSD parameters, $\alpha = 2.5$, $d_{min} = 10\ m$, $d_{max} = 30,000\ m$, will be referred *stan-fsd* (see table 2).

Figure 3 displays the percentage difference in sea ice extent and volume for *stan-fsd* compared to *ref*. In addition, it shows the spread of twice the standard deviation of these simulations as a measure of the interannual variability. The impact on the pan-Arctic scale is small with sea ice extent and volume reductions of up to 1.2 %. The difference in sea ice area reaches a maximum in August whereas the difference in sea ice volume peaks in September. The differences in both extent and volume evolve over an annual cycle, with minimum differences of -0.1 % and -0.2 % observed respectively between December to January for ice area and April to May for volume. The annual cycles correspond with periods of melting and freeze-up and is a product of the nature of the imposed FSD. The interannual variability shows that the impact of the WIPoFSD model with standard parameters varies significantly depending on the year. In some years the difference between the *stan-fsd* and *ref* set-ups can be negligible, in other years it can be up to 2 %. Lateral melt rates are a function of floe size but freeze-up rates are not and hence model differences only increase during periods of melting and not during periods of freeze-up. The difference in sea ice extent reduces rapidly during the freeze-up conditions; this is a consequence of the fact this lateral freeze-up behaviour is predominantly driven by ocean surface properties, which are strongly coupled to atmospheric conditions in areas of low sea ice extent. In comparison, whilst atmospheric conditions initiate the vertical sea ice growth, this atmosphere-ocean coupling is rapidly lost due to insulation of the warmer ocean from the cooler atmosphere once sea ice extends across the horizontal plane. Hence a residual difference in sea ice thickness and therefore volume propagates throughout the winter season. The difference in sea ice extent shows an additional trough in June. This feature is something also seen consistently within the data for individual years and can most likely be attributed to particular weather patterns that occur during the spring season.

Figure 4 shows the absolute difference in the mean cumulative annual melt components between the two simulations. The plot shows lateral, basal, top and total melt (as defined in section 2.1). A large increase can be seen in the lateral melt, however the change in total melt is negligible. This is because the lateral melt increase is largely compensated by a reduction in basal melt. The top melt also shows a negligible change.

Figure 4 also shows the change in basal melt in *stan-fsd* only accounting for the loss of basal surface area available for melting. To explain how this is calculated, imagine for a given time step the sea ice fraction for that grid cell in the *stan-fsd* simulation is 0.81 and in the *ref* simulation it is 0.90. If this physical reduction is the only factor causing changes to the total basal melt, then the basal melt rate per unit grid cell area would also reduce by the same factor of 10% from *ref* to *stan-fsd*. The reduction in the total basal melt volume can then be calculated for this grid cell accounting only for the reduction in sea ice fraction as the product of 0.1, the basal melt rate per unit grid cell area, and the area of the grid cell. This process can be repeated over every grid cell to obtain the total reduction in basal melt volume accounting only for reduction in sea ice concentration. The agreement (to within one standard deviation) between this synthetic reduction in basal melt and the actual reduction in basal melt suggests that the loss of ice area by lateral melt is sufficient to explain most of the basal melt compensation effect. Figure 5 shows the spatial distribution for the predicted reduction in basal melt from *stan-fsd* to *ref*, the actual reduction in basal melt, and the difference between the actual reduction and predicted reduction in basal melt. These map plots are presented as monthly averages for March, June and September averaged over 2007 – 2016. Figure 5 shows that the predicted basal melt can capture the regional distribution of the changes in basal melt from *ref* to *stan-fsd*, not just the area-integrated quantity.

Figure 6 explores the spatial distribution in the changes in ice extent and volume for three months over the melting season, March, June and September. Data is shown only for regions where the sea ice cover exceeds 5 % of the total grid cell. These results show the differences increase in magnitude through the melting season. Although the pan-Arctic differences in extent and volume are marginal, Fig. 6 shows distinct regional variations in sea ice area and thickness metrics. Reductions in the sea ice concentration and thickness are seen both within and beyond the MIZ with reductions of up to 0.1 and 50 cm observed respectively in September. Within the pack ice, increases in the sea ice concentration of up to 0.05 and ice thickness of up to 10 cm can be seen. In September the biggest increases in thickness are directed along the North American coast, particularly within the Beaufort Sea. To understand the non-uniform spatial impacts of the FSD, it is useful to look at the behaviour of $l_{eff}$. Regions with an $l_{eff}$ greater than 300 m will experience less lateral melt than the equivalent location in *ref* (all other things being equal) whereas locations with an $l_{eff}$ below 300 m will experience more lateral melt. The distribution of $l_{eff}$ is shown in Fig. 6 where in general we see a transition from larger floes to smaller floes moving from the pack ice into the MIZ, with the transition to an $l_{eff}$ of a size less than 300 m observed within the MIZ. Most of the sea ice area must therefore experience less lateral melting compared to *ref*. This result shows that the increase in lateral melt observed in Fig. 4 is localised to regions where the sea ice concentration is around 50% or below.

## 4.2 Exploration of the parameter space

It has been previously discussed that the floe size parameters used within the WIPoFSD model are poorly constrained by observations. In this section experiments are performed using different permutations of these parameters to assess model sensitivity to the form of the FSD. It is valuable to consider how changes to each FSD parameter is likely to impact the distribution: increasing the magnitude of $\alpha$ increases the number of small floes in the distribution and reduces the number of larger floes; increasing $d_{min}$ removes smaller floes from the distribution entirely, increasing the number of floes across the rest of the distribution; increasing $d_{max}$ adds larger floes to the distribution, reducing the number of floes across the rest of the distribution.

For the first study the $\alpha$ is changed from 2.5 to 3.5, previously identified as the most extreme value within a reasonable observed range for the power law exponent. This simulation will be referred to as (A). Figure 7 is analogous to Fig. 4, comparing the component and total melt evolution for an FSD with an $\alpha = 3.5$ compared to one with an $\alpha = 2.5$ (with $d_{min}$ and $d_{max}$ set to standard values). The plot shows an increase in the cumulative lateral melt, as seen before for *stan-fsd* compared to *ref*. Now, however, the basal melt is less effective at compensating the lateral melt resulting in a significant increase in the total melt. There is also now a non-negligible reduction in the top melt, with the interannual variability showing the increase in total melt and reduction in top melt is consistently produced for each year of the simulations. The difference in

cumulative total melt reaches a maximum in August and subsequently decreases slightly. This suggests that increasing the magnitude of $\alpha$ results in an earlier melting season and a correspondingly reduced melt in the late season. The predicted change in basal melt based on the reduced sea ice area is again plotted and is able to account for 90% of the actual reduction in basal melt. This is in contrast to Fig. 4, where the predicted reduction in basal melt was too high compared to the simulated reduction.

The interannual variability shows that this underprediction of the reduction in basal melt is consistent throughout individual years. This implies the presence of additional mechanisms such as albedo and other mixed layer feedbacks causing non-negligible changes in the basal melt rate, however reduction in the sea ice concentration remains the leading order impact. Figure 8 shows difference map plots between the two simulations. The ice area and thickness are reduced across the sea ice cover with reductions of over 5 % and 0.5 m respectively seen in particular locations during September. However, even in

March, after the freeze-up period, reductions of 0.1 m or more in sea ice thickness can be seen within the ice pack. The response of sea ice can once again be understood through the behaviour of the $l_{eff}$. $l_{eff}$ is below 30 m across the entire ice cover throughout all three months studied, leading to increased lateral melt rates across the sea ice.

A further 17 sensitivity studies using different permutations of the parameters have been completed. These are formed by varying the three key defining parameters of the FSD shown in Fig. 1 in order to span the range of values reported in

observational studies. For $\alpha$ values of 2, 2.5, 3 and 3.5 to span the general range of values reported in observations (Stern et al., 2018a). For $d_{min}$ values of 1 m, 20 m and 50 m are selected. These have been selected to reflect the different behaviours reported in studies, with some showing power law behaviour extending to 1 m (Toyota et al., 2006) and others showing a tailing off at an order of 10 s of m (Stern et al., 2018b). A further limitation for $d_{min}$ is the smallest floe size where individual floes can be distinguished i.e. the transition from a floe regime to a brash ice regime. For the upper cut-off, $d_{max}$, values of

1000 m, 10,000 m, 30,000 m and 50,000 m are selected, again to represent the distributions reported in different studies. 50 km is taken as the largest value for $d_{max}$ as this serves as an upper limit to what can be resolved within an individual grid cell on a CICE 1° grid. In addition, this model does not account for processes that are expected to be important for the evolution of floes at km scale and above, such as wind stresses and melt ponds (Arntsen et al., 2015; Wilchinsky et al., 2010).

of the 17 permutations for these sensitivity studies are generated by selecting all the different $\alpha$-$d_{min}$ permutations (except

the two already investigated). Each of these simulations has $d_{max} = 30000$ m. The further three simulations vary $d_{max}$ with the $\alpha$ and $d_{min}$ fixed to 2.5 and 10 m respectively. Figure 9 shows the change in mean September sea ice extent and volume relative to *ref* plotted against mean annual $l_{eff}$, averaged over the sea ice extent. The impacts range from a small increase in extent and volume to large reductions of -22 % and -55 % respectively, even within the parameter space defined by observations. Furthermore, there is almost a one-to-one mapping between mean $l_{eff}$ and extent and volume reduction. This

suggests $l_{eff}$ is a useful diagnostic tool to predict the impact of a given set of floe size parameters. The system varies most in response to the changes in the $\alpha$, but it is also particularly sensitive to $d_{min}$.

**4.3 Sensitivity runs to explore specific model components and additional relevant parameters**

A series of sensitivity studies have been performed to explore the behaviour of the WIPoFSD model and understand how it interacts with other model components. Table 1 defines the important parameters considered in this section and Table 2

provides a summary of the sensitivity experiments performed. The first two entries in table 2 (*stan-fsd*) and (*ref*) refer to a standard setup using the standard FSD parameters described above and a constant floe size of 300 m respectively. Studies (A) – (C) are a selection of the simulations described in section 4.2 to allow a comparison between model sensitivity to the parameters that define the FSD and model sensitivity to other relevant parameters and components within the WIPoFSD model. In the following section a bracketed letter will follow descriptions of sensitivity studies, which corresponds to the letter

assigned in table 2.

Table 3 reports key metrics for the sensitivity studies described in table 2, plus a selection of the different sensitivity studies described in section 4.2. For each experiment the September sea ice extent and volume size are reported for both the full sea

ice extent and MIZ only (taken as a mean between 2007 – 2016), with the MIZ defined here as regions with between 15 and 80% sea ice cover. In addition, the mean cumulative lateral, basal, top and total melts until September is reported in each case, and the September mean $l_{eff}$ and mean sea ice perimeter per m² of ocean area are both reported averaged over the MIZ. For each value reported (except for the $l_{eff}$) the difference from *stan-fsd* is also stated. Cells highlighted in yellow and orange deviate by one and two standard deviation(s) respectively from the *stan-fsd* mean value (the standard deviation is calculated from the set of 10 annual values for each metric).

### 4.3.1 Imposing a variable exponent on the floe size distribution

The shape of the FSD between its limiting values is defined by $\alpha$. Recent evidence suggests this may not be constant in time or space (Stern et al., 2018b). We have investigated the impact of this behaviour through the use of two alternative modelling approaches. The first approach imposes a sinusoidal annual cycle on $\alpha$ (D):

$$\alpha = 2.35 - 0.45 \cos \frac{2\pi(d-100)}{d_{ann}}. \tag{20}$$

Here $d$ refers to the current day of the year (for example 45 would refer to 14th February) and $d_{ann}$ is the total number of days in the year (here taken to be 365). This curve was selected as a reasonable fit to the observations of Stern et al. (2018b), though it should be noted that these observations were taken from the Beaufort and Chukchi seas so should not be assumed to be representative of the entire Arctic Ocean.

The second sensitivity experiment assumes that $\alpha$ is a function of sea ice concentration, $A$ (E). This is derived from the observation that $\alpha$ increases in magnitude as the melting season advances and in locations of lower sea ice concentration:

$$\alpha = 4 - 2.1A. \tag{21}$$

The limits were selected to try and capture the variability of the exponent seen within observations.

The results in table 3 shows imposing the time-varying $\alpha$ (D) has a very small impact on the sea ice cover, whereas the spatial-varying $\alpha$ (E) causes a moderate reduction in September ice extent and volume of about 3 % and 5 % respectively. It is worth noting that the mean $l_{eff}$ over the MIZ does not correlate well with the size of the response of the system in these cases compared to simulations with a fixed $\alpha$, with $l_{eff}$ being much higher than expected given the size of the sea ice extent and volume reduction. The value of the sea ice perimeter averaged over the MIZ is more consistent with the observed changes in sea ice extent and volume, particularly for experiment (E). This shows that it is useful to have multiple approaches to collapsing the FSD into a representative value. Whilst map plots of $l_{eff}$ can be very useful for understanding the regional impacts of an FSD, as in Fig. 6, the mean value can be misleading. Figures 10 and 11 shows how $\alpha$ and the resultant $l_{eff}$ respectively evolve in these two simulations averaged over both the overall sea ice cover and the MIZ. The region spanned by twice the standard deviation of individual years within the simulation is also shown. Whilst $l_{eff}$ in both regions behaves in corresponding ways for the simulation with a time-varying $\alpha$ (D), experiment (E) shows the mean $\alpha$ and hence $l_{eff}$ within the MIZ is small and approximately constant throughout the year, despite the overall sea ice pack showing strong seasonal variability for these quantities. During the peak melting period between May and August the mean $l_{eff}$ is lower for experiment (D) within the pack ice and experiment (E) within the MIZ. Given the much stronger changes seen for experiment (E) compared to experiment (D) relative to *stan-fsd*, this supports previous findings that the impact of the WIPoFSD model is primarily dependent on the behaviour of the FSD within the MIZ. (D) shows the strongest interannual variation in $l_{eff}$ between March and May, whereas for (E) it is strongest in the peak melting season between July and August. Figure 11 also includes the annual evolution of $l_{eff}$ for the *fsd-stan* simulation. Unlike (D) and (E), *fsd-stan* shows no strong annual oscillation in the $l_{eff}$ across the overall pack ice.

### 4.3.2 Other parameters affecting the floe size distribution

The two processes currently represented in the model that actively reduce $l_{var}$ are lateral melting and wave induced fragmentation of floes. Two simulations are undertaken where either waves are no longer able to influence $l_{var}$ (F) or lateral melting is no longer allowed to influence $l_{var}$ (G). An additional three simulations are performed to focus on how waves may be influencing sea ice via reductions in $l_{var}$: the incident significant wave height at the point of entering the sea ice cover is increased by a factor of 10 (H); the floe breaking strain is reduced by a factor of 10 (I); and the wave attenuation coefficients under the sea ice are reduced by a factor of 10 (J).

The results in table 3 show that the wave-$l_{var}$ interaction is more important than the lateral melt-$l_{var}$ interaction in driving the increase in lateral melt observed by imposing the standard FSD. Study (F), where waves no longer reduce $l_{var}$, shows a 3 % increase in MIZ volume compared to *stan-fsd*, whereas study (G), where $l_{var}$ does not change as a result of lateral melt, shows an increase in MIZ volume of less than 1 %. For the three simulations performed to explore the behaviour of the wave advection model, i.e. (H), (I) and (J), the strongest response is produced by reducing the wave attenuation rate of the model (J). The weakest response is produced by increasing the ice vulnerability to wave fracture (I). Figure 12 shows difference plots of sea ice concentration and $l_{eff}$ between *stan-fsd* and (J), where the attenuation rate of waves under sea ice is reduced. The plots show a reduction in the sea ice concentration of around 1 % across the MIZ throughout the year for (J). This can be attributed to the reduction of $l_{eff}$ in the same region by magnitudes of greater than 100 m.

The floe restoring rate is the parameter, $T_{rel}$, used in Eq. (19). As a standard it is set to 10 days, however this value is not well constrained. This effectively means that $l_{var}$ restores rapidly during freezing conditions, and hence the FSD is effectively initiated in each melting season with no memory of the previous year. There is not enough evidence available to either validate or invalidate the assumption that the FSD retains no memory of the previous melting or freeze-up season. An experiment (K) has been performed where $T_{rel}$ is increased from 10 days to 365 days to explore the impact of inter-seasonal memory retention within the FSD model. The results in table 3 show that, whilst this change to the model did reduce the $l_{eff}$ and increase the perimeter density metrics by significant amounts, it did not produce a significant change in either the melt components or sea ice extent and volume.

In Figure 13 we show the evolution of simulations *stan-fsd*, (F) and (K) over 2015 averaged over selected grid cells. 2015 has been chosen as a representative year over the 2007 – 2016 period. There are two subplots: the first gives $l_{eff}$ averaged over grid cells with a sea ice concentration within the MIZ on the 31$^{st}$ August 2015, selected as the approximate date of the 2015 minimum sea ice extent in simulations. This set of grid cells is chosen to capture grid cells that are marginal for at least some of the year without also becoming ice free, which would create an artificial seasonal cycle in $l_{eff}$. For the second subplot, the same set is further constrained to grid cells with between 15% and 30% sea ice concentration on the 31$^{st}$ August 2015. Figure 6 shows that significant reductions in $l_{eff}$ are generally seen at the outer edge of the sea ice extent, so further restricting the maximum sea ice concentration in this way will capture this region. The significant reduction of $l_{eff}$ by up to 120 m between (K) and *stan-fsd* in August and September shows that the wave break-up of floes is a significant component of both the floe size reduction and the subsequent reduction in sea ice concentration seen in Fig. 6 for these locations. The difference between (K) and the maximum possible $l_{eff}$, of just over 540 m, during the melting season primarily captures the impact of lateral melting on floe size as floe restoring will not be active during this period. We see a reduction of up to 50 m for the more marginal set of grid cells, so whilst not insignificant, the impact is a factor of around 3 – 4 times lower than the wave fragmentation in these regions. This suggests that mechanical break-up of floes is a necessary precondition for the lateral melting feedback on floe size to become significant. This effect will not be as strong for other selections of FSD parameters, particular those where $l_{eff}$ is below 50 m even when $l_{var} = d_{max}$. For these simulations we expect the much larger increase in lateral melt, as seen in Fig. 7, to produce a stronger lateral melt impact on the FSD. For (K), where $l_{var}$ restoring rates during freezing conditions are reduced, $l_{eff}$ is significantly lower throughout the year including during the melting season.

$l_{eff}$ varies between 360 m – 480 m for the full MIZ grid cell selection, significantly reduced from the 450 m – 540 m seen for the *stan-fsd* simulation. We also see a well-defined seasonal cycle, unlike with *stan-fsd.*

### 4.3.3 Lateral melt parameters

The first order impact of introducing a variable floe size is on the lateral melt volume. Equation 1 shows the lateral melt volume is calculated from several parameters beyond just floe diameter, L, including lateral melt rate, $w_{lat}$, and floe shape, $\alpha_{shape}$. $\alpha_{shape}$ is currently fixed to a constant value, 0.66. There has been significantly less interest in characterising how the shape of floes varies and to characterise a floe shape distribution, particularly given available evidence suggesting floe size and shape may be uncorrelated parameters (Gherardi and Lagomarsino, 2015). Two sensitivity studies are performed: one with $\alpha_{shape}$ reduced to 0.44 (L), corresponding to 3:1 rectangular floes or similar distortions from a perfect circle; and one with $\alpha_{shape}$ increased to 0.79, corresponding to approximately circular floes (M). $w_{lat}$ is a function of two parameters, $m_1$ and $m_2$ (see Eq. 2). These parameters have been estimated from observations and hence are subject to uncertainty. Experiments are undertaken with either both $m_1$ and $m_2$ reduced by 10% (N) or both increased by 10% (O). A reduction in these parameters reduces the lateral melt rate and an increase, the converse.

Table 3 shows that all four of these sensitivity studies did not produce a large model response in terms of the overall sea ice extent and volume. Reducing the floe shape parameter (L) produced the strongest response in the lateral melt volume, and more generally the model metrics were more sensitive to $\alpha_{shape}$ than the melt coefficients, $m_1$ and $m_2$. The much stronger model sensitivity to the floe size parameters justifies the focus on floe size as the main uncertainty in lateral melt volume calculation.

### 4.3.4 Minimum mixed layer depth

The minimum ocean mixed layer depth is a constant within the prognostic mixed layer model required to prevent the mixed layer depths reaching unrealistically small values. As a standard it is set to 10 m. The depth of the mixed layer is important for the strength of mixed layer feedbacks, with a deeper mixed layer acting to damp any feedbacks via mixed layer properties. These feedbacks include the albedo feedback mechanism and the negative feedback of increased lateral and basal melts (meltwater perturbs the mixed layer properties towards less favourable melting conditions). Sensitivity studies are performed with both the minimum mixed layer depth reduced to 7 m (P) and increased to 20 m (Q).

The challenge with this set of experiments is that, unlike the other sensitivity studies presented here, it acts to influence the evolution of the sea ice both via changes in the lateral melt and also via the basal melt and sea ice freeze-up rates, determined by ocean properties. Experiment (P) shows a small increase in the total sea ice extent and volume, and (Q) a small decrease, however both result in larger increases in the MIZ extent and volume. In comparison to other sensitivity studies, the changes in the lateral and basal melt are small, suggesting that mixed layer feedbacks do not have a significant role in the impacts of the FSD found in *stan-fsd* compared to *ref*. It should be noted, however, that the evidence presented here is not enough to rule out the existence of multiple compensating feedback processes.

### 5. Discussion

We present here a series of simulations and additional sensitivity studies completed with the newly developed WIPoFSD model to explore the impacts of a variable power law derived floe size distribution model on the Arctic sea ice. It is useful to consider the physical mechanisms that drive the simulation results. It was previously noted that the increase in lateral melt observed when imposing the WIPoFSD model was compensated by a loss in basal melt, resulting in a more moderate increase in the total melt. Within the model there are three possible mechanisms causing the limited basal melt. Firstly, the increase in lateral melt will correspond to a reduction in available ice area for basal melting. It is shown in Fig. 4 and Fig. 7 that this mechanism

is able to explain most of the reduction in basal melt, but the difference remains large enough that further mechanisms need to be considered. The second mechanism concerns the melting potential of the ocean. If there is a large enough increase in the lateral melt to result in insufficient melting potential, both the lateral and basal melt will be reduced proportionally, as described in section 2.1. A simulation (not presented) to explore this impact shows it has only a limited impact on the basal melt, and

not enough to explain the observed compensation effect. The third mechanism concerns lateral melt feedback on the basal melt rate via the perturbation of mixed layer properties. Higher freshwater release from the increase in lateral melt will lower the temperature and salinity of the ocean mixed layer, which will reduce the basal melt rate. However, the lateral melt increase also reduces the ice concentration, lowering the albedo of the ice-ocean system. This increases the absorption of shortwave solar radiation into the mixed layer, raising the temperature of the mixed layer i.e. it has the opposite effect of the increased

freshwater input. These two competing feedbacks explain the overprediction of basal melt in Fig. 4 but underprediction of basal melt in Fig. 7. The increase in total melt observed in Fig. 7 will likely correspond to a more efficient use of the available melt potential and the aforementioned albedo-feedback mechanism. The interaction between the mixed layer and FSD is further explored through the (P) and (Q) sensitivity studies where the minimum mixed layer depth was reduced and increased respectively. These studies provide further evidence that mixed layer feedbacks are not a leading order effect of the FSD, given

the very small perturbations of the melt component from the *stan-fsd* simulation. Larger changes are seen for the sea ice extent and volume metrics. However, the same mixed layer feedbacks that change the melt rates can also independently influence the freeze-up rate of sea ice, hence it is not possible to directly attribute the changes produced by varying the minimum mixed layer depths specifically to WIPoFSD related feedbacks. It should also be noted that the prognostic mixed layer model used here provides a limited representation of sea ice-ocean interactions and feedbacks. The strength of these interactions may

increase within a fully coupled sea ice-ocean model (Rynders, 2017).

The series of sensitivity studies to both the floe size parameters and other aspects of the WIPoFSD model are useful to understand the limitations of the model. An important result is the limited sensitivity of the model to the $m_1$, $m_2$, and $\alpha_{shape}$ parameters, i.e. experiments (L) – (O), with significant perturbations of these parameters reducing the sea ice extent by around 1 % or less. These are additional constants needed to calculate the lateral melt rate beyond floe size. If a strong sensitivity was

found to these parameters, it would suggest that these should be considered as alternative targets rather than the FSD for future model development. Instead, these experiments support the focus on floe size as the primary uncertainty in lateral melt calculation. Experiment (K) showed very little model response to increasing the floe freeze-up timescale, T$_{rel}$, from 10 to 365 days. This result suggests that the use of more physically derived parameterisations of the floe growth during freezing conditions (e.g. Roach et al., 2018b) would not have a significant impact within the model framework presented here. However,

Fig. 13 shows that the seasonal $l_{eff}$ evolution is dependent on the floe restoring rate and there may be specific events, such as strong winter break-up events, where accurate modelling of floe growth is important to then understand the sea ice evolution during the subsequent melting season.

The sensitivity studies also give insight into the impact of waves on the sea ice cover. In particular, the two sensitivity studies that switch off the lateral melt-$l_{var}$ (G) and wave-$l_{var}$ (F) feedback mechanisms respectively showed that the latter had a

stronger influence on both the evolution of $l_{eff}$ and the changes in sea ice area and extent when imposing the WIPoFSD with standard parameters. This impact was enhanced through various perturbations to the wave model. The increase in significant wave height (H) and reduction in ice strength (I) are representative of future Arctic conditions when the sea ice is expected to be thinner (Aksenov et al., 2017) with storms of increasing strength and duration (Basu et al., 2018). The results presented here suggest that these changes will have only a limited impact on sea ice extent and volume via the floe size feedback

mechanism. The strongest response in sea ice extent and volume was observed with a reduction in the attenuation rate (J). It is important to note that the attenuation rate is a function of floe size, with smaller floes driving stronger attenuation. This creates a feedback where the fragmentation caused by one wave changes the way subsequent waves propagate through the MIZ. It should be noted that the wave component of the WIPoFSD model is a simplified representation of waves propagating

into sea ice and involves a number of approximations. In particular, the directional behaviour of the waves will be more suitable for wind waves than swell waves. As discussed in section 2.2, swell waves have been observed to have longer wavelengths and reduced attenuation rates suggesting they would interact differently with the FSD than wind waves. More generally, modelling the propagation and energy loss of waves as they travel under sea ice is a complex problem and an area of active research (Meylan et al., 2017), and there are recent efforts to produce coupled wave-sea ice models (Boutin et al., 2019; Herman, 2017). However, any increase in complexity in modelling the waves will result in increased computational cost. Further observations about wave attenuation in sea ice are needed to judge the complexity of the model approach required to produce sufficient accuracy.

As stated above, the model shows a strong sensitivity to the floe size parameters with some selections of the WIPoFSD parameters showing moderate increases in the sea ice extent and volume, and other selections driving reductions of these values by over 50 % in September. The limited observational data available to constrain the selected parameters is therefore a significant challenge of this modelling approach. Furthermore, a not insignificant model response of order 5% relative to *ref* has been observed to sensitivity experiment (E) performed here to explore the impacts of the non-uniform $\alpha$. Sensitivity experiments (D) and (E) were performed on the basis of evidence from Stern et al. (2018 a & b) that $\alpha$ is not a fixed value and instead evolves spatially and temporally. Whilst it would be interesting to explore the impact of a variable $d_{min}$, especially considering the strong sensitivity of the model response to this parameter, we do not have an analogous set of observations focusing on how $d_{min}$ may vary in space and time.

The WIPoFSD model used here assumes a power law distribution with the exponent $\alpha$, lower cut-off $d_{min}$ and upper cut-off $d_{max}$ all fixed at constant values. Each grid cell has a locally defined variable FSD tracer, $l_{var}$, which evolves in response to wave break-up events, lateral melt, and freezing conditions. The use of $l_{var}$ to represent variability within the FSD puts limits on the physical fidelity of the parameterisations of processes that change the FSD in our model. However, if $l_{var}$ is not used to represent variability in the distribution, then within a power law framework over a fixed floe size range the only other component of the system that can change is the exponent. The exponent in such a setup becomes an emergent parameter rather than one determined from observations. An important component of this study is to perform sensitivity studies of the sea ice mass balance to the range of exponents seen in observations. An investigation of the evolution of the floe size distribution itself, power law or otherwise, is better approached with a prognostic model of the proximate physical processes, such as in the manner of Roach et al. (2018a). Future improved understanding of the FSD may then allow the development of improved parameterisations of floe size and related processes that do not require the assumptions made in this study regarding the shape and floe size range of the distribution. Upcoming studies, including MOSAiC, should provide further observational evidence to develop these parameterisations. The longer-term aim is the development of a floe size model for use in climate models that can reasonably capture the physical impacts of the FSD on the complete sea ice-ocean-atmosphere system without the full complexity of the prognostic floe size-thickness distribution model. The identification of $l_{eff}$ as a useful floe size parameter may provide a method to report useful FSD information over a larger spatial and temporal scale, as this value can be calculated from the ice perimeter length within a unit area and avoids the need to report a full distribution. This would allow an assessment of the regional, intra-annual and inter-annual variability of the FSD and identify the FSD parameters and components that best reproduce these desired features. There have been recent efforts to develop techniques to obtain a representative floe size metric from satellite imagery over large spatial and temporal scales, though so far these techniques have only been demonstrated at low resolution (Horvat et al., 2019).

The reference simulation (*ref*) used in this study underpredicts summer sea ice concentration in the pack ice but overpredicts the concentration at the sea ice edge, consistent with other studies that use the CICE sea ice model (such as Schröder et al., 2019). An analysis of the historically forced simulations used within phase 5 of the Coupled Model Intercomparison Project (CMIP5) found that coupled models consistently performed poorly in capturing the regional variation in sea ice concentration, showing this problem is not specific to CPOM CICE simulations (Ivanova et al., 2016). This suggests that models currently

underestimate the role of the MIZ in driving the seasonal sea ice loss. The WIPoFSD model is shown here to have a non-uniform impact on the sea ice cover, with an enhancement in lateral melt and a corresponding reduction in sea ice concentration within the MIZ, as shown in Fig. 6. Whilst the changes are generally small, it shows that the use of an FSD model, either in the described form or otherwise, may be an important step towards improving the accuracy of sea ice models.

**6. Conclusion**

Climate model representations of sea ice currently assume that the size of floes that make up the sea ice is constant; however, observations show that floes adopt a distribution of sizes. A power law generally produces a good fit to observations of the floe size distribution (FSD), though the size range and exponent reported for this distribution can vary significantly between different studies. A power law derived FSD model including a waves-in-ice module (WIPoFSD) has been incorporated into the Los Alamos sea ice model coupled to a prognostic mixed layer model, CICE-ML. In the WIPoFSD model, the FSD is defined by a lower floe size cut-off, upper floe size cut-off and exponent. A variable FSD tracer is also introduced, which varies in response to lateral melting, wave break-up events and freezing conditions. The lower and upper floe size cut-offs and exponent are set to fixed values. A standard set of parameters for the WIPoFSD model is identified from observations and the results of a sea ice simulation using these parameters is compared to one with a constant floe size of 300 m. Inclusion of the WIPoFSD model within CICE-ML results in increased lateral melt compensated by reductions in basal melt, resulting in only moderate impacts on the total melt. The primary mechanism by which the increased lateral melt reduces the basal melt is shown to be the reduction in available ice area for basal melt. The impact is not spatially homogeneous, with losses in sea ice area and volume dominating in the marginal ice zone (MIZ). These impacts partially correct existing model biases in the standalone CICE-ML model, suggesting the inclusion of an FSD is an important step forward in ensuring that models can produce realistic simulations of the Arctic sea ice.

A series of sensitivity experiments explore the limitations of the model. The model does show a strong response to a reduction in wave attenuation rate, suggesting this is an important component in understanding wave-sea ice interactions. Different selections of parameters for the FSD show a large impact on the modelled sea ice state, with some showing a moderate increase in mean September sea ice extent and volume, with others reducing these metrics by over 20 and 50 % respectively. A newly defined parameter, effective floe size, is found to be a good predictor of model response for simulations where the lower floe size cut-off and power law exponent are fixed. The impact of a non-uniform exponent was also explored based on observations that these parameters evolve for a given region of sea ice. Results suggest that this parameter could further enhance the differential behaviour seen between pack ice and the MIZ in response to the imposition of an FSD. These sensitivity studies also showed that the choice of WIPoFSD parameters are a source of much larger model uncertainty than other constants used within the lateral melt parameterisation, justifying the focus on developing an FSD model as a priority for improved accuracy of sea ice modelling.

Whilst the model presented here does make a major assumption that the floe size distribution adopts a power law, this is consistent with most observations. Furthermore, it has been shown that the model can easily be modified to adapt to additional findings such as the inclusion of a non-uniform exponent. This means the WIPoFSD model is a useful tool for assessing the importance of the FSD in the evolution of sea ice, particularly the seasonal retreat. Climate models require an important balance to be maintained between physical fidelity and computational expense. The simplicity of the WIPoFSD model makes it a useful stepping-stone for the development of new parameterisations of floe size within climate models that can reasonably capture the physical impacts of the FSD without a large computational cost. Planned observational studies such as MOSAiC should help in the development of these novel parametrisations.

**Data Availability**

Model output used in this manuscript is publicly available via the University of Reading research data archive (http://dx.doi.org/10.17864/1947.223). Please contact the corresponding author to discuss access to the code.

**Author Contributions**

LH, with support from YA, developed the original version of the WIPoFSD model with a coupled CICE-NEMO framework. DS adapted the model into the CPOM CICE standalone setup. AB further developed the WIPoFSD model and completed the simulations and analysis under the supervision of DF, DS, LH, JR and YA. DS provided additional technical support. AB composed the paper with feedback and contributions from all authors.

**Competing interests**

The authors declare that they have no conflict of interest.

**Acknowledgements**

AB is funded through a NERC industrial CASE studentship with the UK Met Office, reference NE/M009637/1. The contributions of DF and DS were supported by NERC grant, number NE/R016690/. JR is supported by the Joint UK BEIS/Defra Met Office Hadley Centre Climate Programme (GA01 101). LH and YA acknowledge support from European Union Seventh Framework Programme SWARP (Grant agreement 607476). YA was also supported by "Towards a marginal Arctic sea ice cover" NE/R000654/1. LH and YA would also like to express gratitude towards Dr Timothy D. Williams

(Nansen Environmental and Remote Sensing Center, NERSC, Norway), Prof. Dany Dumont (Institut des sciences de la mer de Rimouski, Québec, Canada), and Prof. Vernon A. Squire (University of Otago, New Zealand) for their kind advice on ice-waves interaction. Dr Stefanie Rynders (National Oceanography Centre, Southampton) should also receive credit for also contributing to the development of the WIPoFSD model within the coupled CICE-NEMO framework alongside LH and YA. Coupled CICE-NEMO simulations were performed on the ARCHER UK National Supercomputing Service (http://www.

archer.ac.uk). We would also like to thank the Isaac Newton Institute for Mathematical Sciences for support and hospitality during the 'Mathematics of Sea Ice Phenomena' programme when work on this paper was undertaken. Finally, we would like to thank both the reviewers and the editor whose detailed and constructive comments have enabled us to significantly improve the manuscript.

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

# (a)

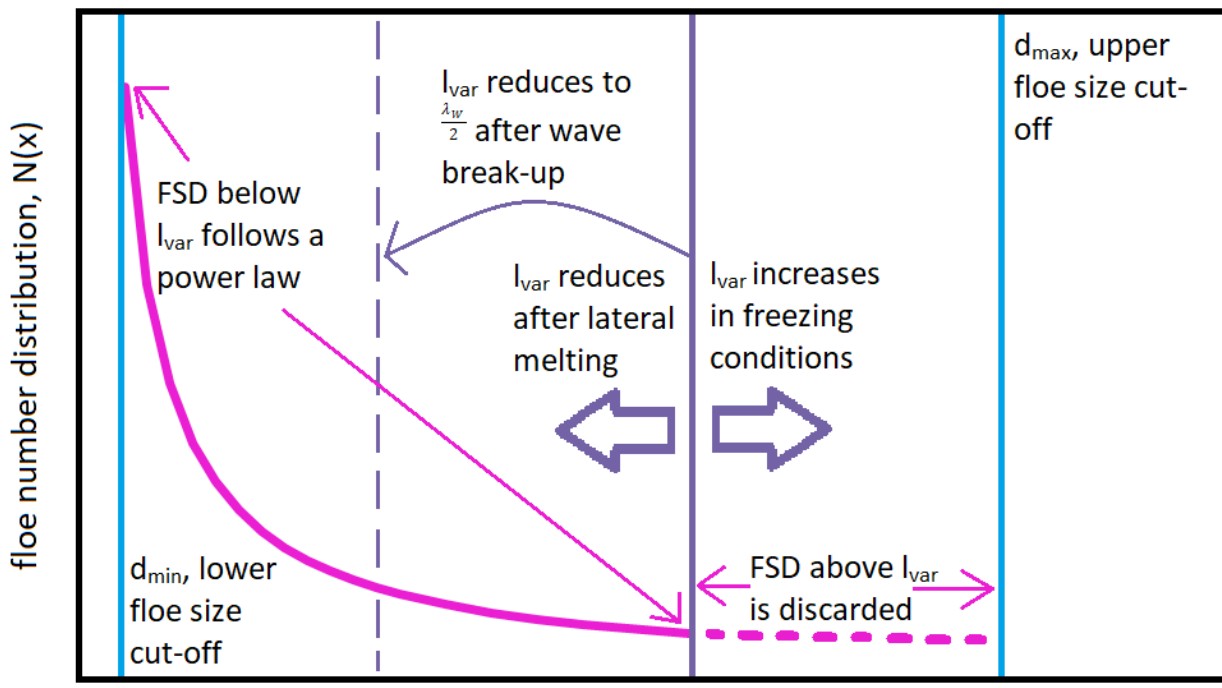

# (b)

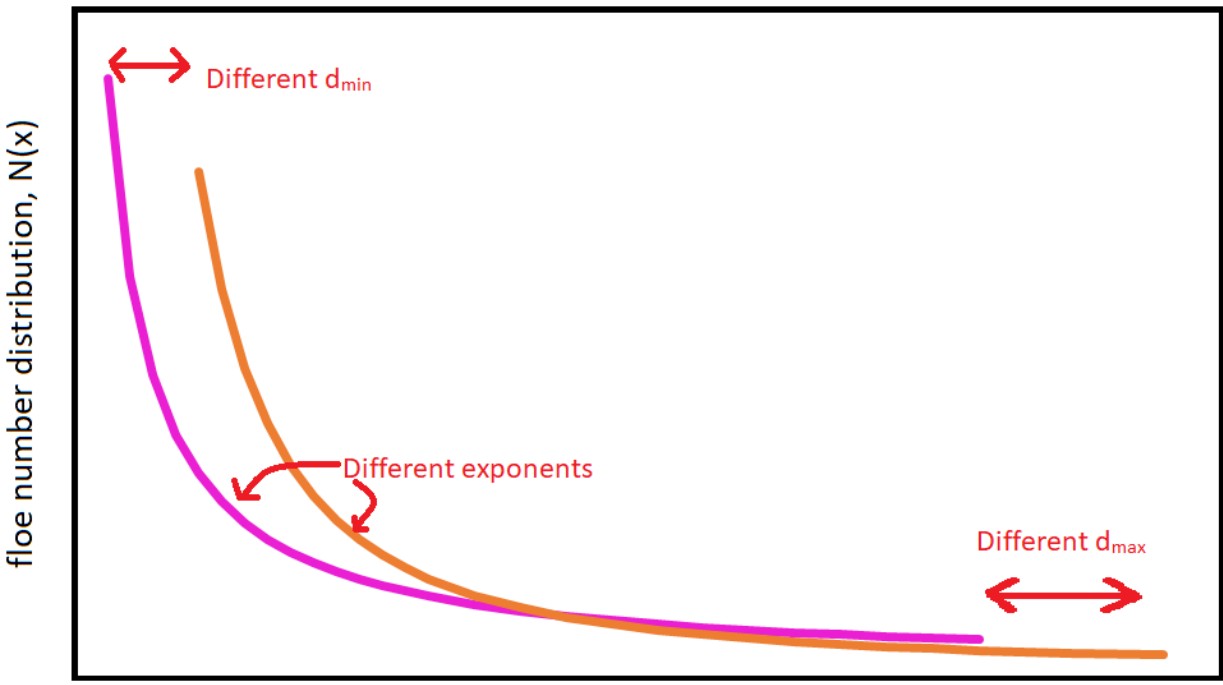

**Figure 1:** Panel (a) is a schematic of the imposed FSD model. This model is initiated by prescribing a power law with an exponent, $\alpha$, and between the limits $d_{min}$ and $d_{max}$. Within individual grid cells the variable FSD tracer, $l_{var}$, varies between these two limits. $l_{var}$ evolves through lateral melting, wave break-up events, freezing and advection. Not shown is how changes in $l_{var}$ will also impact the floe size number distribution factor, C. Panel (b) shows the how $d_{min}$, $d_{max}$ and $\alpha$ can all be varied to produce floe size number distributions. All axes within both panels are logarithmic to base 10.

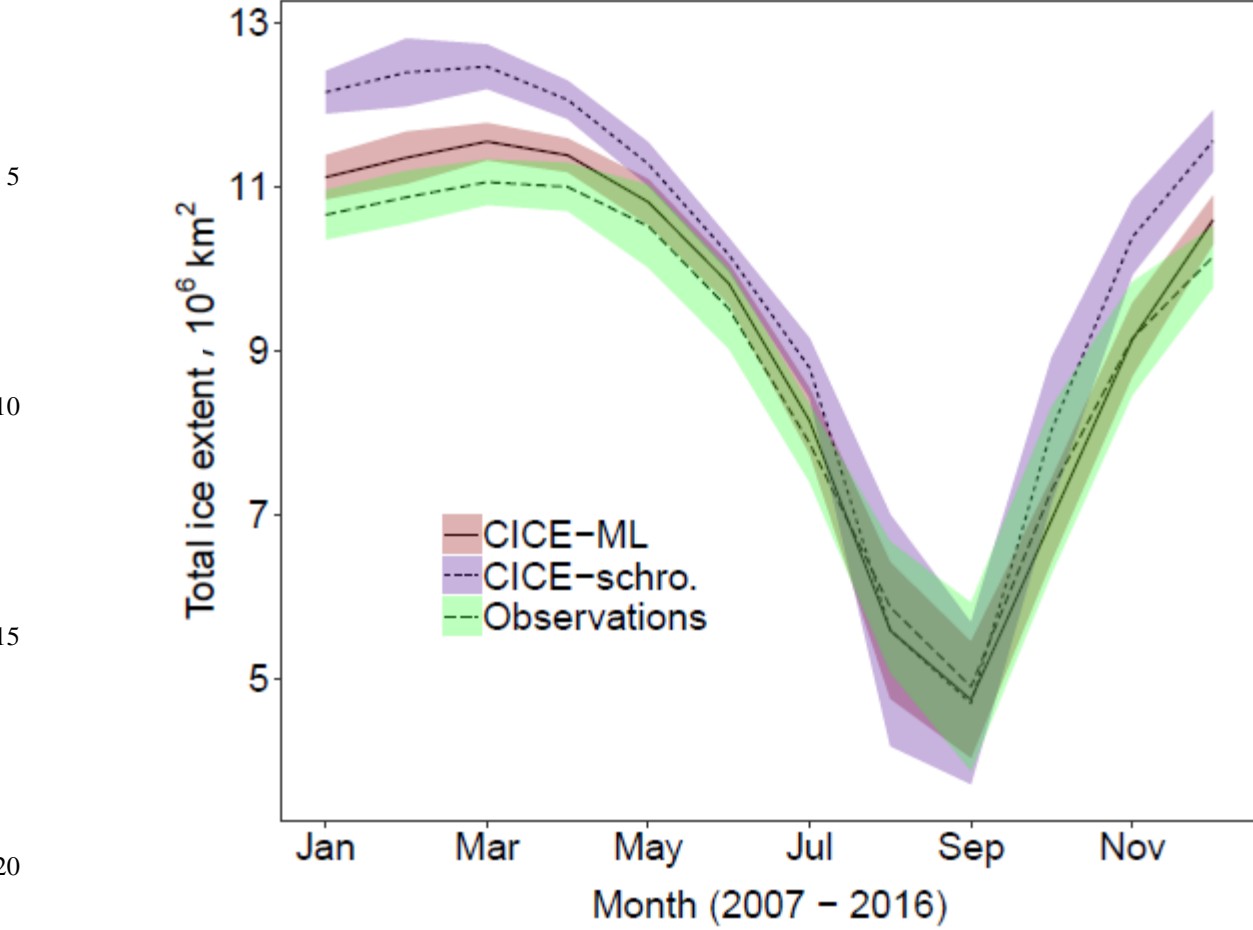

**Figure 2:** Comparison of the 2007 – 2016 mean cycle for the total Arctic sea ice extent simulated in the coupled CICE-prognostic mixed layer reference setup (marked CICE-ML, red ribbon, solid) with the results from the standard optimised CPOM CICE model (Schröder et al., marked CICE-schro., 2018, blue ribbon, small dashes) and observed sea ice extent derived from Nimbus-7 SMMR and DMSP SSM/I-SSMIS satellites using Bootstrap algorithm version 3 (Comiso, 2017, marked Observations, green ribbon, large dashes). The ribbon shows, in each case, the region spanned by the mean value plus or minus two times the standard deviation for each simulation. This gives a measure of the interannual variability over the 10-year period. Results show the new model performs either comparably to or better than the previous optimum setup throughout the year. In addition, the mean CICE-ML sea ice extent falls within the interannual variability of the observations between June and December i.e. most of the melting season, suggesting this reference state is suitable for studies focusing on this period.

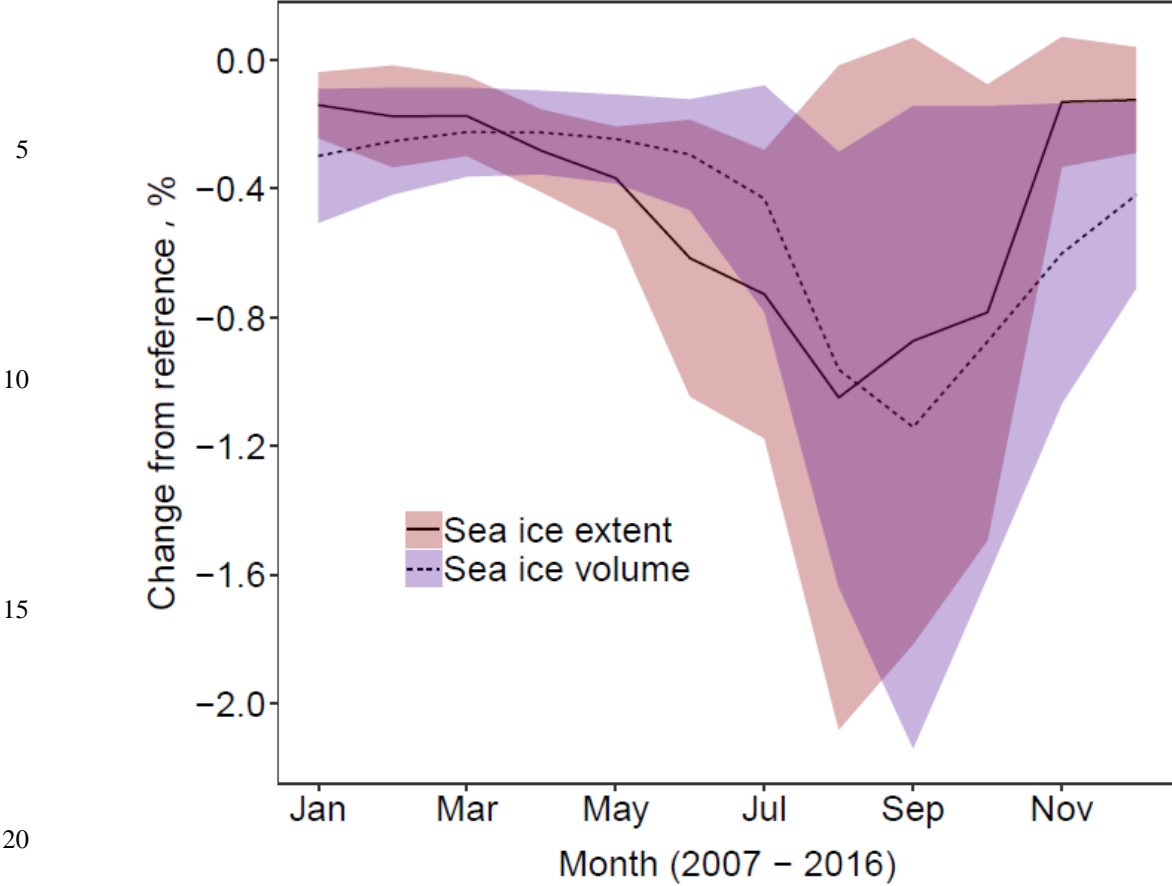

**Figure 3:** Difference in sea ice extent (solid, red ribbon) and volume (dashed, blue ribbon) between *stan-fsd* relative to *ref* (using a constant floe size) averaged over 2007 - 2016. The ribbon shows, in each case, the region spanned by the mean value plus or minus two times the standard deviation for each simulation. This gives a measure of the interannual variability over the 10-year period. The mean behaviour is a reduction in the sea ice extent and volume, with losses of up to 1 % and 1.2 % respectively seen in September during the period of minimum sea ice. The interannual variability shows that the impact of the WIPoFSD model with standard parameters varies significantly between years, with some years potentially showing negligible change in extent and volume and others showing a maximum reduction of over 2 %.

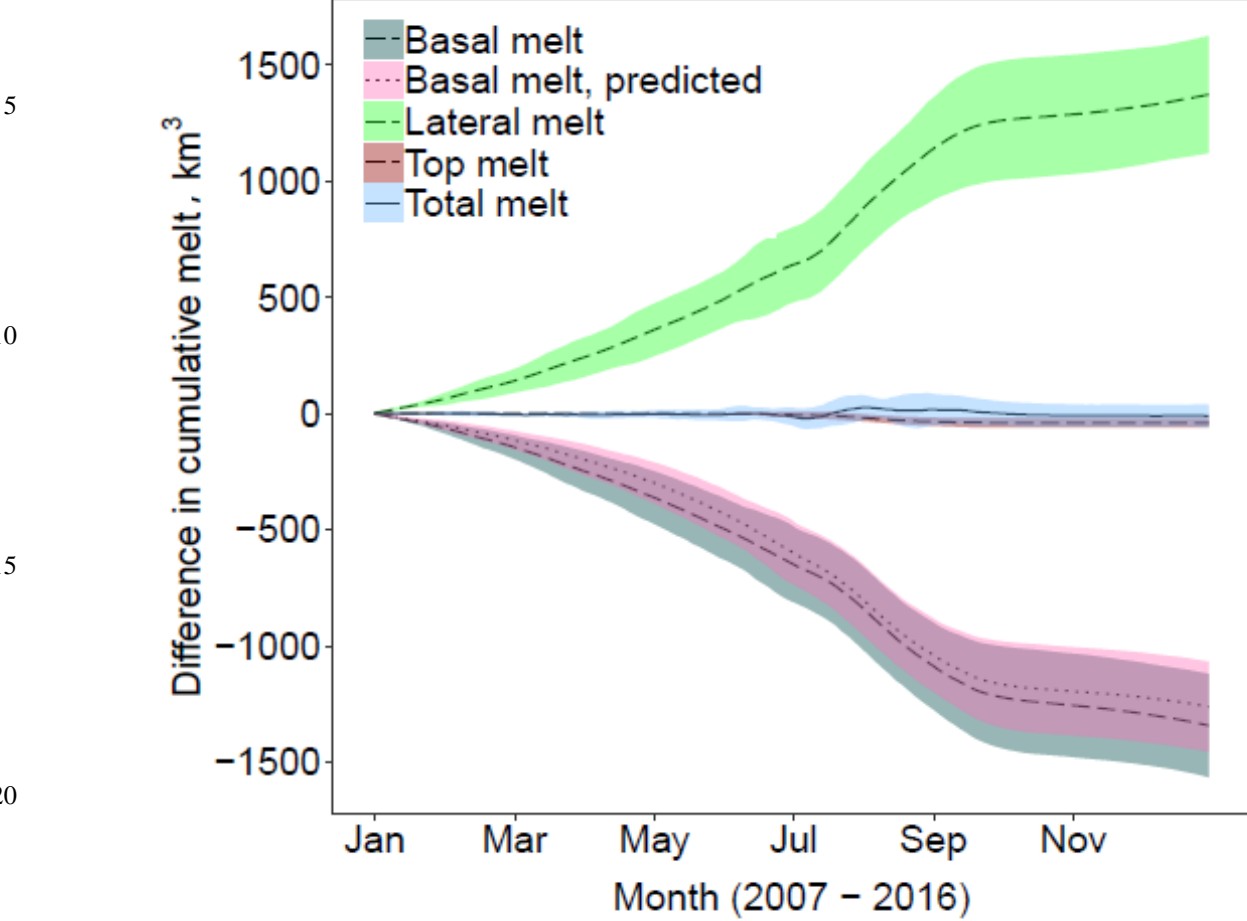

**Figure 4:** Difference in the cumulative lateral (green ribbon, dashed), basal (grey ribbon, dashed), top (red ribbon, dashed) and total (blue ribbon, solid) melts averaged over 2007 - 2016 between *stan-fsd* relative to *ref*. The ribbon shows, in each case, the region spanned by the mean value plus or minus two times the standard deviation for each simulation. A large increase is observed in the total lateral melt, however this is mostly compensated by a reduction in the basal melt, leading to a negligible change in total melt. A small reduction in top melt can be seen. The predicted difference in basal melt is also shown on the plot (pink ribbon, dotted); this shows the expected change in basal melt accounting only for the reduction in sea ice concentration at grid cell scale from *ref* to *stan-fsd*.

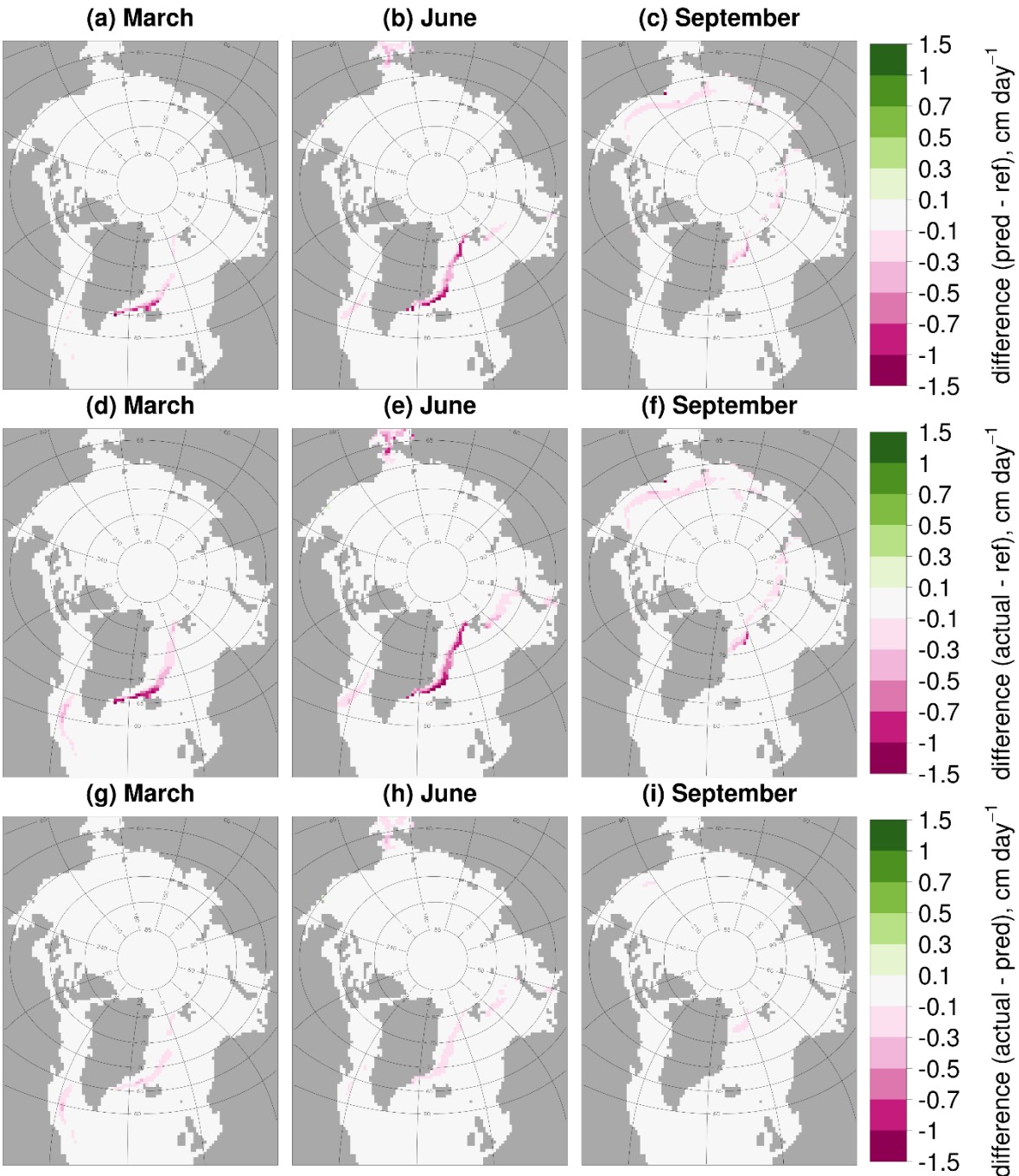

**Figure 5:** Predicted reduction in basal melt rate from *stan-fsd* to *ref* (top row, a-c), actual reduction in basal melt rate from *stan-fsd* to *ref* (middle row, d-f), and difference between the actual reduction and predicted reduction in basal melt rate (bottom row, g-i) averaged over 2007 – 2016. Results are presented for March (left column, a, d, g), June (middle column, b, e, h) and September (right column, c, f, i). Values are shown only in locations where the sea ice concentration exceeds 5 %. The predicted reduction in basal melt rate refers to the expected reduction if the change in sea ice area fraction is the only factor driving the change in basal melt rate. This is calculated by multiplying the basal melt rate for *ref* by the relative percent change in ice area fraction from *ref* to *stan-fsd* for each grid cell.

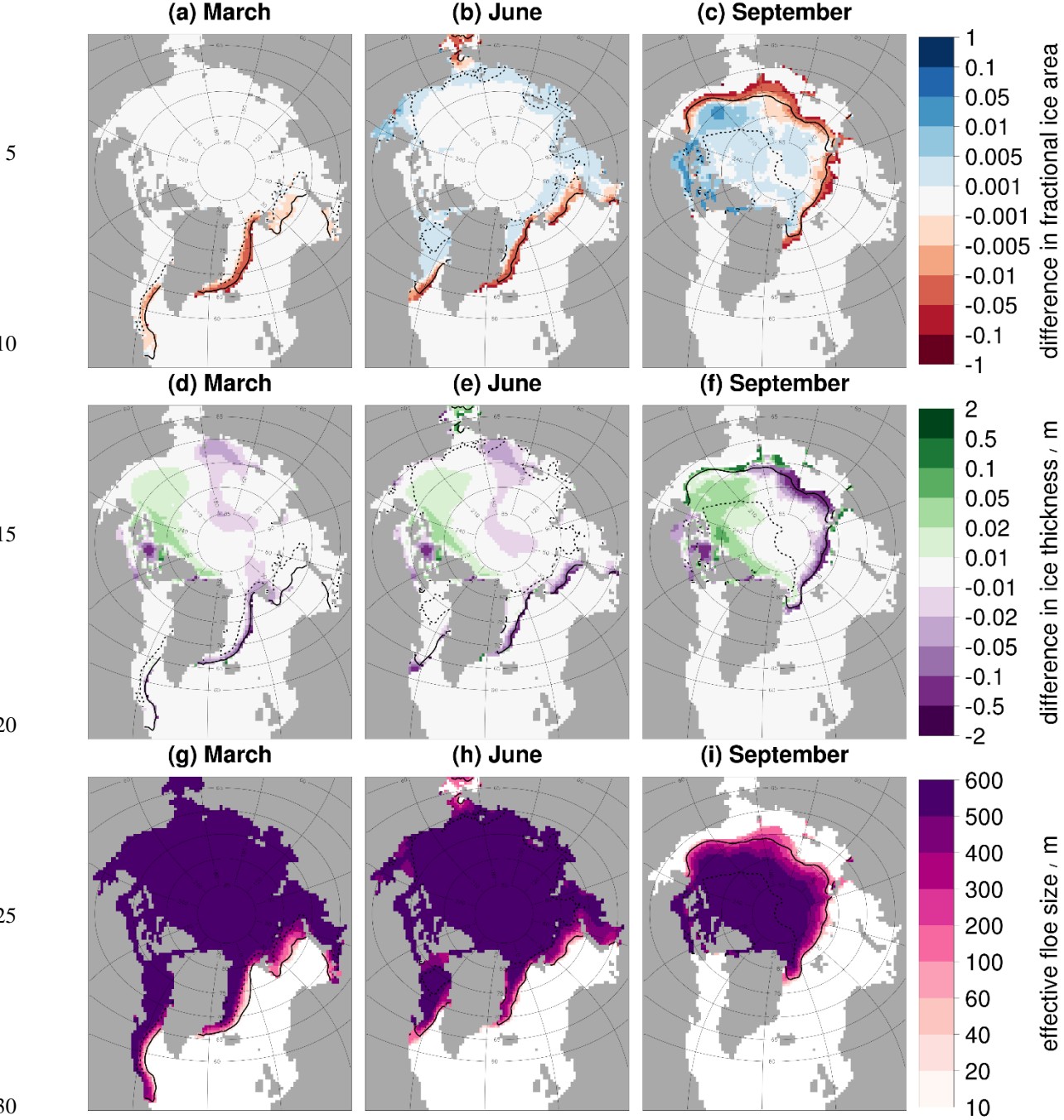

**Figure 6:** Difference in the sea ice concentration (top row, a-c) and ice thickness (middle row, d-f) between *stan-fsd* and *ref* and $l_{eff}$ (bottom row, g-i) for *stan-fsd* averaged over 2007 – 2016. Results are presented for March (left column, a, d, g), June (middle column, b, e, h) and September (right column, c, f, i). Values are shown only in locations where the sea ice concentration exceeds 5 %. The inner (dashed black) and outer (solid black) extent of the MIZ averaged over the same period is also shown. In general, the plots show an increase in the sea ice concentration and thickness in the pack ice, but a reduction in the MIZ. This corresponds to the behaviour of the $l_{eff}$, with increases in regions where the $l_{eff}$ is above 300 m and reductions where it is below 300 m.

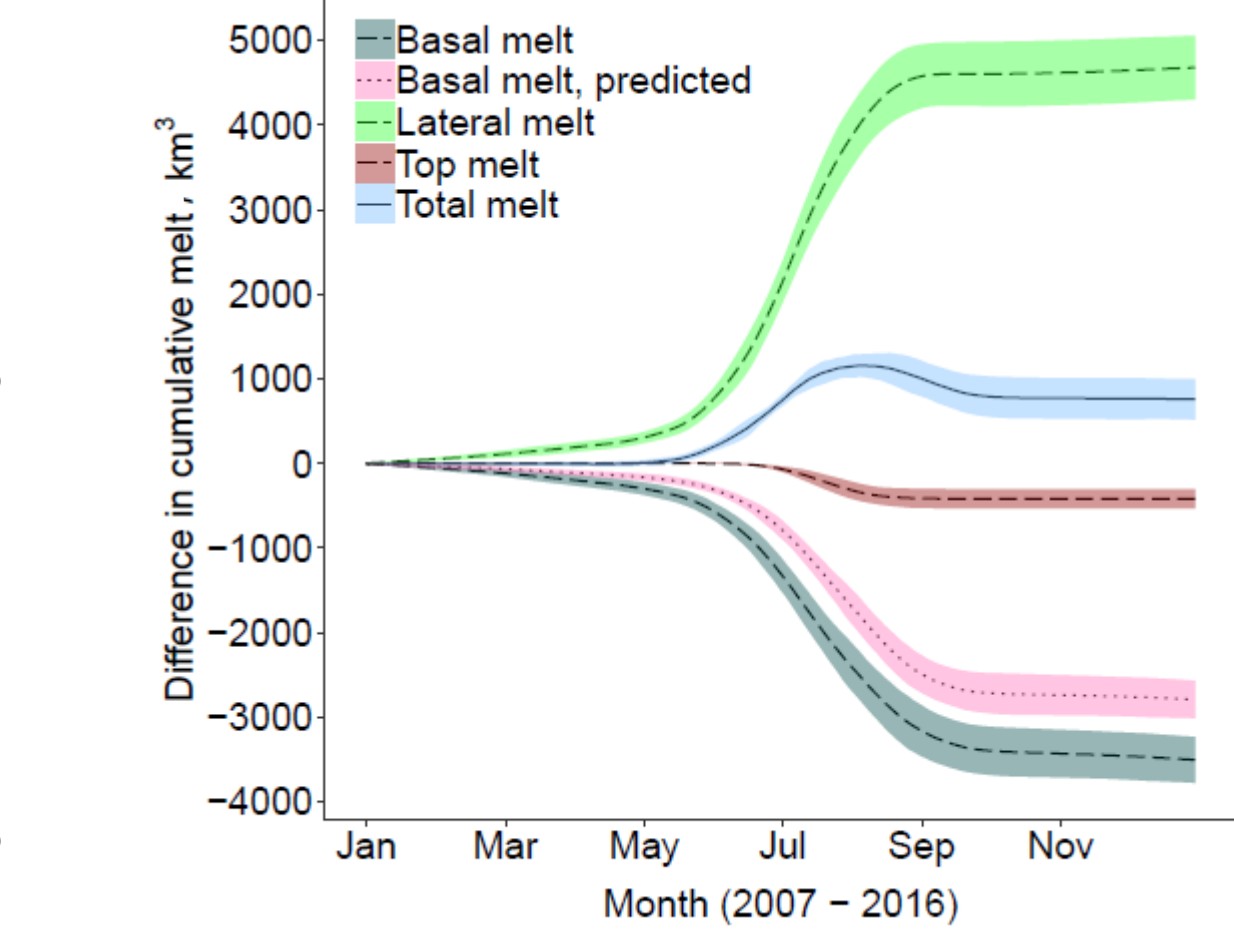

**Figure 7:** As Fig. 4 but the difference between (A) compared to *stan-fsd* i.e. the impact of changing $\alpha$ from 2.5 to 3.5 with the other FSD parameters held at standard values. A large increase in lateral melt is partly compensated by a reduction in basal melt, however this time a large increase is seen in the total melt.

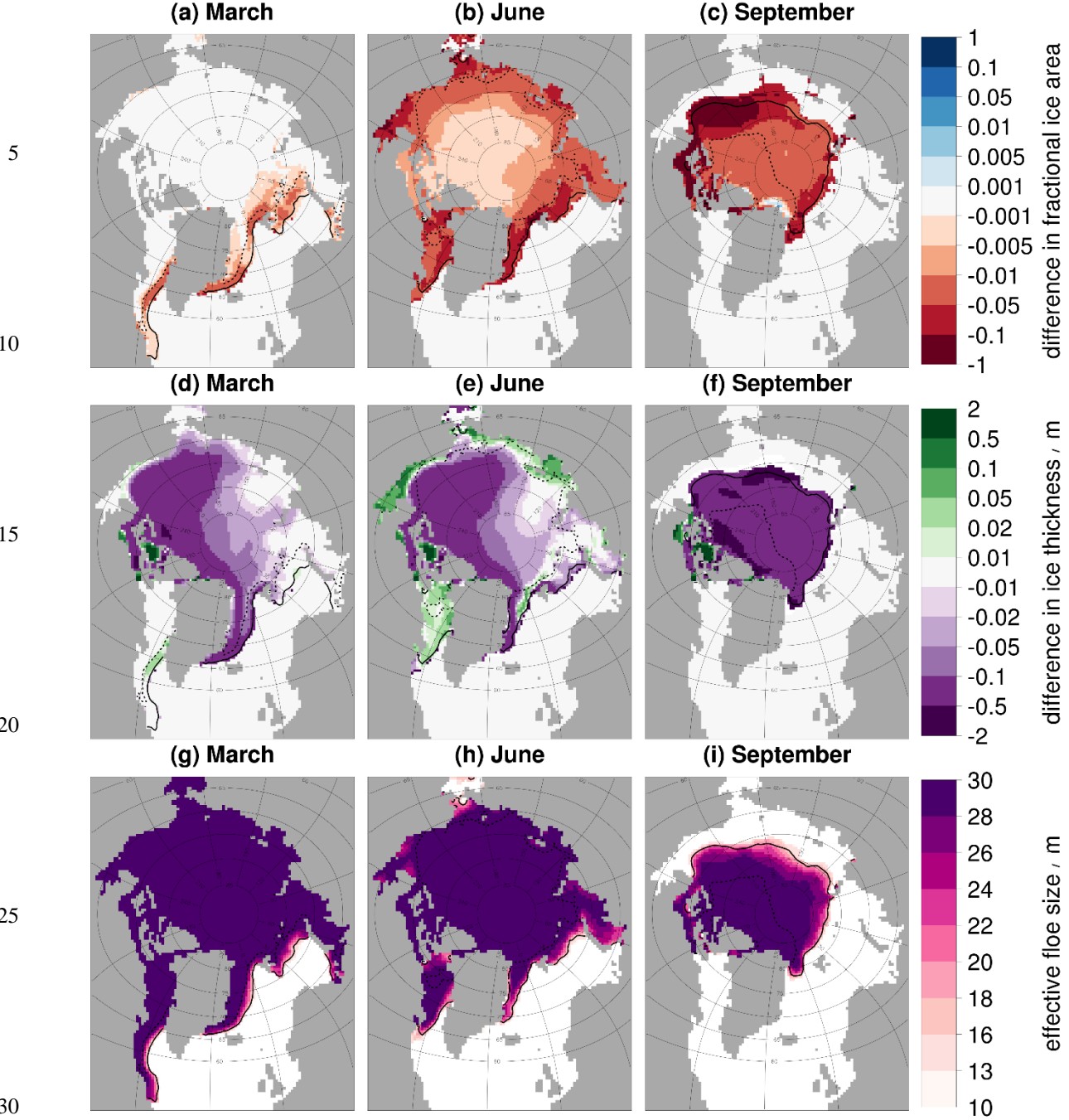

**Figure 8:** As Fig. 6 except now the difference between (A) compared to *stan-fsd* is given i.e. the impact of changing $\alpha$ from 2.5 to 3.5 with the other FSD parameters held at standard values. $l_{eff}$ is reported for the simulation with the higher magnitude $\alpha$. In general, the plots show a reduction in the sea ice concentration and ice thickness across the sea ice cover. This corresponds to the behaviour of the $l_{eff}$, with the $l_{eff}$ 30 m or below across the sea ice cover.

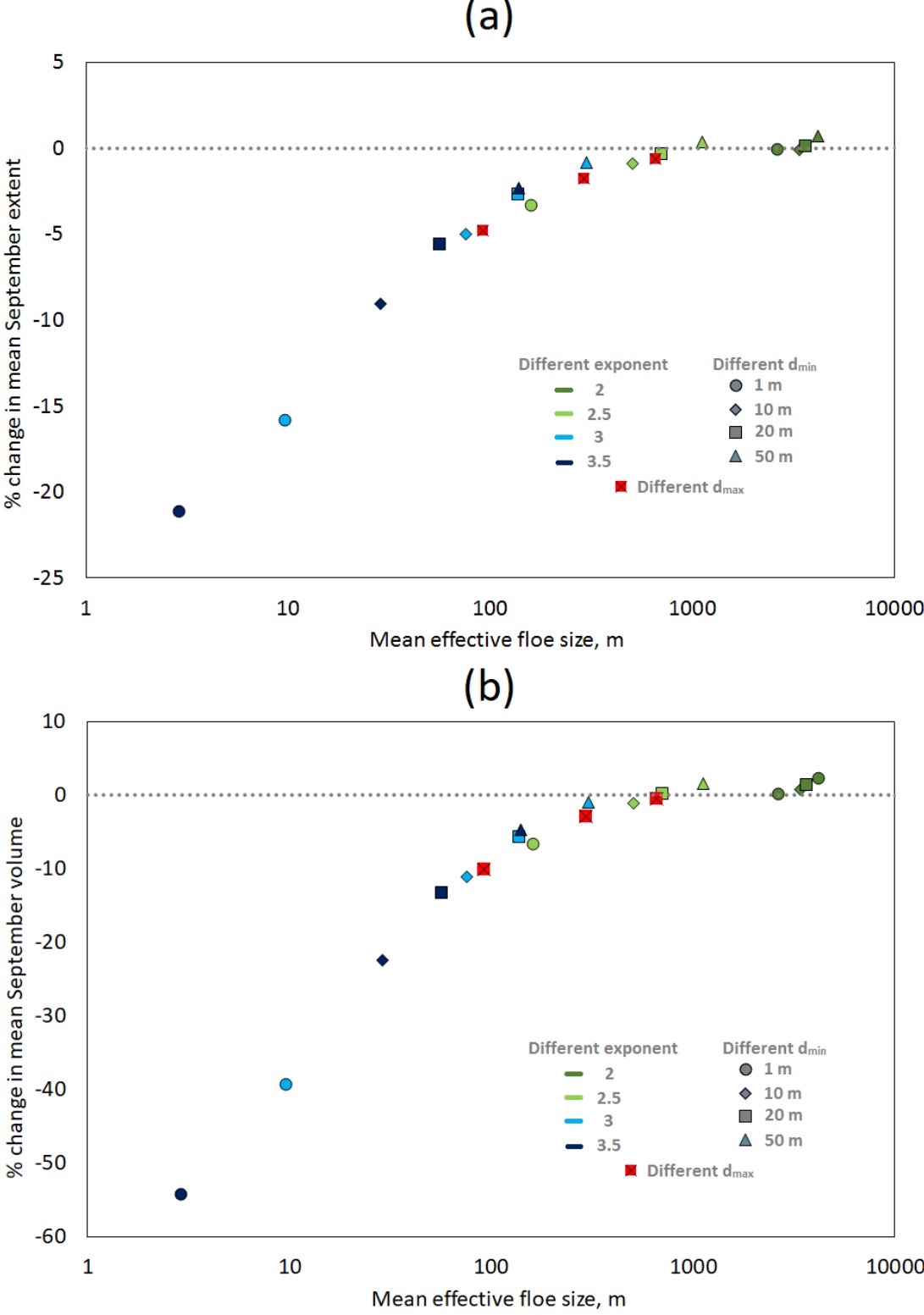

**Figure 9:** Relative change (%) in mean September sea ice extent (a) and volume (b) from 2007 - 2016 respectively, plotted against mean $l_{eff}$ for simulations with different selections of parameters relative to *ref*. The mean $l_{eff}$ is taken as the equally weighted average across all grid cells where the sea ice concentration exceeds 15%. The colour of the marker indicates the value of the $\alpha$, the shape indicates the value of $d_{min}$, and the three experiments using standard parameters but different $d_{max}$ (1000 m, 10000 m and 50000 m) are indicated by a crossed red square. The parameters are selected to be representative of a parameter space for the WIPoFSD that has been constrained by observations. Model response ranges from small increases in the sea ice extent and volume to reductions of over 20 and 50 % respectively. The mean $l_{eff}$ is shown to be a good predictor of the response of the sea ice extent and volume.

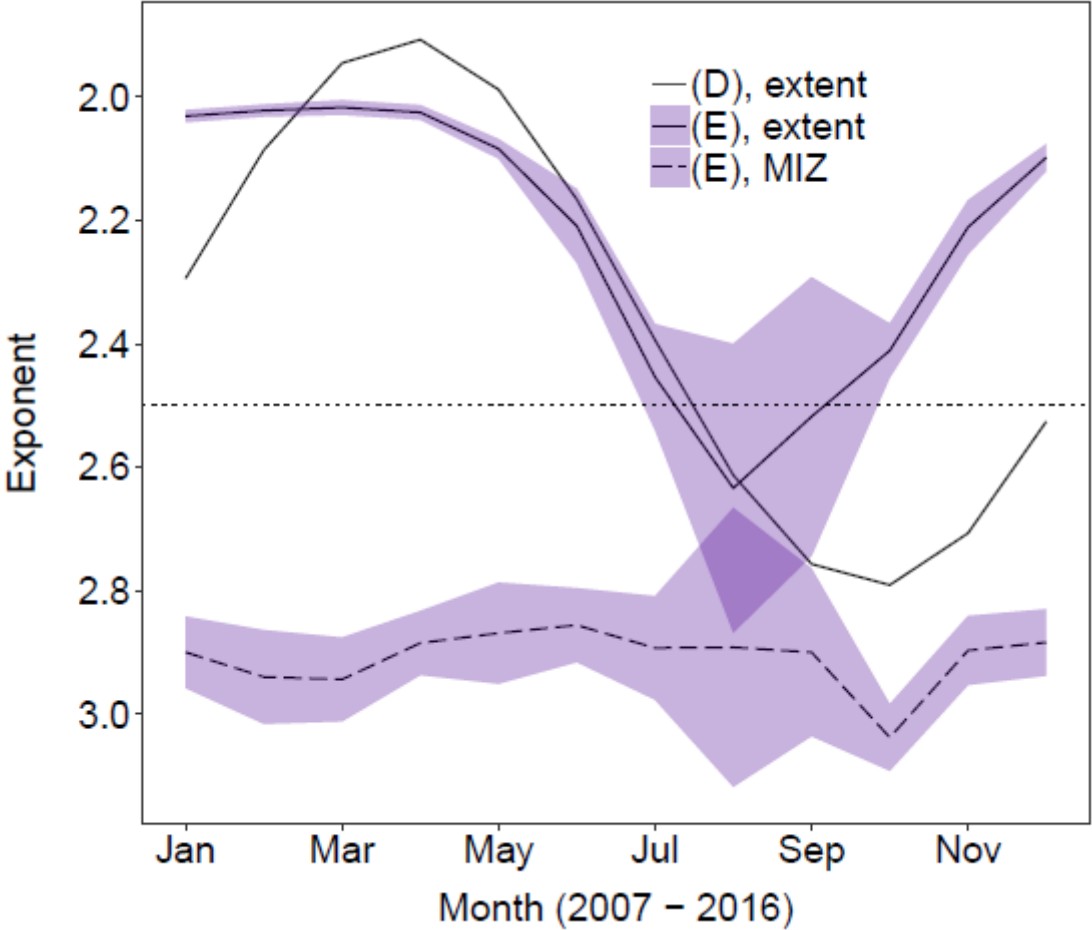

**Figure 10:** Annual variation in $\alpha$ (top) averaged over $2007 - 2016$ for two simulations with variable $\alpha$. The plots show results for an $\alpha$ which varies depending on time through the year (D, no ribbon) or on the sea ice concentration (E, blue ribbon). Results are given as the mean $\alpha$ for the total sea ice extent (solid) and MIZ only (dashed). The mean $\alpha$ is taken as the equally weighted average across all grid cells where the sea ice concentration exceeds 5% (total extent) or is between 15% - 80% (MIZ only). The imposed annual oscillation in $\alpha$ is identical for all grid cells for (D), hence the MIZ behaviour has not been plotted as it will be identical to the annual oscillation in $\alpha$ across the total sea ice extent. The ribbon shows, in each case, the region spanned by the mean value plus or minus two times the standard deviation for each simulation. Both setups show an annual oscillation in the value of $\alpha$ averaged over the total sea ice extent. For experiment (E), no obvious annual trend in the mean value of $\alpha$ can be seen when averaged over the MIZ, though the interannual variation is at a maximum during the peak melting season between July and September.

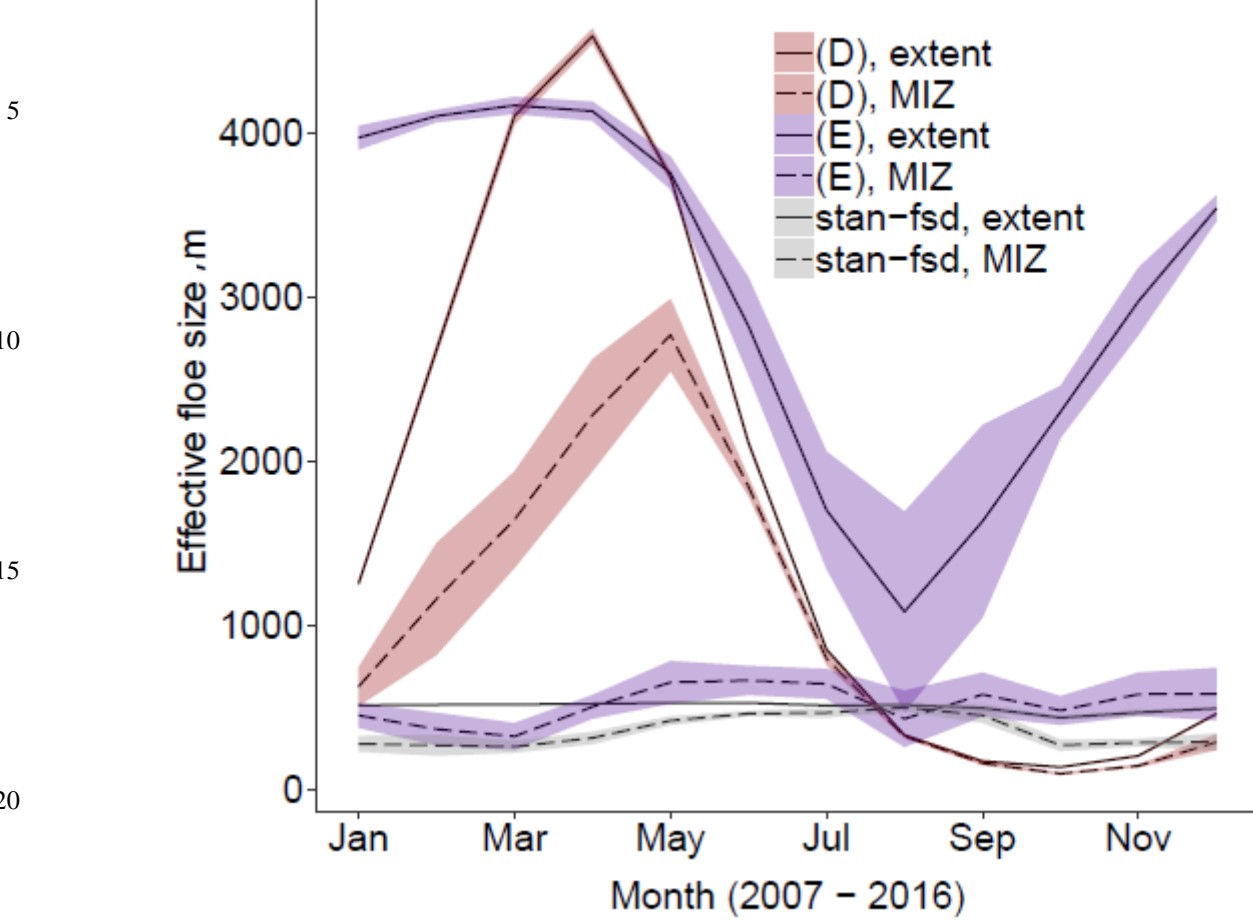

**Figure 11:** Annual variation in mean $l_{eff}$ averaged over 2007 – 2016 for two simulations with variable $\alpha$. The plots show the evolution of $l_{eff}$ throughout the year for a simulation with a time-dependent $\alpha$ (D, red ribbon) or a sea ice concentration-dependent $\alpha$ (E, blue ribbon). Also shown is the behaviour of $l_{eff}$ for a simulation with a fixed $\alpha$ of 2.5 (*fsd-stan*, grey ribbon). Results are shown for the total sea ice area (solid) and MIZ only (dashed). The mean $l_{eff}$ is taken as the equally weighted average across all grid cells where the sea ice concentration exceeds 5% (total extent) or is between 15% - 80% (MIZ only). The ribbon shows, in each case, the region spanned by the mean value plus or minus two times the standard deviation for each simulation. The results show that introducing a variable $\alpha$ produces much larger intra-annual variations in $l_{eff}$ across the overall sea ice extent than with a fixed $\alpha$. (D) and (E) show an annual oscillation in the value of $l_{eff}$ averaged over the total sea ice extent. Within the MIZ, only experiment (D) continues to show this strong variation in $l_{eff}$; (E) and *fsd-stan* show variations of around an order less. (D) shows the strongest interannual variation between March and May, whereas for (E) it is strongest in the peak melting season between July and August.

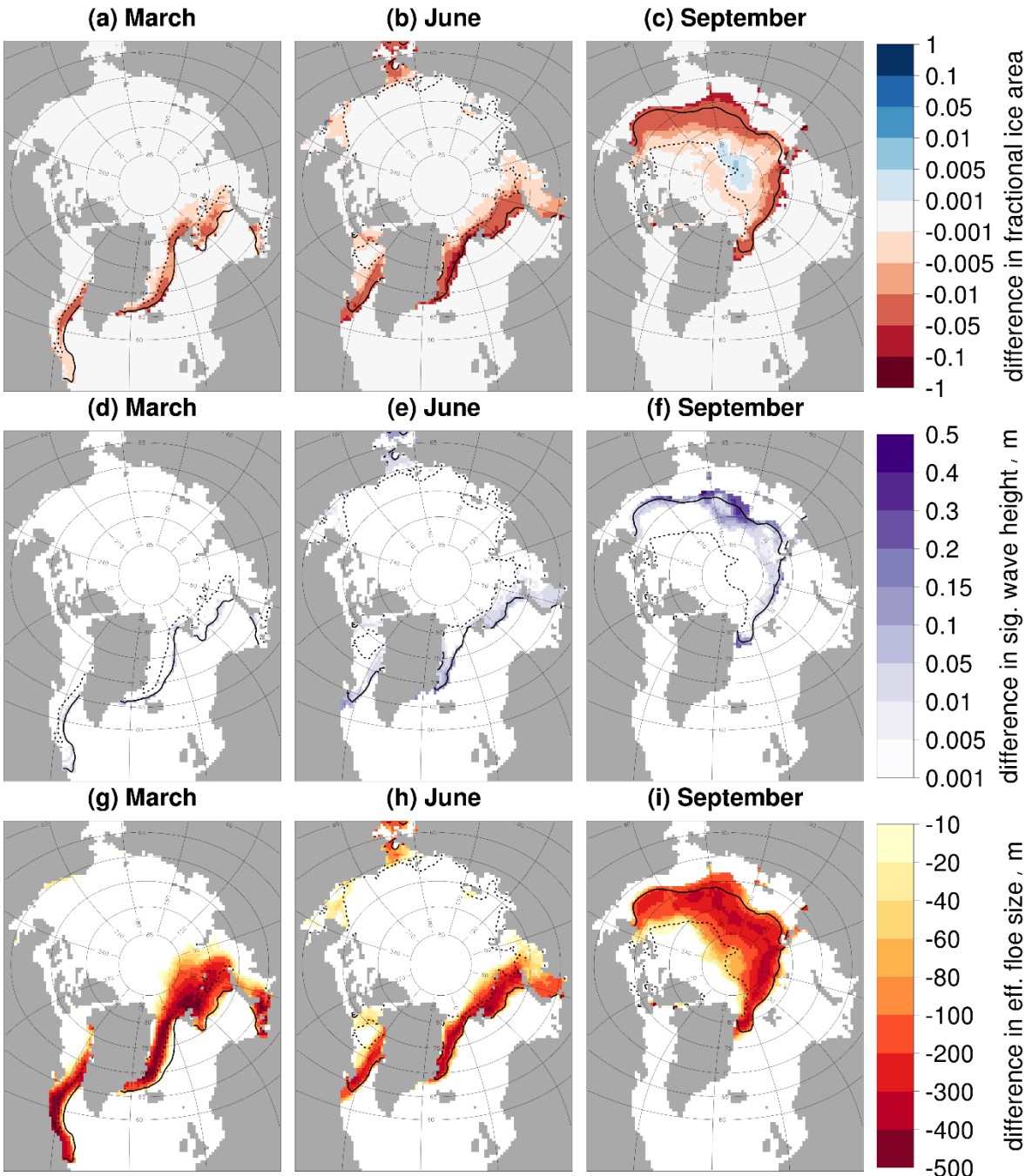

**Figure 12:** Difference in the sea ice concentration (top row, a - c), significant wave height (middle row, d - f) and $l_{eff}$ (bottom row, g - i) for (J), with the wave attenuation rate reduced by 90 %, compared to *stan-fsd*, both using standard FSD parameters. Plots show results for March (left column, a, d, g), June (middle column, b, e, h) and September (right column, c, f, i) averaged over 2007 - 2016. Each plot shows the inner (dashed black) and outer (solid black) extent of the MIZ averaged over the same period. Values are shown only in locations where the sea ice concentration exceeds 5 %. The plots show that despite very small differences in the significant wave height, the reduced attenuation rate still drives reductions in $l_{eff}$ and in consequence the sea ice concentration across the MIZ.

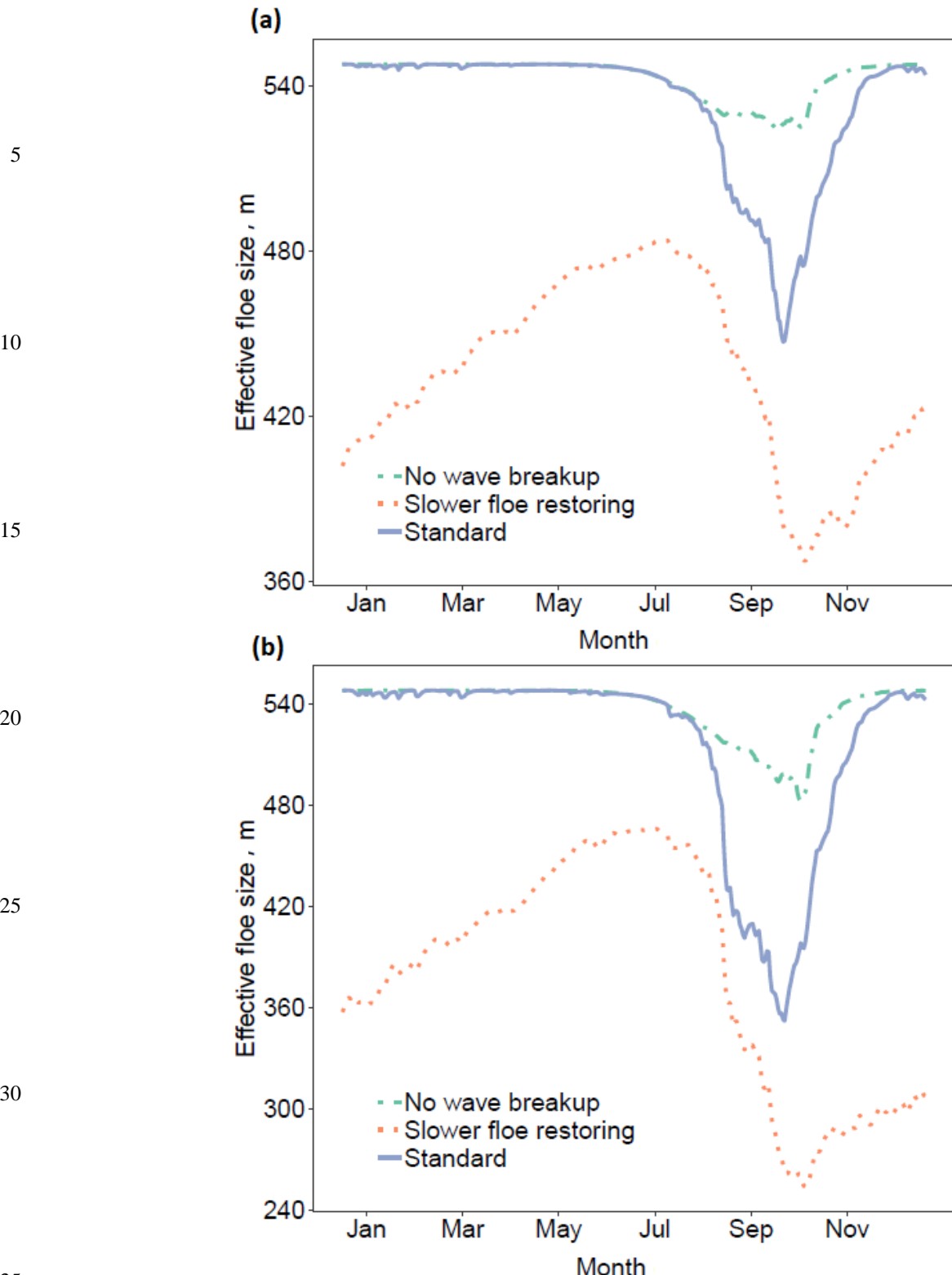

**Figure 13:** Daily variation in $l_{eff}$ over 2015 averaged over (panel a) regions with between $15 - 80$ % sea ice concentration on $31^{st}$ August 2015 and (panel b) regions with between $15 - 30$ % sea ice concentration of $31^{st}$ August 2015. The three simulations demonstrate $l_{eff}$ tendencies with respect to different processes. The plots show the evolution of $l_{eff}$ throughout the year for the standard simulation (*fsd-stan*, blue solid), without wave break-up of floes (F, green dot-dashed), and with a reduced floe size restoring rate in freezing conditions (K, orange dotted). Means for $l_{eff}$ and ice perimeter are taken as averages over the selected grid cells with each grid cell equally weighted. The plots show that a strong seasonal cycle in $l_{eff}$ can be observed, particularly in grid cells on the edge of the sea ice cover where waves are expected to have a particularly strong impact.

| Variable | Description |
|---|---|
| $d_{min}$ | Lower floe size cut-off within the WIPoFSD model. Standard value of 10 m. |
| $d_{max}$ | Upper floe size cut-off within the WIPoFSD model. Standard value of 30000 m. |
| $l_{var}$ | Variable FSD tracer. Allowed to vary between $d_{min}$ and $d_{max}$. |
| $\alpha$ | Power law exponent within the WIPoFSD model. Standard value fixed at 2.5. |
| $l_{eff}$ | The effective floe size is defined as the floe size of a distribution of identical floes that would produce the same lateral melt rate in a given instant to a distribution of non-uniform floes, when under the same conditions with the same total ice cover. See Eq. (15). |
| $\alpha_{dim}$ | The dimensional attenuation coefficient, as used in Eq. (6). |
| $P_{crit}$ | The critical probability that must be exceeded for wave breaking events to occur, as used in Eq. (7). |
| $T_{rel}$ | The floe restoring rate, as used in Eq. (19). Set to 10 as default. |
| $\alpha_{shape}$ | Floe shape parameter to account for the deviation of floes from a perfect circle. Standard value of 0.66 (Rothrock and Thorndike, 1984). |
| $w_{lat}$ | Lateral melt rate, as calculated within Eq. (2). |
| $m_1$ | Melt rate parameter, as used in Eq. (2) to calculate the lateral melt rate $w_{lat}$. Default value of 1.6 x 10$^{-6}$ $m\ s^{-1}K^{-m_2}$ (Perovich, 1983). |
| $m_2$ | Melt rate parameter, as used in Eq. (2) to calculate the lateral melt rate $w_{lat}$. Default value of 1.36 (Perovich, 1983). |

**Table 1:** Definitions of the parameters relating to the sensitivity studies described in table 2.

| Sensitivity study | Description | Technical details |
|---|---|---|
| stan-fsd | CICE-ML with standard FSD | $d_{min} = 10\ m$, $d_{max} = 30,000\ m$, $\alpha = 2.5$ |
| Ref | CICE-ML with constant floe size | Floe size of 300 m for all floes |
| (A) | Low $\alpha$ | $d_{min} = 10\ m$, $d_{max} = 30,000\ m$, $\alpha = 3.5$ |
| (B) | Minimum $l_{eff}$ | $d_{min} = 1\ m$, $d_{max} = 30,000\ m$, $\alpha = 3.5$<br>This is the selection of FSD parameters that produces the lowest average $l_{eff}$ |
| (C) | Maximum $l_{eff}$ | $d_{min} = 50\ m$, $d_{max} = 30,000\ m$, $\alpha = 2$<br>This is the selection of FSD parameters that produces the highest average $l_{eff}$ |
| (D) | $\alpha$ evolves over a fixed annual cycle | An annual cycle, as described by Eq. (20), is imposed on the exponent based on the observations of Stern et al. (2018 a). The exponent does not vary spatially. |
| (E) | $\alpha$ as a function of local ice concentration | The exponent becomes a function of the local sea ice concentration (i.e. fractional sea ice area) according to Eq. (21). |
| (F) | Waves no longer break-up floes | The waves-in-ice module operates normally, however Eq. (18) is no longer applied after a floe break-up event is identified. |
| (G) | No lateral melt feedback on floe size | The model operates normally, however $l_{var}$ is no longer reduced based on the amount of lateral melt i.e. Eq. (17) is removed from the model. |
| (H) | Big waves | The significant wave heights read into the model from ERA-interim data at ice free locations is increased by a factor of 10. |
| (I) | Weak ice | $P_{crit}$ is reduced by a factor of 10. |
| (J) | Weaker wave attenuation | $\alpha_{dim}$ is reduced by a factor of 10. |
| (K) | Reduced floe growth rates | $T_{rel}$ is increased from 10 to 365. |
| (L) | Less circular floes | $\alpha_{shape}$ is reduced from 0.66 to 0.44. |
| (M) | Perfectly circular floes | $\alpha_{shape}$ is increased from 0.66 to 0.79. This is the approximate value of this parameter for a perfect circle. |
| (N) | Reduced lateral melt rate | The parameters $m_1$ and $m_2$ are reduced by 10 % each to 1.44 x $10^{-6}\ m\ s^{-1}K^{-m_2}$ and 1.22 respectively. |
| (O) | Increased lateral melt rate | The parameters $m_1$ and $m_2$ are increased by 10 % each to 1.76 x $10^{-6}\ m\ s^{-1}K^{-m_2}$ and 1.48 respectively. |
| (P) | Shallow mixed layer | The minimum mixed layer depth is reduced from 10 m to 7 m. |
| (Q) | Deep mixed layer | The minimum mixed layer depth is increased from 10 m to 20 m. |

**Table 2:** The details of the sensitivity studies to explore the behaviour of the CICE-ML-WIPoFSD model. Parameters discussed here defined in table 1.

| Study | Description | Area metrics, $10^6$ km² | | Volume, $10^3$ km³ | | Mean MIZ $l_{eff}$, m | Mean MIZ ice perimeter, m⁻¹ | Annual cumulative melt by end of September, $10^3$ km³ | | | |
|---|---|---|---|---|---|---|---|---|---|---|---|
| | | Extent | MIZ | Total | MIZ | | | Top | Basal | Lateral | Total |
| stan-fsd | CICE-ML with standard FSD | 4.70 (0) | 2.54 (0) | 7.72 (0) | 2.07 (0) | 453.9 | 0.0070 | 5.21 (0) | 14.58 (0) | 2.43 (0) | 22.22 (0) |
| ref | CICE-ML with constant floe size | 4.74 (0.04) | 2.61 (0.06) | 7.81 (0.09) | 2.12 (0.05) | 300 | 0.0081 | 5.25 (0.04) | 15.79 (1.22) | 1.17 (-1.26) | 22.21 (-0.01) |
| (A) | Low $\alpha$ | 4.31 (-0.39) | 2.55 (0.01) | 6.06 (-1.67) | 1.75 (-0.32) | 27.7 | 0.0862 | 4.79 (-0.42) | 11.19 (3.39) | 7.03 (-4.60) | 23.01 (0.79) |
| (B) | Minimum $l_{eff}$ | 3.76 (-0.96) | 2.75 (0.21) | 3.56 (-4.16) | 1.18 (-0.90) | 2.7 | 0.8151 | 3.60 (-1.60) | 4.34 (-10.23) | 16.56 (14.13) | 24.50 (2.28) |
| (C) | Maximum $l_{eff}$ | 4.77 (0.07) | 2.58 (0.04) | 7.98 (0.26) | 2.14 (0.07) | 3656.3 | 0.0023 | 5.27 (0.06) | 15.54 (0.96) | 1.30 (-1.13) | 22.11 (-0.11) |
| (D) | $\alpha$ evolves over fixed annual cycle | 4.69 (-0.01) | 2.53 (-0.01) | 7.70 (-0.02) | 2.06 (-0.01) | 162.9 | 0.0161 | 5.23 (0.02) | 14.68 (0.11) | 2.31 (-0.12) | 22.22 (0.00) |
| (E) | $\alpha$ is a function of ice concentration | 4.55 (-0.15) | 2.39 (-0.15) | 7.34 (-0.38) | 1.85 (-0.22) | 580.6 | 0.0184 | 5.13 (-0.07) | 13.46 (-1.11) | 3.75 (1.32) | 22.34 (0.12) |
| (F) | Waves no longer break-up floes | 4.78 (0.08) | 2.62 (0.08) | 7.93 (0.21) | 2.15 (0.08) | 531.8 | 0.0045 | 5.27 (0.06) | 15.95 (1.37) | 0.93 (-1.50) | 22.15 (-0.07) |
| (G) | No lateral melt feedback on floe size | 4.70 (0.01) | 2.55 (0.01) | 7.75 (0.03) | 2.08 (0.01) | 465.3 | 0.0068 | 5.21 (0.01) | 14.69 (0.12) | 2.30 (-0.13) | 22.20 (-0.02) |
| (H) | Big waves | 4.60 (-0.10) | 2.44 (-0.10) | 7.47 (-0.26) | 1.94 (-0.14) | 299.8 | 0.0212 | 5.16 (-0.05) | 13.62 (-0.96) | 3.53 (1.10) | 22.31 (0.09) |
| (I) | Weak ice | 4.66 (-0.04) | 2.51 (-0.04) | 7.65 (-0.08) | 2.03 (-0.04) | 412.4 | 0.0095 | 5.19 (-0.02) | 14.26 (-0.32) | 2.79 (0.36) | 22.24 (0.02) |
| (J) | Weaker wave attenuation | 4.57 (-0.17) | 2.42 (-0.12) | 7.40 (-0.33) | 1.90 (-0.18) | 236.6 | 0.0328 | 5.15 (-0.05) | 13.42 (-1.16) | 3.76 (1.34) | 22.33 (0.11) |
| (K) | Reduced floe growth rates | 4.68 (-0.02) | 2.54 (0.00) | 7.67 (-0.05) | 2.06 (-0.01) | 372.9 | 0.0100 | 5.18 (-0.03) | 14.40 (-0.18) | 2.67 (0.24) | 22.25 (0.03) |
| (L) | Less circular floes | 4.64 (-0.05) | 2.50 (-0.04) | 7.56 (-0.16) | 2.01 (-0.06) | 442.2 | 0.0109 | 5.17 (-0.03) | 14.04 (-0.53) | 3.05 (0.63) | 22.26 (0.04) |
| (M) | Perfectly circular floes | 4.72 (0.02) | 2.56 (0.02) | 7.79 (0.07) | 2.11 (0.03) | 459.1 | 0.0058 | 5.22 (0.02) | 14.92 (0.34) | 2.05 (-0.38) | 22.19 (-0.03) |
| (N) | Reduced lateral melt rate | 4.71 (0.01) | 2.56 (0.01) | 7.74 (0.02) | 2.08 (0.01) | 456.3 | 0.0069 | 5.22 (0.01) | 14.77 (0.19) | 2.23 (-0.20) | 22.21 (-0.01) |
| (O) | Increased lateral melt rate | 4.69 (-0.01) | 2.53 (-0.01) | 7.70 (-0.02) | 2.06 (-0.01) | 451.5 | 0.0071 | 5.20 (-0.01) | 14.41 (-0.17) | 2.61 (0.18) | 22.22 (0.00) |
| (P) | Shallow mixed layer | 4.70 (0.01) | 2.57 (0.03) | 7.84 (0.12) | 2.14 (0.07) | 447.4 | 0.0067 | 5.19 (-0.02) | 14.56 (-0.01) | 2.46 (0.03) | 22.21 (-0.01) |
| (Q) | Deep mixed layer | 4.64 (-0.06) | 2.65 (0.10) | 7.62 (-0.11) | 2.34 (0.27) | 473.6 | 0.0071 | 5.26 (0.05) | 14.57 (-0.01) | 2.34 (-0.08) | 22.17 (-0.05) |

**Table 3:** A summary of the metrics for each of the sensitivity studies described in table 2. Metrics are reported for sea ice extent, MIZ extent, total sea ice volume, MIZ volume, mean $l_{eff}$ within the MIZ, mean sea ice perimeter per m² of ocean area within the MIZ, and cumulative melt top, basal, lateral and total melt. All metrics are reported for September, except the cumulative melt which is reported for all months up to and including September and given as an average between 2007 – 2016. Means for $l_{eff}$ and ice perimeter are taken as averages over the MIZ with each grid cell equally weighted. The values in red within the parentheses give the change from *stan-fsd*. Cells highlighted in yellow and orange deviate by one and two standard deviation(s) respectively from the *stan-fsd* mean value (the standard deviation is calculated from the set of 10 annual values for each metric).