# Peer review of "Impact of sea ice floe size distribution on seasonal fragmentation and melt of Arctic sea ice"

_The Cryosphere, 2019_

## Referee Comment (RC1) · Anonymous Referee #1 · 18 Apr 2019

In their manuscript „Impact of floe size distribution on seasonal fragmentation and melt of Arctic sea ice", Adam Bateson and colleagues present results of a numerical sensitivity study concentrating on the role of the floe size distribution (FSD) in shaping the seasonal cycle of the sea ice cover in the Arctic. The study is based on a numerical sea ice model coupled with a mixed layer ocean model. Several processes influencing/influenced by the FSD are investigated, including the lateral melting of sea ice and ice fragmentation induced by waves. The FSDs considered are truncated power laws described by three parameters: the minimum and maximum floe size, and the exponent.

The manuscript is devoted to a subject which is very important for the performance of numerical sea ice models on both short (synoptic) and long (seasonal, climate) time scales. As the authors point out in the introduction, recent climate change and the associated negative trends in sea ice extent, thickness and strength have produced conditions in which interactions with waves are becoming more and more important over larger and larger areas. Fragmentation of sea ice by waves in turn influences ice-ocean-atmosphere heat flux and sea ice melting/freezing rates. At present, our limited understanding of many aspects of these coupled processes is a serious limitation for development of reliable parameterizations suitable for numerical sea ice models. In my opinion, the proposed manuscript is an important contribution to the subject, even though many solutions used in the model and assumptions underlying them are oversimplified (or maybe even wrong). As the authors correctly remark, these simplifying assumptions are to a large degree a result of the lack of observational data and/or theoretical understanding available. In this respect, the most important contribution of the manuscript is that it develops a framework in which future developments can be integrated, as new data and insights become available. In other words, in spite of some clear limitations of the solutions presented, I find this contribution very valuable, as it paves the way for further development.

Several aspects of the results are very interesting, for example the findings related to the (partial) compensation of the increased lateral-melting rates by decreased bottom-melting rates in simulations with power-law FSD compared to simulations with constant floe size; or the important role of the wave attenuation rates in shaping the seasonal cycle of sea ice. I think that those results are worth publishing in "The Cryosphere". My comments to the manuscript are listed below. My recommendation to the Editors is "major revision".

General comments:

1. The text of the manuscript is written in an untidy manner, it contains a lot of (mostly small) mistakes and should be carefully checked before publication. Moreover, the text is full of technical slang, colloquialisms, informal expressions which (in most cases) are comprehensible for a reader familiar with numerical modelling of ocean/sea ice, but should be avoided in a paper. It makes reading of the text tiresome.

2. I find the description in section 2.2 quite chaotic. The impression is that the choice of topics that are described there is pretty random, even though the first sentence announces that this section contains only "elements pertinent to our study". For example, is the CFL criterion relevant for the results/discussion presented further? In turn, some more details from the papers of Williams and colleagues regarding computation of wave-induced breaking etc. would be very useful, even though they can be found in those papers.
But the most important comment to section 2.2 is related to the description of the wave forcing. It is very imprecise. Also, I think the Authors should better justify the methodology they use. As written in section 3, the only wave information from ERA reanalysis used as input to the sea ice model are $H_s$ and $T_p$ (I think this information should be provided in section 2.2, so that the text there can be better understood). Why don't the Authors use

wave directions from ERA? Or even the whole energy spectra?

The approach described seems very complicated and requires several arbitrary assumptions. First, the assumption about the correspondence between the wave direction and wind direction might be justified, e.g, in the Beaufort Sea, where waves are predominantly locally generated, but the wave climate in the North Atlantic is dominated by swell, so that the direction of waves reaching the ice edge likely has little in common with the local wind. Moreover, swell typically has a much narrower directional distribution than 45° used by the Authors (by the way: what does "total spread" mean? how is the wave energy distributed among the 5 directional bins?). This might seem a minor detail, but narrower directional distribution of wave energy means that the waves can penetrate deeper into the ice cover (with the same attenuation rate $\alpha_{dim}$) – and the results presented in this paper suggest this might have a significant influence on the results.

Second, from the technical perspective, the approach requires advecting wave energy into the ice in a bin-by-bin manner, and waves from several locations can reach any given ice-covered grid cell. Using full ERA energy spectra would be much more elegant, but not much more complicated.

3. It is hard to prove without numerical simulations, but my impression is that some conclusions from this study might be affected by the – very artificial – assumption that the minimum floe size is constant. For example, during freezing conditions the FSD becomes wider due to increasing $l_{max}$, but the small floes remain as small as they were initially. Thus, it is not surprising that the model sensitivity to changes of the relaxation time $T_{rel}$ is rather limited: the effective floe size is all the time dominated by the small floes, as they contribute a lot to the total perimeter, but much less to the total surface area of the ice. No rapid increase of $l_{eff}$ is possible, even if freezing is fast. To represent lateral freezing, it seems more natural to shift the whole distribution towards larger floe sizes.

I'm not suggesting that the Authors should extend their study to simulations with variable minimum floe sizes, but some discussion of the expected consequences of constant $d_{min}$ would be very useful.

4. The important role of the wave attenuation is particularly interesting from the point of view of possible feedbacks with floe size. If the wave attenuation rates are dependent on floe size, with stronger attenuation in fields of small floes (due, e.g., to floe-floe collisions), then sea ice fragmentation close to the ice edge might modify wave propagation into the MIZ, in turn impacting the floe sizes in the inner MIZ. The sensitivity studies presented in this manuscript suggest that accounting for processes that influence wave attenuation might be important from the point of view of reproducing annual evolution of MIZ.

Other comments:

1. Page 2, lines 24-26: A strange sentence, formulated in a complicated way, but stating something obvious: that if the floe size in a model is constant, it cannot be modified by processes that modify the floe size in the real world.

2. Page 3, line 39: I think you should introduce notation for the minimum floe size, maximum floe size and the power-law exponent already here, and consequently use those symbols throughout the paper (e.g., in order to avoid formulations of the type: "all the different exponent-minimum permutations" (page 10, line30); I had to think for a while to realize what an "exponent-minimum permutation" is).

3. Page 4, line 20: "developed developed"

4. The same notation, α, is used for the floe-shape parameter (e.g., equation 1) and for the exponent of the FSD. Please use different symbols for those two things – for example, use $\alpha_{shape}$, which is used just once on page 14, line 19.

5. Is *A* ice area or ice concentration? You introduce it in equation (1) as ice area, but then the same symbol appears in different equations where it has to be non-dimensional to make sense (e.g., eq. 20 and eq. 10, to which I return in one of the next comments).

6. Δ*T* in equation (2) is not defined.

7. Page 6, line 12: wave cover? And further: wave spectrum? Rather wave energy spectrum.

8. First line in section 2.3: I think it would be good to state explicitly that *N* is the number-weighted FSD (there are other alternatives, e.g., area-weighted FSD).

9. Equation (10): Replace '=' with '→' for consistency with eqs. (12) and (13).
   Also, something is wrong with this equation. First, if *A* denotes ice area, then $1/A$ cannot be subtracted from 1; if *A* is ice concentration, then $1-1/A$ is negative. Further, the whole expression should be multiplied by the time step or another quantity expressed in seconds.

10. Equation (12): What is $d_{frag}$?

11. Page 7, lines 11-14: What happens when new ice is formed in a grid cell that was ice free?

12. Page 7, line 17: I don't understand the second part of this sentence.

13. Is *x* floe radius or floe diameter? From eq. (9) it follows that the floe surface area is $\pi x^2$; from eq. (14) – $\pi x^2/4$.

14. Equation (16): Note that this is valid for α≠3 and α≠2 (for α=3, the integral in eq.(14) is $\log(l_{max}) - \log(d_{min})$; the same is true for eq.(15) and α=2). This is important as α=2 and α=3 are among parameters considered in simulations.

15. Page 7, line 32: it should be $P_{fsd}$ instead of *P*.

16. Page 8, lines 4-7: Why introduce $d_{con}$ if $l_{eff}$ can be used already in eq. (17)?

17. Page 8, line 22-23: The sentence "A reference run is…" is a repetition of the sentence in lines 20-21.

18. Starting from section 4.1, the Authors use negative values of α (e.g., lines 35-36 on page 8). This is inconsistent with section 2.3, in which the FSD is formulated as $N\sim x^{-\alpha}$. Usually in the literature, when references are made to "an exponent of the power-law distribution", a positive number is meant, as in the expression $N\sim x^{-\alpha}$. I'd suggest therefore speaking of exponents as positive numbers, and modifying the text accordingly. For example, line 7 on page 10 should be: "The exponent is increased from 2.5 to 3.5" instead of "reduced from -2.5 to -3.5".

19. Page 11, line 25: "lower ice cover"? Rather lower ice concentration.

20. Page 12, lines 19-20: "…even prior to the melting season in March". This part of the sentence suggests that one should expect the wave-induced fragmentation to occur mainly during the melting season. Obviously, it is mainly related to storminess and the related wave "activity".

21. The first paragraph in section 5 is a summary rather than discussion, so I'm not sure if it belongs here or at the beginning of section 6.
   Similarly, I'd suggest moving the last part of section 5 to section 6. Or simply merging sections 5 and 6 into "discussion and conclusions".

---

## Referee Comment (RC2) · Anonymous Referee #2 · 18 Apr 2019

This paper details a new parameterization for the sea ice floe size effects, which assumes a power law floe size distribution over a given size range of variable exponent and endpoints. The benefit of this approach is it may simulate the effective properties of the sea ice FSD without having to parameterize the underlying physics. This work outlines the WIPoFSD model and analyzes the variability of Arctic sea ice over the period 2007-2016 between standard mixed-layer-ocean-CICE simulations and those using the new parameterization, with a number of parameter perturbation experiments used to assess overall Arctic sensitivity to floe size parameters. They find a lateral-melt feedback on September sea ice extent and volume, where the redistribution of heat from basal to lateral melt leads to future reductions in sea ice volume, and find effective floe size is a good predictor of September sea ice volume/extent. This research

is clearly relevant to ongoing operational and predictive modeling efforts, with the potential to be a useful benchmark for understanding the sensitivity of Arctic sea ice to floe size variability and with the potential to be a useful model for simulation purposes.

The manuscript: (A) outlines a new computationally inexpensive model designed to capture aspects of FSD behavior, and (B) demonstrates a floe-size-melt-feedback at a scale only discussed by Asplin (2012, JGR), Kohout (2014, Nature), and others. I find that (B) is generally well-presented and would be an interesting addition to the scientific literature. Yet aspect (A), while again, a promising idea, lacks key details and, as presented, contains inconsistencies that may preclude publication or adoption of this technique.

I recommend the authors carefully re-examine the presentation and formulation of section 2 before reconsidering their results. I have outlined specific comments that relate to their new additions in Section 2.3, along with more general and copy-editing comments subsequently.

**Section 2.3**: This section must be carefully re-analyzed. When doing so, please be cognizant of variable definitions and units. For example in eq. 14, alpha $> 0$, but in Sec.4 (alpha $< 0$), and alpha is also the shape parameter. Is A a concentration (as it must be in eq (10)) or an area? What are the units of N? What is $d_{frag}$? What is $d_{max}$? Their definitions in Table 1 are not sufficient to follow through the text.

- What is the definition of the FSD, N? By eq.(9) you have adopted an number-weighted definition. Rothrock and Thorndike (1984) give a good way of presenting the FSD. Most modeling approaches have used area-weighted distributions for consistency with the ITD (Zhang et al, 2016) but there has been no consistency there, so defining it clearly is necessary.

- (and P15L29, e.g.,) The «truncated»FSD defined here is not the scale-truncated power law used in the literature, defined in Burroughs and Tebbens (2001), and

investigated in Herman (2011) and Stern et al (2018). It is also not zero at $l_{max}$. What about floes larger and smaller than the size ranges here? How much of the full range of area is being cut out? How low does $l_{max}$ get?

- Eq (9) - Quantities like «max floe size» are not area tracers (you can see they are non-linearly related to the area in Eq (9)), thus the CICE scheme is not appropriate here. These variables must be advected differently (see Horvat and Tziperman (2017)).

- Eq (10) is not consistent: the square root requires that A be unitless. Then the units on the LHS are m, and the units on the RHS are m/s. But if A is unitless, it is a number less than 1, and the inconsistently defined $l_{max}$ is imaginary.

- More importantly, Eq (10) is a main new contribution of this model but it is dictated, not derived. The impact of Eq (10) is easily explored, however, by plugging it into Eq (11) via Eq (18), at which point you can solve for A! Once corrected, is this consistent?

- Eq (13) needs to be re-examined as it is inconsistent with Eq (9). Since the growth rate of $l_{max}$ is not derived based on the change in sea ice area, the constraint (9) can only be accomplished by changing the coefficient C introduced in (8). Essentially, this means reducing the area occupied by floes at all scales (except for the very largest), which is not what is happening! Restoring is ok, but $T_{rel}$ should relate to dA/dt. This would eliminate the need for the sentences at L15 justifying this approach. If the authors wish to use an ad-hoc parameterization, then why invoke the FSD and its distributional features at all? Instead, just write a heuristic equation for $l_{eff}$.

- P7L6 - what does this mean? Is lambda the peak wavelength? The spectrum as outlined previously is a frequency spectrum.

[Figure]

**Comments:**

- P2L19 - As you mention, many models (LIM, e.g.) do not have any floe size, so «simplifies model code»isn't quite right - there were no FSD schemes when these models were written!

- P2L24 - I think you should be careful in this section - given the small FSD literature, what «dictates the best approach»may not be known yet.

- P2L35 (and where discussed later, e.g. P8L32) - The cited paper does not given evidence for a unique power law across that range of scales. In the abstract: «We found that the FSDs from the high-resolution images follow power laws over floe sizes from 10 m to 3 km.»Though similar-sounding, this is different to «we found all FSDs follow a unique power law . . .»over that range, and there is a good reason! Picking a fixed exponent to run the model is a great simplifying idea, but should not overemphasize the applicability of a single power law.

- P3L34 - I would re-write this passage to avoid making qualitative judgements about other sea ice and climate modelers. Model developments all come at a cost and it is not within the scope of this study to diagnose and prescribe best practices, particularly when trying to justify one's own parameterization.

- **Model description:** In general, there is too much extraneous information given in the model description. Is it important to discuss standard model physics of the CICE model? Or what NEMO is, especially since NEMO is not used in this study? There is much more detail given to previous work than the new work, I recommend cutting this substantially, and focusing on what directly affects the FSD model here. This could go in the Supporting Information if it is necessary.

- P4L14 - I'm not sure why NEMO is included here or the references to the SWARP project. You later use a mixed layer model to produce all results!

- P5L31 - because of the many types of wave spectra can become confusing, it may be helpful to write this as $S(\omega)d\omega$ to make comparison easier with other studies using the spectra. See Michel (1968 and 1999).

- P6L23 - it appears as if this is the place that waves reduce the FSD maximum size. But how is not explained, and should be included in the text.

- Sec. 2.4 - I think the use of perimeter per square meter is fantastic - the right approach.

- P7L36 - Equation 16 is a ratio of ice perimeter to ice area, i.e. the 1st moment of N divided by the 2nd moment of N (if N is an area). The shape parameter (which is not introduced as part of N prior to this comment) does not cancel from this expression.

- P10 - How does this parameter space exploration differ from what was performed by Roach et al (2017), who reduced the CICE floe size parameter in much the same way as did Steele (1992)?

- P13L10 (and discussion on P14) - I don't believe these sensitivity experiments are sufficient to test variability in ocean response - as mentioned the minimum mixed layer depth is a numerical crutch because Kraus-Turner models become singular - is this parameter really what controls mixed layer feedbacks? There is such a wide range of Arctic mixed layers (see Peralta-Ferriz and Woodgate (2015)) that you may be exiting the range of interest here.

- **Discussion:** The sensitivity experiments are referenced to a time-varying base-line (2007-2016). What is the rationale for choosing this period? Why should deviations from such a strongly forced system be used as a sensitivity? I do not think concepts like the multi-year memory of Arctic sea ice can be understood based on two averages across 10-year periods. In general here I would support simply explaining how certain parameter changes affect results, but there is

not enough information presented to support qualitative claims about the Arctic system response.

- We do not know the relative importance of the three factors that affect the FSD: melting, waves, and freezing. A needed figure is one that breaks down the seasonal variation in $l_{max}$ (better, $l_{eff}$) as a function of each forcing components. Furthermore we don't see how N evolves in time, neither $l_{max}$, but these are key variables of the WIPoFSD model!

- **Figures:** This may be a question of style you can ignore, but the figure captions contain descriptions or context that aren't directly describing the figure (e.g., the final 3 sentences in Fig 4). These may be better off in the text. Same for using parentheses instead of division «/»(or a comma, in Fig 1) for units on axis labels. Feel free to ignore this comment.

**Copy-editing comments**

- P1L1 - Authors' choice, but «sea ice floe size distribution»is probably good in the title, even if obvious.

- P1L15 - «climate sea ice models»(pick one! or sea ice models in climate models)

- P1L21 - «this feature is important in correcting existing biases»- true (see Roach et al 2017), but not supported within the text or referenced again.

- P1L36 and elsewhere - be sure to say «sea ice»instead of «ice»throughout.

- P2L2 «described as marginal»- seems mean, perhaps say «the region with X characteristic, referred to as the MIZ, etc». See again the use of marginal as an adjective on L16.

- P2L4 - no need to capitalize Marginal Ice Zone.

- P2L7 - The S+J citation here is about past changes to Arctic cyclones, not about future changes to sea state/storminess.

- P2L18 - «calliper»can be removed here.

- P3L3 - Are these autonomous techniques recovering the FSD?

- P3L9 - Consider looking at Perovich and Jones (2014) here.

- P4L2 - P4L6 «allow time», probably best to say «permit more sensitivity studies». You may also remove the statements about «lends itself to sensitivity. . .»because this is true of any new parameterization with new parameters to tweak.

- P5L4 - Please re-consider using alpha here or later.

- P6L1 - this aside about interpolation needs to be copy-edited as it introduces undefined terminology (e.g., what is a smooth variation?)

- P6L6 - why are waves advected in 5 directions? And is there any scattering or reflection?

- P6L29 - please mention the values of $d_{min}$ and $\alpha$ here.

- P8L16 - Replace this acronym with a description of the reanalysis product.

- P8L36 - this is the first time «cumulative distribution»is introduced - please outline what the distribution is prior to here.

- P9L6 - «peak melting season»- what time period you are referring to?

- P9L21 - index these quantities to their definitions in the text.

- P9L25 - do you mean «between Stan-ref and ref?»

- P9L34 - «fractional ice area»→ «ice concentration»(throughout)

- P9L36 - The strong restoring and imposed external forcing fields complicate statements like this, I'd consider removing this statement unless you plan to examine the ice flow field.

- P10L7 - explain in words what changing the distribution means - how much does the effective floe size drop? Same at L26.

- P1039 - PL-FSD model?

- P11L26 - Why is this parameterization chosen? Perovich and Jones (2014) provide a useful relationship between power law exponent and area, as does Birnbaum and Lupkes (2001).

- P11L30 - Which effective floe size? Averaged over the Arctic? How is this done? What does «predictor»mean here?

- P11L41 «earlier observations»→ «the previous result»?

- P12L21 - In general, I do not see how changing the restoring is reflective of memory in the system. You have not presented any evidence as to the magnitude of the restoring response. You could look at seasonal tendencies in $l_{eff}$ which would be far more helpful when analyzing these sensitivity experiments.

- P12L29 - «are uncorrelated parameters»→ «may be uncorrelated parameters».

- P12L30 - As before, consider re-defining this term, alpha is already in use.

- P14L1 - «melt potential»is not previously defined.

- P14L1 - which simulation?

- P15L32 - «The WIPoFSD»→ In the «WIPoFSD model, the FSD»- I know, this is ugly, but you have to!

- Figure 1 - please include some schematic depiction of the axis and its scale (logarithmic or linear).

- Table 3 - Runs are in lowercase, but uppercase in the text.

- Table 3 - why not put in parentheses the percentage change instead of absolute change?
* * *

---

## Author Comment (AC1) · 19 Jul 2019

**Response to referee comments**

**(Referee comments are shown in black, our response is in blue and changes to the manuscript are shown in red. The revised manuscript is also included in this document.)**

**Page references are given to the updated manuscript as PXLY indicating that the manuscript has been updated on page X line Y.**

**Anonymous Referee #1**

Firstly, we would like to thank the reviewer for providing such thorough and thoughtful comments on our manuscript. They have been invaluable in improving the manuscript.

In their manuscript „Impact of floe size distribution on seasonal fragmentation and melt of Arctic sea ice", Adam Bateson and colleagues present results of a numerical sensitivity study concentrating on the role of the floe size distribution (FSD) in shaping the seasonal cycle of the sea ice cover in the Arctic. The study is based on a numerical sea ice model coupled with a mixed layer ocean model. Several processes influencing/influenced by the FSD are investigated, including the lateral melting of sea ice and ice fragmentation induced by waves. The FSDs considered are truncated power laws described by three parameters: the minimum and maximum floe size, and the exponent.
The manuscript is devoted to a subject which is very important for the performance of numerical sea ice models on both short (synoptic) and long (seasonal, climate) time scales. As the authors point out in the introduction, recent climate change and the associated negative trends in sea ice extent, thickness and strength have produced conditions in which interactions with waves are becoming more and more important over larger and larger areas. Fragmentation of sea ice by waves in turn influences ice-ocean-atmosphere heat flux and sea ice melting/freezing rates. At present, our limited understanding of many aspects of these coupled processes is a serious limitation for development of reliable parameterizations suitable for numerical sea ice models. In my opinion, the proposed manuscript is an important contribution to the subject, even though many solutions used in the model and assumptions underlying them are oversimplified (or maybe even wrong). As the authors correctly remark, these simplifying assumptions are to a large degree a result of the lack of observational data and/or theoretical understanding available. In this respect, the most important contribution of the manuscript is that it develops a framework in which future developments can be integrated, as new data and insights become available. In other words, in spite of some clear limitations of the solutions presented, I find this contribution very valuable, as it paves the way for further development.

Several aspects of the results are very interesting, for example the findings related to the (partial) compensation of the increased lateral-melting rates by decreased bottom-melting rates in simulations with power-law FSD compared to simulations with constant floe size; or the important role of the wave attenuation rates in shaping the seasonal cycle of sea ice. I think that those results are worth publishing in "The Cryosphere". My comments to the manuscript are listed below. My recommendation to the Editors is "major revision".

We would like to thank the reviewer for pointing out the value of our study for future improvement of numerical sea ice models.

General comments:

1. The text of the manuscript is written in an untidy manner, it contains a lot of (mostly small) mistakes and should be carefully checked before publication. Moreover, the text is full of technical

slang, colloquialisms, informal expressions which (in most cases) are comprehensible for a reader familiar with numerical modelling of ocean/sea ice, but should be avoided in a paper. It makes reading of the text tiresome.

We have carefully checked and modified our manuscript to improve readability and comprehensibility.

2. I find the description in section 2.2 quite chaotic. The impression is that the choice of topics that are described there is pretty random, even though the first sentence announces that this section contains only "elements pertinent to our study". For example, is the CFL criterion relevant for the results/discussion presented further? In turn, some more details from the papers of Williams and colleagues regarding computation of wave-induced breaking etc. would be very useful, even though they can be found in those papers.

We have modified the structure and removed some less important aspects including the additional sentence regarding the CFL criterion, which has been deemed unnecessary. The model description has been expanded to give a better explanation of how the waves-in-ice component of the model is currently operating (see section 2.2). This includes a more detailed outline of how the wave spectra is reconstructed and key properties calculated after advection (P7L27-P7L35) and an explanation of the breaking strain amplitude and a more detailed overview of the terms required to identify a wave break-up event (P7L35 – P8L3).

R1: But the most important comment to section 2.2 is related to the description of the wave forcing. It is very imprecise. Also, I think the Authors should better justify the methodology they use. As written in section 3, the only wave information from ERA reanalysis used as input to the sea ice model are $H_s$ and $T_p$ (I think this information should be provided in section 2.2, so that the text there can be better understood). Why don't the Authors use wave directions from ERA? Or even the whole energy spectra?

Information regarding Stokes drift within the ice field was not available as a standard output at the time of model development. Whilst this may no longer be the case, updating the schemes to use the Stokes drift wave directions was not judged to be an effective use of resources for this present study. $H_s$ and $T_p$ are used rather than the full wave energy spectra for consistency with Williams et al. (2013a, 2013b).

The following general statement regarding waves-in-ice model choices has now been added to the start of section 2.2 and included here for reference: 'The waves-in-ice module described here reproduces wave conditions near the sea ice edge within the MIZ. Local wind direction determines the direction of wave propagation with adjustments made for attenuation imposed by the sea ice cover. This is a compromise dictated by availability of forcing data, lack of observational studies and the course resolution of the CICE model.' Specific comments have now been made in the manuscript regarding the decision not to use Stokes drift (P7L1 – P7L3) and the choice of $H_s$ and $T_p$ are used rather than the full wave energy spectra (P6L26 – P6L27) as outlined here. The description of the dataset used to obtain $H_s$ and $T_p$ has been moved to section 2.2 as suggested.

The approach described seems very complicated and requires several arbitrary assumptions. First, the assumption about the correspondence between the wave direction and wind direction might be justified, e.g, in the Beaufort Sea, where waves are predominantly locally generated, but the wave climate in the North Atlantic is dominated by swell, so that the direction of waves reaching the ice edge likely has little in common with the local wind.

Stopa et al. (2016) discuss wave climate in the Arctic between 1992 and 2014 and they find that in the Atlantic side of the Arctic basin 'wave seasons follow the winds', however they also state that regions exposed to the North Atlantic wave climate will be strongly influenced by swells generated within the North Atlantic Ocean. Semi-enclosed and isolated seas e.g. Laptev, Kara are more event driven and have an equal mix of wind driven and swell driven waves.

It has been clarified in the manuscript that the approach used here to derive wave directions is a simplification and neglects wave direction from swell effects, as outlined here (P7L3 – P7L9).

Moreover, swell typically has a much narrower directional distribution than 45° used by the Authors (by the way: what does "total spread" mean? how is the wave energy distributed among the 5 directional bins?). This might seem a minor detail, but narrower directional distribution of wave energy means that the waves can penetrate deeper into the ice cover (with the same attenuation rate $\alpha_{dim}$) – and the results presented in this paper suggest this might have a significant influence on the results.

'Total spread' is the angular size of the surface wave spread. The energy is distributed amongst the bins according to $\frac{2}{\pi}(\cos \Delta\theta)^2$ where $\Delta\theta$ is the deviation from the principal wave direction. Modelling the propagation of waves under the sea ice cover is an area of ongoing research. Wadhams et al. (1986) produced observations that show that waves propagating into the MIZ could experience significant wave spreading until it was 'essentially isotropic'. However this paper also noted a distinction between wind seas where the isotropic state could be achieved within a few km and swell seas where spreading occurs much more slowly, if at all. A recent study from Montiel et al. (2016) was able to specifically link spreading behaviour to wave period, with two regimes identified. Shorter wavelengths experienced spreading and longer wavelengths did not with a transition between these two regimes defined by the maximum floe size. This is consistent with the observed behaviour of wave driven regimens and swell driven regimes. Using a fixed surface wave spread across a limited number of categories is a significant simplification of the rather complex spreading behaviour of waves, however it represents a balance between short wave periods that quickly achieve an isotropic state and longer wave periods that propagate much further into the MIZ before they experience significant spreading.

The information about how 'total spread' is defined and how energy is distributed is now included in the manuscript (P6L31 – P6L33). The discussion outlined here regarding the choice for the direction distribution has also now been included in the manuscript (P7L9 – P7L18).

Second, from the technical perspective, the approach requires advecting wave energy into the ice in a bin-by-bin manner, and waves from several locations can reach any given ice-covered grid cell. Using full ERA energy spectra would be much more elegant, but not much more complicated.

See above for motivation to use significant wave height and peak wave period rather than the full ERA wave spectra.

3. It is hard to prove without numerical simulations, but my impression is that some conclusions from this study might be affected by the – very artificial – assumption that the minimum floe size is constant. For example, during freezing conditions the FSD becomes wider due to increasing $l_{max}$, but the small floes remain as small as they were initially. Thus, it is not surprising that the model sensitivity to changes of the relaxation time $T_{rel}$ is rather limited: the effective floe size is all the time dominated by the small floes, as they contribute a lot to the total perimeter, but much less to the total surface area of the ice. No rapid increase of $l_{eff}$ is possible, even if freezing is fast. To represent lateral freezing, it seems more natural to shift the whole distribution towards larger floe sizes. I'm

not suggesting that the Authors should extend their study to simulations with variable minimum floe sizes, but some discussion of the expected consequences of constant $d_{min}$ would be very useful.

It is not necessarily true to say no rapid increase in $l_{eff}$ is possible; it depends on the choice of parameters e.g. in figure 5 the effective floe size rapidly transitions from 10 m to over 500 m within the MIZ. It should also be noted that when $l_{max}$ increases in value, the number frequency of smaller floes must decrease to maintain consistency between the FSD and sea ice concentration. We explored the impact of a variable exponent on the basis of evidence from Stern et al. (2018 a & b) that the exponent isn't a fixed value and instead evolves spatially and temporally. There isn't a clear set of evidence to show that the minimum floe size also varies in a similar manner and other FSD modelling approaches e.g. the prognostic FSTD approach adopted by Roach et al. (2018) also shows a high frequency of small floes across the sea ice extent throughout the year.

We have included a section within the discussion section (P17L12 – P17L16) of the manuscript regarding why we do not investigate a fixed $d_{min}$, specifically pointing out the lack of observations motivating such a sensitivity study.

4. The important role of the wave attenuation is particularly interesting from the point of view of possible feedbacks with floe size. If the wave attenuation rates are dependent on floe size, with stronger attenuation in fields of small floes (due, e.g., to floe-floe collisions), then sea ice fragmentation close to the ice edge might modify wave propagation into the MIZ, in turn impacting the floe sizes in the inner MIZ. The sensitivity studies presented in this manuscript suggest that accounting for processes that influence wave attenuation might be important from the point of view of reproducing annual evolution of MIZ.

Agreed. The attenuation rate is inversely proportional to floe size (as according to Eq. (3), Williams et al., 2013 a). We present a sensitivity study here (Section 4.3.2) where the attenuation rate is reduced by a factor of 10 with a modest model response. However, there are definitely interesting feedbacks to be further investigated between floe size and attenuation rate and to properly understand the role in MIZ evolution we need to better understand and represent the attenuation of waves within models.

We include a section within the discussion commenting on the importance of wave-floe size feedbacks and the need to better capture the attenuation of waves in sea ice within models (P16L40 – P17L8).

Other comments:

Page 2, lines 24-26: A strange sentence, formulated in a complicated way, but stating something obvious: that if the floe size in a model is constant, it cannot be modified by processes that modify the floe size in the real world.

The purpose of this sentence was to express the idea that to accurately capture the impact of waves on sea ice, for example, some form of floe size change mechanism is required.

The sentence has been updated for clarity from 'Since the power law is the cumulative outcome of all the mechanical and thermodynamic processes that influence the FSD, some of the impact of an individual process will be tied up within the imposed power law itself' to 'The assumption of a fixed floe size also prevents sea ice models from accurately representing the impact of processes on the

sea ice evolution that act via the perturbation of floe size such as lateral melting and wave induced fragmentation of floes.'

2. Page 3, line 39: I think you should introduce notation for the minimum floe size, maximum floe size and the power-law exponent already here, and consequently use those symbols throughout the paper (e.g., in order to avoid formulations of the type: "all the different exponent-minimum permutations" (page 10, line30); I had to think for a while to realize what an "exponent-minimum permutation" is).

Agreed. The changes you have suggested have been implemented within the paper.

3. Page 4, line 20: "developed developed"

Noted and fixed.

4. The same notation, α, is used for the floe-shape parameter (e.g., equation 1) and for the exponent of the FSD. Please use different symbols for those two things – for example, use $\alpha_{shape}$, which is used just once on page 14, line 19.

Agreed.

α will now be used solely to represent the exponent of the power law FSD, and $\alpha_{shape}$ will be used to refer to the floe-shape parameter.

5. Is A ice area or ice concentration? You introduce it in equation (1) as ice area, but then the same symbol appears in different equations where it has to be non-dimensional to make sense (e.g., eq. 20 and eq. 10, to which I return in one of the next comments).

A refers to the ice concentration.

This has been made clear in the manuscript. The statement 'A refers to the ice area' now reads 'A refers to the ice concentration'.

6. ΔT in equation (2) is not defined.

ΔT is the elevation of the surface water temperature above freezing.

ΔT is now defined in the manuscript as above.

R1: 7. Page 6, line 12: wave cover? And further: wave spectrum? Rather wave energy spectrum.

Noted and fixed. Line updated from 'Next, the attenuation of the wave cover over each wave timestep is calculated. This will be calculated for each individual wave spectrum' to 'Next, the attenuation of waves over each wave timestep is calculated. This will be calculated for each individual wave energy spectrum'.

R1: 8. First line in section 2.3: I think it would be good to state explicitly that N is the number-weighted FSD (there are other alternatives, e.g., area-weighted FSD).

Agreed.

I have now explicitly stated that N is the number-weighted FSD as suggested.

9. Equation (10): Replace '=' with '→' for consistency with eqs. (12) and (13). Also, something is wrong with this equation. First, if A denotes ice area, then 1/A cannot be subtracted from 1; if A is ice concentration, then 1-1/A is negative. Further, the whole expression should be multiplied by the time step or another quantity expressed in seconds.

Equation (10) is indeed wrong as written in the manuscript. Thank you for pointing out the error. The correct expression is:

$$l_{max,final} = l_{max,initial} \sqrt{1 - \frac{\Delta A_{lm}}{A}}.$$

Where $\Delta A_{lm}$ is the reduction in ice concentration that can be attributed to lateral melting. Note that the change in $l_{max}$ is a function of another time dependent variable, $\Delta A_{lm}$, hence the expression does not need to be multiplied by the time step. We have used the correct expression in the model and it was only incorrect in the manuscript.

'=' has been replaced with '→' as suggested and Equation (10) has also now been corrected in the manuscript.

10. Equation (12): What is $d_{frag}$?

Should be $d_{min}$ not $d_{frag}$.

This has been corrected in the manuscript.

11. Page 7, lines 11-14: What happens when new ice is formed in a grid cell that was ice free?

$l_{max}$ is set to $d_{min}$ in a grid cell where new sea ice is formed.

This has been clarified in the text as follows: 'In grid cells that transition from being ice free to having a sea ice cover, the maximum floe size is initiated with its minimum value i.e. $d_{min}$.'.

12. Page 7, line 17: I don't understand the second part of this sentence.

This is supposed to recognise the fact that we can't represent the full impact of individual processes through this modelling approach as it assumes a power law rather than being a fully prognostic model.

The sentence has been removed as its mention here is confusing and unnecessary.

13. Is x floe radius or floe diameter? From eq. (9) it follows that the floe surface area is $\square x_2$; from eq. (14) – $\square x_2/4$.

It is the floe diameter.

This is now clarified when the FSD is defined: 'We employ a number-weighted FSD, N( x ), where x is the floe diameter.'

14. Equation (16): Note that this is valid for α≠3 and α≠2 (for α=3, the integral in eq.(14) is $\log(l_{max})$ − $\log(d_{min})$; the same is true for eq.(15) and α=2. This is important as α=2 and α=3 are among parameters considered in simulations.

To simplify the model code, rather than defining a different equation to evaluate for α=2 and α=3, 0.001 is added to their values. This will have a negligible impact on the accuracy of the model.

A statement has been added to the manuscript to clarify this.

15. Page 7, line 32: it should be $P_{fsd}$ instead of P.

Noted and updated.

16. Page 8, lines 4-7: Why introduce $d_{con}$ if $l_{eff}$ can be used already in eq. (17)?

$d_{con}$ should be replaced replaced with L, an already defined parameter for the floe size where the floe size is fixed. Equation (17) is the definition of perimeter density for simulations with a constant, fixed floe size, and L is the relevant floe size parameter in this case.

The manuscript has been updated to use L within Eq. (17) rather than $d_{con.}$

17. Page 8, line 22-23: The sentence "A reference run is…" is a repetition of the sentence in lines 20-21.

Noted and fixed.

18. Starting from section 4.1, the Authors use negative values of α (e.g., lines 35-36 on page 8). This is inconsistent with section 2.3, in which the FSD is formulated as $N\sim x_{-\alpha}$. Usually in the literature, when references are made to "an exponent of the power-law distribution", a positive number is meant, as in the expression $N\sim x_{-\alpha}$. I'd suggest therefore speaking of exponents as positive numbers, and modifying the text accordingly. For example, line 7 on page 10 should be: "The exponent is increased from 2.5 to 3.5" instead of "reduced from -2.5 to -3.5".

Noted and fixed. The exponent will now be treated as positive in the manuscript, as suggested.

19. Page 11, line 25: "lower ice cover"? Rather lower ice concentration.

Noted and fixed.

20. Page 12, lines 19-20: "…even prior to the melting season in March". This part of the sentence suggests that one should expect the wave-induced fragmentation to occur mainly during the melting season. Obviously, it is mainly related to storminess and the related wave "activity".

We are referring to the impact of the model on sea ice concentration in the MIZ and lateral melting, however, have decided that highlighting this point is unnecessary and misleading.

The phrase referred to here has now been removed from the manuscript.

21. The first paragraph in section 5 is a summary rather than discussion, so I'm not sure if it belongs here or at the beginning of section 6. Similarly, I'd suggest moving the last part of section 5 to section 6. Or simply merging sections 5 and 6 into "discussion and conclusions".

The final part of section 5 is included in the discussion rather than the conclusion as it introduces new material.

[revised manuscript text omitted]

---

## Author Comment (AC2) · 19 Jul 2019

**Response to referee comments**

**(Referee comments are shown in black, our response is in blue and changes to the manuscript are shown in red. The revised manuscript is also included in this document.)**

**Page references are given to the updated manuscript as PXLY indicating that the manuscript has been updated on page X line Y.**

**Anonymous Referee #2**

*Firstly, we would like to thank the reviewer for providing such thorough and thoughtful comments on our manuscript. They have been invaluable in improving the manuscript.*

This paper details a new parameterization for the sea ice floe size effects, which assumes a power law floe size distribution over a given size range of variable exponent and endpoints. The benefit of this approach is it may simulate the effective properties of the sea ice FSD without having to parameterize the underlying physics. This work outlines the WIPoFSD model and analyzes the variability of Arctic sea ice over the period 2007-2016 between standard mixed-layer-ocean-CICE simulations and those using the new parameterization, with a number of parameter perturbation experiments used to assess overall Arctic sensitivity to floe size parameters. They find a lateral melt feedback on September sea ice extent and volume, where the redistribution of heat from basal to lateral melt leads to future reductions in sea ice volume, and find effective floe size is a good predictor of September sea ice volume/extent. This research is clearly relevant to ongoing operational and predictive modeling efforts, with the potential to be a useful benchmark for understanding the sensitivity of Arctic sea ice to floe size variability and with the potential to be a useful model for simulation purposes.

The manuscript: (A) outlines a new computationally inexpensive model designed to capture aspects of FSD behavior, and (B) demonstrates a floe-size-melt-feedback at a scale only discussed by Asplin (2012, JGR), Kohout (2014, Nature), and others. I find that (B) is generally well-presented and would be an interesting addition to the scientific literature. Yet aspect (A), while again, a promising idea, lacks key details and, as presented, contains inconsistencies that may preclude publication or adoption of this technique.

I recommend the authors carefully re-examine the presentation and formulation of section 2 before reconsidering their results. I have outlined specific comments that relate to their new additions in Section 2.3, along with more general and copy-editing comments subsequently.

Section 2.3: This section must be carefully re-analyzed. When doing so, please be cognizant of variable definitions and units. For example in eq. 14, alpha > 0, but in Sec.4 (alpha < 0), and alpha is also the shape parameter.

*Noted and fixed. The shape parameter will now be referred to as $\alpha_{shape}$.*

Is A a concentration (as it must be in eq (10)) or an area?

*A is a concentration.*

*This has now been clarified in the manuscript.*

What are the units of N?

$m^{-1}$ noting that $\frac{\alpha_{shape}}{A_{grid}} \int_{d_{min}}^{l_{max}} N x^2 \, dx = 1$, where $A_{grid}$ is the grid cell size.

*This has now been clarified in the manuscript.*

What is dfrag?

*Should be $d_{min}$ not $d_{frag}$.*

*Now fixed in manuscript.*

What is dmax? Their definitions in Table 1 are not sufficient to follow through the text.

*$d_{max}$ is the fixed global maximum floe size (as opposed to the variable grid cell specific maximum floe size, $l_{max}$).*

*The following line has been added to the introduction of the manuscript to ensure this parameter is carefully defined: 'The distribution is defined by three parameters: $d_{min}$, minimum floe size; $d_{max}$, global maximum floe size and α, the power law exponent.' To ensure the relationship between $d_{max}$ and $l_{max}$ is clear, where the FSD, N(x), is defined the following description is now included: '$l_{max}$ is the local maximum floe size. The parameters can be defined independently for each grid cell, however in this study $d_{min}$ and α will be fixed across the ice cover within an individual simulation, such that only $l_{max}$ will vary in response to processes which would be expected to change the floe size. $l_{max}$ is allowed to vary between $d_{min}$ and $d_{max}$, the global minimum and maximum floe sizes respectively.'*

• What is the definition of the FSD, N? By eq.(9) you have adopted an numberweighted definition. Rothrock and Thorndike (1984) give a good way of presenting the FSD. Most modeling approaches have used area-weighted distributions for consistency with the ITD (Zhang et al, 2016) but there has been no consistency there, so defining it clearly is necessary.

*N is a number-weighted FSD. The number-weighted FSD is selected over the area-weighted FSD to be consistent with how observations are reported e.g. Stern et al. (2018 a, b).*

*We have now defined N more clearly in the text: 'We employ a number-weighted FSD, N( x ), where x is the floe diameter.'*

• (and P15L29, e.g.,) The «truncated»FSD defined here is not the scale-truncated power law used in the literature, defined in Burroughs and Tebbens (2001), and investigated in Herman (2011) and Stern et al (2018).

*The truncated FSD defined here is consistent with other literature in the field. Note the literature has considered both the complementary cumulative number distribution and the standard number distribution for plotting observations of the FSD. Here we specifically use a standard number distribution as a result of the complications outlined in Stern et al. (2018 b) associated with the use of the complementary cumulative number distribution (i.e. a system that adopts an upper-truncated power law in the probability density function does not necessarily produce a power law in its cumulative form).*

It is also not zero at lmax.

*This is a standard feature of observed FSDs that display a truncated power law form.*

What about floes larger and smaller than the size ranges here? How much of the full range of area is being cut out?

*This model aims to represent the full FSD, and hence it is a model assumption that there are no floes larger or smaller that the range modelled.*

*This point is now made explicitly in the manuscript (P8L16 – P8L17).*

How low does lmax get?

*Its minimum value is set to $l_{min}$. Grid cells that transition from ice-free to including sea ice initiate the FSD with $l_{max} = d_{min}$ i.e. the range of $l_{max}$ is $d_{min}$ to $d_{max}$.*

*This has made clear in the manuscript (P10L2 – P10L3).*

• Eq (9) - Quantities like «max floe size»are not area tracers (you can see they are non-linearly related to the area in Eq (9)), thus the CICE scheme is not appropriate here. These variables must be advected differently (see Horvat and Tziperman (2017)).

*$l_{max}$ is treated as an area tracer. In this model it is a specific property assigned to areas of sea ice alongside $\alpha$ and $d_{min}$ to define the FSD and should not be considered a property specific to any individual floe. Hence the use of the CICE scheme is appropriate in this case.*

*This explanation has now been included in the manuscript (P9L29 – P9L31).*

• Eq (10) is not consistent: the square root requires that A be unitless. Then the units on the LHS are m, and the units on the RHS are m/s. But if A is unitless, it is a number less than 1, and the inconsistently defined lmax is imaginary.

*Agreed. Equation (10) is indeed wrong as written in the manuscript and we apologise for this. Thank you for pointing out the error. The correct expression is:*

$$l_{max,final} = l_{max,initial}\sqrt{1 - \frac{\Delta A_{lm}}{A}}.$$

*Where $\Delta A_{lm}$ is the reduction in ice concentration that can be attributed to lateral melting. We have used the correct expression in the model and it was only incorrect in the manuscript.*

*The manuscript has been updated to correct this error.*

• More importantly, Eq (10) is a main new contribution of this model but it is dictated, not derived. The impact of Eq (10) is easily explored, however, by plugging it into Eq (11) via Eq (18), at which point you can solve for A! Once corrected, is this consistent?

*Eq. (10) is derived from assuming the reduction in $l_{max}^2$ from lateral melting is proportional to the reduction in $A$, the sea ice concentration, from lateral melting: $\left(\frac{l_{max,final}}{l_{max,initial}}\right)^2 = \frac{A_{final}}{A_{initial}}$.*
*It should be noted that the changes in both $A$ and $l_{max}$ will perturb the entire distribution via the normalisation constant, $C$. This parameterisation is not supposed to represent the impact of lateral melting specifically on a floe of size $l_{max}$, rather it aims to represent changes in $l_{eff}$ resulting from reductions in $N$ across all floe sizes.*

*The equation and explanation given here have now been included in the manuscript to provide context to Eq (10) (P9L16 – P9L23).*

• Eq (13) needs to be re-examined as it is inconsistent with Eq (9). Since the growth rate of lmax is not derived based on the change in sea ice area, the constraint (9) can only be accomplished by changing the coefficient C introduced in (8). Essentially, this means reducing the area occupied by floes at all scales (except for the very largest), which is not what is happening!

*This is effectively what is happening, however. The changes in the FSD currently only impact the model via the lateral melt volume, which is calculated as a function of effective floe size. In the context of the model, effective floe size is scale invariant and the calculation of the total lateral melt volume also requires knowledge of the ice concentration. Therefore Eq. (13) and Eq. (9) can still be consistent with each other.*
*This feature of the model has been clarified in the manuscript with the following statement: 'It is important to note that changes in $lmax$ impact the entire FSD because C, the normalisation constant, is a function of $lmax$, as defined in Eq. (5). Processes that change the total sea ice concentration, such as lateral melting, also change C. This is because C is calculated to ensure the FSD is normalised to be consistent with the total sea ice surface area.'*

Restoring is ok, but Trel should relate to dA/dt. This would eliminate the need for the sentences at L15 justifying this approach.

*The motivation of this paper is to explore the role of the floe size distribution in driving the seasonal retreat of the Arctic sea ice. This means priority has been given to processes that reduce the size of floes in summer i.e. lateral melting, and wave breakup. Given floe growth is a winter process it was deemed less important to represent floe growth processes in a physically realistic manner, and instead a restoring approach has been used to avoid making any assumptions about how floes grow. It should be noted that the real mechanism of floe growth is more complex e.g. new floes forming as both pancake and nilas ice, welding, lateral growth.*

If the authors wish to use an ad-hoc parameterization, then why invoke the FSD and its distributional features at all? Instead, just write a heuristic equation for leff .

*The WIPoFSD model includes complicated feedbacks that could not be represented using a heuristic equation for $l_{eff}$ e.g. the attenuation rate of the WIM is a function of the mean floe size (as opposed to the effective floe size).*

• P7L6 - what does this mean? Is lambda the peak wavelength? The spectrum as outlined previously is a frequency spectrum.

*The origin of this term is outlined in Williams et al. (2013 a), where it is labelled $\lambda_w$ and described as a representative wavelength.*

*We have updated the model description to more clearly explain how this term is calculated (P8L1 – P8L3).*

• P2L19 - As you mention, many models (LIM, e.g.) do not have any floe size, so «simplifies model code»isn't quite right - there were no FSD schemes when these models were written.

*Noted. The sentence 'this assumption both simplifies the model code and reduces computational costs of running the model' has been removed from the manuscript.*

• P2L24 - I think you should be careful in this section - given the small FSD literature, what «dictates the best approach» may not be known yet.

*The point that was supposed to be made here is that there is value in having multiple FSD models. The prognostic approach described in Roach et al. (2018) has some clear advantages to the WIPoFSD model, however that doesn't mean there aren't applications where the approach presented here is more appropriate.*

*The statement has been softened from ' Context dictates the most suitable approach; higher complexity floe size representation will be required for high resolution regional sea ice modelling than for large scale climate models' to 'The most suitable modelling approach will be context dependent; for example, high resolution regional sea ice models would be expected to require a higher complexity of floe size treatment than large scale climate models'.*

• P2L35 (and where discussed later, e.g. P8L32) - The cited paper does not given evidence for a unique power law across that range of scales. In the abstract: «We found that the FSDs from the high-resolution images follow power laws over floe sizes from 10 m to 3 km.»Though similar-sounding, this is different to «we found all FSDs follow a unique power law . . .»over that range, and there is a good reason! Picking a fixed exponent to run the model is a great simplifying idea, but should not overemphasize the applicability of a single power law.

*The results displayed by Stern et al. (2013 a) do show a remarkably consistent exponent over the range, however there are obviously caveats to that and it is unnecessary for us to make such strong claims here.*

*We have softened the statements made on this point: 'This same study was also able to use two satellite data sets with different resolutions but operating over the same region to show evidence of power law behaviour over floes from as small as 10 m and as large as 30,000 m.'*

• P3L34 - I would re-write this passage to avoid making qualitative judgements about other sea ice and climate modelers. Model developments all come at a cost and it is not within the scope of this study to diagnose and prescribe best practices, particularly when trying to justify one's own parameterization.

*Point taken on board, and in hindsight this may have come across too much as a direct attack. We were trying to convey the point that the best approach is not necessarily the one with the highest physical fidelity. However, this point has been made in an earlier section so it is unnecessary to reiterate it here.*

*The sentence 'This may be acceptable for use within a standalone sea ice model but coupled-climate modellers will be reluctant to accept a significant cost without evidence that it has a significant impact on how the sea ice interacts with the climate system' has now been removed from the manuscript.*

• Model description: In general, there is too much extraneous information given in the model description. Is it important to discuss standard model physics of the CICE model? Or what NEMO is, especially since NEMO is not used in this study? There is much more detail given to previous work than the new work, I recommend cutting this substantially, and focusing on what directly affects the FSD model here. This could go in the Supporting Information if it is necessary.

*We think it is useful to include details of the standard CICE model that pertain to the themes discussed in this manuscript.*

*References to NEMO have now been removed. The balance between new and existing work has been addressed in the manuscript to increase the focus on the new model components presented here. The main changes are: The reduction of material relating to the CFL criterion; the addition of new material to justify the method of advecting the wave energy spectra and its limitations; a more detailed overview of how the wave energy spectra is reconstructed and relevant parameters obtained from the spectra; and a more detailed overview of how the parameters used to identify a wave break-up event are calculated (section 2.2).*

• P4L14 - I'm not sure why NEMO is included here or the references to the SWARP project. You later use a mixed layer model to produce all results!

*The references to NEMO and SWARP have now been removed.*

• P5L31 - because of the many types of wave spectra can become confusing, it may be helpful to write this as S(!)d! to make comparison easier with other studies using th.e spectra. See Michel (1968 and 1999).

*Noted. We have now expressed S analogously to Williams et al. (2013 a) for consistency.*

• P6L23 - it appears as if this is the place that waves reduce the FSD maximum size. But how is not explained, and should be included in the text.

*This is given in section 2.4. We wanted to explicitly describe all the processes that influence floe size in the same section for ease of reference.*

*A explicit reference to section 2.4 is now made in the text.*

• Sec. 2.4 - I think the use of perimeter per square meter is fantastic - the right approach.

*Thank you; we agree.*

• P7L36 - Equation 16 is a ratio of ice perimeter to ice area, i.e. the 1st moment of N divided by the 2nd moment of N (if N is an area). The shape parameter (which is not introduced as part of N prior to this comment) does not cancel from this expression.

*You are indeed correct. This was an error made in writing the manuscript. The floe shape parameter does however cancel out from Eq. (18) i.e. Eq. (18) remains correct.*

*The equations in the manuscript have now been corrected.*

• P10 - How does this parameter space exploration differ from what was performed by Roach et al (2017), who reduced the CICE floe size parameter in much the same way as did Steele (1992)?

*Roach et al. (2017) investigated automated tuning for a series of parameters, however floe size was not one of these. Steel (1992) explored the impact of changing L, the fixed floe diameter, invariant in time and space. Floe diameters of 30 m, 300 m and 3000 m were investigated. Here we explore the sensitivity to the fixed floe size parameters used to construct our FSD. $l_{eff}$, the equivalent to L in our model, is a function of both these fixed parameters and $l_{max}$, which is a variable that is defined per*

*grid cell. Hence, unlike Steele (1992), our equivalent floe size parameter is not fixed either on a spatial or temporal scale.*

• P13L10 (and discussion on P14) - I don't believe these sensitivity experiments are sufficient to test variability in ocean response - as mentioned the minimum mixed layer depth is a numerical crutch because Kraus-Turner models become singular - is this parameter really what controls mixed layer feedbacks? There is such a wide range of Arctic mixed layers (see Peralta-Ferriz and Woodgate (2015)) that you may be exiting the range of interest here.

*We agree that a mixed layer model can not capture the full range of ocean variability, however this was not the focus of this study. Tsamados et al. (2015) have previously discussed the performance the prognostic ML model used here with respect to Peralta-Ferriz and Woodgate (2015). The mixed layer model is generally realistic though does tend to underestimate summer MLD compared to the results presented by Peralta-Ferriz and Woodgate (2015), though their results also show the MLD can be as low as 5 m.*

*The discussion here has now been included in the manuscript (P5L36 – P5L40).*

• Discussion: The sensitivity experiments are referenced to a time-varying baseline (2007-2016). What is the rationale for choosing this period? Why should deviations from such a strongly forced system be used as a sensitivity? I do not think concepts like the multi-year memory of Arctic sea ice can be understood based on two averages across 10-year periods. In general here I would support simply explaining how certain parameter changes affect results, but there is not enough information presented to support qualitative claims about the Arctic system response.

*In this study we are not trying to address decadal scale changes in sea ice, instead we are trying to understand the impact of the FSD and associated processes on the seasonal sea ice loss. 2007 – 2016 has been selected as the baseline as it will capture the current climatology of the Arctic, including extrema of sea ice extent (e.g. 2012 compared to 2013). Whilst we can't make precise quantitative assessments of the impacts of different sensitivity studies on the Arctic system, we can still compare the magnitude of the different sensitivity results.*

*A comment has been added to the manuscript to explain the choice of time period and address the other issues raised here using the same arguments made above (P10L22 – P10L25).*

• We do not know the relative importance of the three factors that affect the FSD: melting, waves, and freezing. A needed figure is one that breaks down the seasonal variation in lmax (better, leff ) as a function of each forcing components. Furthermore we don't see how N evolves in time, neither lmax, but these are key variables of the WIPoFSD model!

*Sensitivity studies (F), (G) and (K) have been performed to explore the importance of these different processes. However, it seems like a sensible idea to plot the evolution of $l_{eff}$ for these different sensitivity studies compared to the standard case. Given $l_{max}$ is the only time dependent variable that N is a function of, we do not think it is necessary to plot how both N and $l_{eff}$ evolve with time, which are both functions of $l_{max}$.*

*Figure 12 has now been added to the paper to address this. Figure 12 shows the daily annual evolution of $l_{eff}$ over 2015 averaged over regions with between 15 – 80 % sea ice concentration on 31st August 2015 and regions with between 15 – 30 % sea ice concentration on 31st August 2015 for three simulations exploring the effective floe size tendencies with respect to different processes. These sets are chosen to capture regions that become part of the MIZ for at least some of the year*

*without becoming sea ice free such that on annual cycle is artificially introduced to the $l_{eff}$. The following lines have been added / modified within the manuscript to reflect the new figure: P14L24 – P1415L2, P16L29 – P16L32, P16L35.*

• Figures: This may be a question of style you can ignore, but the figure captions contain descriptions or context that aren't directly describing the figure (e.g., the final 3 sentences in Fig 4). These may be better off in the text. Same for using parentheses instead of division «/»(or a comma, in Fig 1) for units on axis labels. Feel free to ignore this comment.

*The figure captions are certainly longer than standard, though this choice was made intentionally. We are happy to shorten this but will leave it as an editorial decision. Likewise, with the style for units on axis labels.*

**Copy-editing comments**

• P1L1 - Authors' choice, but «sea ice floe size distribution»is probably good in the title, even if obvious.

*Noted and added.*

• P1L15 - «climate sea ice models»(pick one! or sea ice models in climate models)

*Noted and changed to sea ice models within climate models.*

• P1L21 - «this feature is important in correcting existing biases»- true (see Roach et al 2017), but not supported within the text or referenced again.

*This theme is discussed in the final paragraph of the discussion (P17L35 – P17L43).*

• P1L36 and elsewhere - be sure to say «sea ice»instead of «ice»throughout.

*Noted and fixed.*

• P2L2 «described as marginal»- seems mean, perhaps say «the region with X characteristic, referred to as the MIZ, etc». See again the use of marginal as an adjective on L16.

*Noted and changed. Sentence now reads 'similarly the region of the Arctic identified as the marginal ice zone (MIZ), defined here as where the ice area fraction extends between 15 and 80%, is projected to increase in extent (Aksenov et al., 2017).'*

• P2L4 - no need to capitalize Marginal Ice Zone.

*Noted and changed.*

• P2L7 - The S+J citation here is about past changes to Arctic cyclones, not about future changes to sea state/storminess.

*Noted and replaced with more relevant and recent citations (Casas-Prat et al., 2018; Day and Hodges, 2018).*

• P2L18 - «calliper»can be removed here.

*Noted and fixed.*

• P3L3 - Are these autonomous techniques recovering the FSD?

*No, they are giving a high-resolution picture of sea ice and ocean state that can be used alongside timeseries of high-resolution FSD data to better understand the processes that drive FSD evolution.*

*Text updated to remove this ambiguity: 'Novel techniques, particularly those using autonomous platforms and robotic instruments, are enabling increased high-resolution data capture of sea ice and ocean conditions that can be used alongside time series of up to 1m-resolution FSD data obtained through remote sensing to better understand the factors driving FSD evolution (Thomson and Lee, 2017).'*

• P3L9 - Consider looking at Perovich and Jones (2014) here.

*Reference now included in relevant location, following on from the sentence just quoted: 'This data can be applied within an approach analogous to Perovich and Jones (2014), who used aerial photography alongside simple parameterisations for lateral melting and floe fragmentation by waves, assuming the floe size cumulative distribution adopts a power law, to explore whether these processes could result in the observed changes to the FSD.'*

• P4L2 - P4L6 «allow time», probably best to say «permit more sensitivity studies». You may also remove the statements about «lends itself to sensitivity. . .»because this is true of any new parameterization with new parameters to tweak.

*Text edited accordingly. 'Lends itself to sensitivity studies' has been changed to 'a series of additional experiments are also possible within this framework'.*

• P5L4 - Please re-consider using alpha here or later.

*Noted and fixed.*

• P8L16 - Replace this acronym with a description of the reanalysis product.

*Noted and fixed. Line now reads: 'The mixed layer properties are restored over a timescale of 5 days to a monthly climatology reanalysis at 10 m depth taken from the Ferry et al. (2011) reanalysis dataset.'*

• P8L36 - this is the first time «cumulative distribution»is introduced - please outline what the distribution is prior to here.

*It has now been made clear in the model description that we are using a number-weighted FSD. Cumulative distributions are also mentioned in the introduction. Nevertheless, 'as opposed to a cumulative distribution' has been removed here to avoid confusion.*

• P9L6 - «peak melting season»- what time period you are referring to?

*Sentence poorly constructed.*

*This has now been changed to 'The difference in sea ice area reaches a maximum in August whereas the difference in sea ice volume peaks in September.'*

• P9L21 - index these quantities to their definitions in the text.

*These quantities have previously been defined in the model description.*

*A reference to this section has been added into the text.*

• P9L25 - do you mean «between Stan-ref and ref?»

*Sentence has been updated to remove ambiguities: 'the difference in ice area between stan-fsd and ref'.*

• P9L34 - «fractional ice area»! «ice concentration»(throughout)

*Noted and fixed.*

• P9L36 - The strong restoring and imposed external forcing fields complicate statements like this, I'd consider removing this statement unless you plan to examine the ice flow field.

*Agreed; thank you.*

*'The distribution observed suggests the Beaufort Gyre is redistributing pack ice of higher concentration and volume to more marginal locations. In particular this appears to reverse the reduction in sea ice thickness at the outer edge of the MIZ off the Alaskan coast that is otherwise observed along the rest of the outer MIZ region.' has been removed.*

• P10L7 - explain in words what changing the distribution means - how much does the effective floe size drop? Same at L26.

*Figures 7 and 8 provide information to show what the mean effective floe size is for each of the simulations presented in this section (noting that this parameter is not fixed either spatially or temporally). It is not possible to give a singular answer of how much the effective floe size will drop as this parameter can vary both spatially and temporally.*

*We have updated the manuscript to include commentary on how changing the model parameters changes the distribution at the start of section 4.2: 'It is valuable to consider how changes to each FSD parameter is likely to impact the distribution: increasing α increases the number of small floes in the distribution and reduces the number of larger floes; increasing $d_{min}$ removes smaller floes from the distribution entirely, increasing the number of floes across the rest of the distribution; increasing $d_{max}$ adds larger floes to the distribution, reducing the number of floes across the rest of the distribution.'*

• P1039 - PL-FSD model?

*Apologies; this was an earlier iteration of the name of the WIPoFSD model.*

*Noted and fixed.*

• P11L26 - Why is this parameterization chosen? Perovich and Jones (2014) provide a useful relationship between power law exponent and area, as does Birnbaum and Lupkes (2001).

*The parameterisation selected here was chosen to capture the range of exponents seen in observations (c.f. Stern et al., 2018 b), as stated in the manuscript. The relationship between ice concentration and area described by Perovich and Jones seems to specifically apply to obtaining an estimate of the exponent from observations as it also required an estimate of perimeter density, something that is an emergent property of the FSD in our case and not a variable that evolves independently. We have been unable to find the Birnbaum and Lupkes reference you suggested. There is a paper from the same authors published in 2002, however this does not provide any relationship between power law exponent and area.*

• P11L30 - Which effective floe size? Averaged over the Arctic?

*This is referring to the effective floe size value averaged over the MIZ and reported in table 3.*

*Text updated to: 'the mean September effective floe size in the MIZ, as reported in table 3,'.*

How is this done? What does «predictor»mean here?

*To average effective floe size over the Arctic the average is taken across each grid cell where ice concentration exceeds 15%, with equal weighting given to each grid cell.*

*The method to take the average is now mentioned in the figure caption to table 3. Sentence edited to remove the word predictor.*

• P11L41 «earlier observations»! «the previous result»?

*Noted and fixed.*

• P12L21 - In general, I do not see how changing the restoring is reflective of memory in the system. You have not presented any evidence as to the magnitude of the restoring response. You could look at seasonal tendencies in leff which would be far more helpful when analyzing these sensitivity experiments.

*The way the model is currently setup, any fragmentation events that happen in the freeze-up season will not impact the model in any meaningful way as $l_{max}$ will be restored to its maximum value before the melting season begins. By reducing the rate of restoring, any fragmentation events that occur during the freeze-up season will still have a residual impact on $l_{max}$ during the subsequent melting season i.e. the system has a stronger 'memory' of fragmentation events. We agree that it is useful to compare how $l_{eff}$ evolves during the season and have included an additional plot to explore this.*

*Figure 12, as described above, includes the evolution of $l_{eff}$ for the simulation with reduced floe restoring rates. This is discussed in the manuscript (P14L40 – P15L2, P16L30 – P16L32).*

• P12L29 - «are uncorrelated parameters»! «may be uncorrelated parameters».

*Noted and fixed.*

• P12L30 - As before, consider re-defining this term, alpha is already in use.

*Noted and fixed.*

• P14L1 - «melt potential»is not previously defined.

*The melting potential has now been defined in section 2.1 (P5L24 – P5L29).*

• P14L1 - which simulation?

*The results of this simulation have not been presented here. Note Roach et al. (2018) report a similar result.*

*(not presented) has been added in brackets after this reference to make it clear this simulation has not been presented.*

• P15L32 - «The WIPoFSD»! In the «WIPoFSD model, the FSD»- I know, this is ugly, but you have to!

*Noted and fixed.*

• Figure 1 - please include some schematic depiction of the axis and its scale (logarithmic or linear).

*Noted and fixed.*

• Table 3 - Runs are in lowercase, but uppercase in the text.

*Noted and fixed.*

• Table 3 - why not put in parentheses the percentage change instead of absolute change?

*It is useful to be able to compare the changes in adjacent cells i.e. how does the change in lateral melt compare to the change in basal melt for a given simulation. Similarly, to what extent can losses in total ice volume be explained by loses in the MIZ. This is a lot easier to do for absolute values than percentages.*

[revised manuscript text omitted]

---

## Author Response (AR3)

**Response to referee comments**

**(Referee comments are shown in black, our response is in blue and changes to the manuscript are shown in red. The revised manuscript is also included in this document.)**

**Page references are given to the updated manuscript as PXLY indicating that the manuscript has been updated on page X line Y.**

General comments: Thank you again to both reviewers and the editor for your helpful feedback. Before we address the specific concerns, we would like to make some comments on the aims of our study and the overall structure of the model.

The main purpose of this manuscript is to learn how a floe size distribution will affect the sea ice mass balance, and not to study the specifics of how the form of the floe size distribution emerges. We wish to use a computationally light approach such as would be appealing for climate model use. As summarised by Stern et al. (2018a), observations of the floe size distribution are generally fitted to a power law. In this manuscript we aim to establish how this general form of distribution will impact the sea ice mass balance.

We construct our FSD as a power law using three fixed parameters:

- $d_{min}$: the lower floe size cut-off of the FSD.
- $d_{max}$: the upper floe size cut-off for the FSD.
- $\alpha$: the power law exponent.

The standard values for each of these parameters are selected from observations, as discussed in the manuscript. We establish model sensitivity to these parameters, over ranges also determined by the variation seen in observations.

We also introduce a new tracer, $l_{max}$, to capture the impact of processes that can change the floe size distribution within the model. This tracer is not intended to represent a physical maximum floe size, instead it is an internal model tool to capture the physical effects of processes that will affect the floe size distribution. $l_{max}$ is effectively a tracer that we use to represent the history of a given area of sea ice in terms of events that affect the floe size distribution. Due to some (understandable) confusion, we propose that we rename $l_{max}$ as $l_{var}$.

Throughout the manuscript $l_{max}$ has been relabelled as the variable FSD tracer, $l_{var}$. We have also clarified the description of the WIPoFSD model and $l_{var}$ at P8L8 – P8L12.

**Reviewer 1**

**1. The advection scheme used in the model is not suitable, as maximum floe size is not an area-conserved quantity.**

We are not sure if the concern here relates specifically to the advection of an individual floe size metric or the broader concerns regarding the advection of averaged distribution properties raised in Horvat and Tziperman (2017), however we will address both.

To address the first concern, consider the following scenario. We have two grid cells with the same sea ice area fraction. The first is filled with a uniform set of floes of 300 m diameter, the second is filled with a uniform set of floes of 600 m diameter. If the floe size property was area-conserved, the number average floe diameter would be 450 m. However, to ensure the total sea ice area is the same in each grid cell, there must be more 300 m diameter floes than 600 m diameter floes. This

means the number average floe diameter must be smaller than 450 m. Hence the floe diameter is clearly not an area-conserved quantity. We previously noted that $l_{var}$ (previously $l_{max}$) is not a property assigned to specific floes. It is instead a variable we assign to specific areas of sea ice to represent processes that have occurred previously to that sea ice area that will impact the floe size distribution. Hence, it is appropriate in this case to treat it as an area-conserved quantity.

Turning to the second point. It is true that section 3.1 of Horvat and Tziperman (2017) gives a clear demonstration that mean floe size does not advect as an area-conserved property. More broadly, this applies to the advection of normalised or mean properties relating to the floe size distribution. However, $l_{var}$ is calculated independently to the floe size distribution, i.e. it is not a diagnostic property calculated from the floe size distribution. Therefore, the issues raised by Horvat and Tziperman (2017) regarding the use of the tracer advection scheme are not applicable here.

P10L19 – P10L25: A summary of the response here has been included in the manuscript to address the possible concerns regarding the advection scheme.

**2. I wanted to ask for clarification on how they compute the change in basal melt as a result of change in sea ice area. It was not clear to me from the manuscript how they computed this when I tried to reproduce the calculation with my own model results. I think it would also be important to look at spatial maps here, rather than simply the area-integrated quantities.**

In the manuscript we state that this quantity was 'calculated by multiplying the difference in ice area between *stan-fsd* and *ref* by the basal melt rate in *ref*'. We acknowledge that this has not been phrased in an unambiguous way.

To explain what we have done to calculate the reduction in basal melt, consider a specific grid cell. Imagine for a given time step the sea ice fraction for that grid cell in the *stan-fsd* simulation is 0.81 and in the *ref* simulation it is 0.90. If this physical reduction is the only factor causing changes to the total basal melt, then the basal melt rate per unit grid cell area (the metric outputted by CICE) would also reduce by the same factor of 10% from *ref* to *stan-fsd*. We can then calculate the reduction in the total basal melt volume for this grid cell accounting only for the reduction in sea ice fraction as the product of 0.1, the basal melt rate per unit grid cell area, and the area of the grid cell. We can then repeat this exercise over every grid cell to obtain the total reduction in basal melt volume accounting only for reduction in sea ice concentration as expressed in figure 4 in the manuscript. We are happy to make available the code used to calculate the predicted basal melt if required.

We present a new figure here, figure 1, to address the suggestion to look at spatial maps and not just area-integrated quantities. Here we show that the predicted basal melt can capture the regional distribution of the expected changes in basal melt, not just the area-integrated quantity. We do not propose to include this plot in the manuscript given the substantial number of plots already in the manuscript, however it is presented here ready to be inserted into the manuscript if the editor and / or reviewer think it is necessary.

P11L39 – P12L2: Description of figure 4 has been updated to clarify how the predicted basal melt is calculated.

[Figure]

**Figure 1:** Predicted reduction in basal melt rate from *stan-fsd* to *ref* (top row, a-c), actual reduction in basal melt rate from *stan-fsd* to *ref* (middle row, d-f), and difference between the actual reduction and predicted reduction in basal melt rate (bottom row, g-i) averaged over 2007 – 2016. Results are presented for March (left column, a, d, g), June (middle column, b, e, h) and September (right column, c, f, i). Values are shown only in locations where the sea ice concentration exceeds 5 %. The predicted reduction in basal melt rate refers to the expected reduction if the change in sea ice area fraction is the only factor driving the change in basal melt rate. This is calculated by multiplying the basal melt rate for *ref* by the relative percent change in ice area fraction from *ref* to *stan-fsd* for each grid cell.

**Reviewer 2**

**1) I reiterate the use of an "upper-truncated FSD" is not well-justified. The origin of the terminology "truncated power-law distribution" in the current FSD literature comes from Burroughs and Tebbens (2001) - regarding measurement truncation. Stern (2018) hypothesized that an observed tail-off of the FSD at high sizes in a well-known paper of Toyota \*could\* be explained by a combination of measurement truncation error and an erroneous use of a cumulative distribution. If this hypothesis is generally true, the implication is that the power law extends across all sizes - not that it is truncated. If this hypothesis is false, Toyota's suggestion of different scaling behavior at large scales is true. Regardless of which is correct, neither supports the use of a single power-law distribution between two arbitrary sizes, where the highest size is set by physics, not observational limitations.**

We recognise here that the way we have previously expressed the model has not been clear. We have a fixed upper cut-off to our floe size distribution of $d_{max}$, determined from observations. We believe that an upper floe size cut-off is necessary because we define the floe size distribution for each grid cell, hence the size of each grid cell serves as an upper limit to floes that can be resolved within an individual grid cell. In the case of 1° CICE, this size is around 55 – 60 km. In addition, we do not represent several processes that we expect to be important in determining the evolution of larger floes. We expect wind stresses, internal sea ice floe strain and melt ponds to have an increasingly important role in breaking up kilometre scale floes (Wilchinsky et al, 2009; Arntsen et al, 2015).

We set the upper cut-off, $d_{max}$, to 30 km in standard simulations. We determine this value from the observations of Stern et al. (2018b). We also perform sensitivity studies to $d_{max}$, increasing it to 50 km and reducing it to 10 km and 1 km, to establish its importance in our model for the purpose of assessing impact on sea ice mass balance. We find a much weaker model response for changes in $d_{max}$ compared to changes in $d_{min}$ and $\alpha$.

$l_{var}$ (previously referred to as $l_{max}$), as we highlight at the beginning of this response, should be considered an internal model tool to represent the impacts of physical processes on the entire floe size distribution. It does not represent a physical maximum floe size.

We have clarified the meaning of the different parameters, where relevant, throughout the manuscript. In particular this includes P8L8 – P8L12. We have also provided further discussion on the limitations for the values these parameters can take over P13L6 – P13L14.

**Thus issues remain with the use of l_{max} as a controlling parameter in Sec.2.4. As previously noted, changing l_{max} for any reason while maintaining the constraints of Eq (8-9) alters the number/area of floes at all sizes below l_{max}. But physically, why? It is unclear in this manuscript why changes to sea ice at the highest size should be connected to changes at the lowest size, but the normalization makes this connection explicit.**

We assume a power law FSD for computational convenience and because observations indicate that over the range of lengthscales relevant to the total sea ice mass balance it is a reasonable approximation. Our model is not intended to represent the impact of physical processes on the details of the floe size distribution. This is clearly not possible using our computationally light approach of using a power law FSD. For example, Horvat and Tziperman (2017) show how lateral melting drives a distribution away from a power law. We approximate the impact of any given

process on the distribution via the variable $l_{var}$. It is a feature, not a flaw, that varying $l_{var}$ causes changes across the distribution.

We have included some comments to the above effect, including why the whole distribution should change in response to changes in $l_{var}$, on P9L12 – P9L18.

Here's another way of thinking about the l_{max} issue. l_{max} is the scale with (by definition) the least number of floes. Because the exponents of your power-law section are always above 2, the area distribution, x^2 N, also declines with increasing size - so the scale l_{max} is also that with the least floe area. Intuitively, then, how could your model be tested? It is the hardest-to-observe scale in the problem, all the scales above it are dependent on it, but a real measurement would be pressed to decide which scale matters, and how it is evolving because of the known truncation error issues. Would this require simply tracking a single, large floe? Can you justify why in your equations, a small lateral melt on the biggest floes would lead to huge changes in the numbers of small floes?

These two features of the discussion make interpreting the (well-presented) results difficult. Even at the fairly liberal "this is for convenience" level, why multi-scale variability at all scales is so related to this scale is not justified here. The evolution terms for l_{max} are stated, not derived, in a few lines, but the entire work depends on them, and no plots of l_{max} are shown in the paper. What is the time evolution of the FSD required of these equations? Does it make sense?

In terms of validating the model with observations, we would look at the effective floe size. Studies such as Horvat et al. (2019) are already working towards producing floe size products from satellite imagery. We can use such products to evaluate if our model produces the same variability and range of these products as the observational products show.

We have not included plots of $l_{var}$ but we have included plots of $l_{eff}$. As discussed in the manuscript, $l_{eff}$ is the important parameter to consider for the mass balance of the sea ice. It is therefore easier to interpret the physical implications of the $l_{eff}$ evolution in comparison to $l_{var}$. Furthermore, $l_{eff}$ can be calculated for a floe size distribution of any form, power law or otherwise. This provides a method to compare the model performance to observations, even if those observations do not necessarily follow a power law.

As suggested, we will now discuss the three processes that influence the floe size distribution and explain how changes in $l_{var}$ can capture the desired features. We can broadly consider $l_{var}$ to be a scaling variable; as we reduce $l_{var}$ we redistribute floes from larger sizes to smaller sizes, with $l_{var}$ marking the transition of this redistribution.

*Wave break-up:* during a wave break-up event we expect a reduction in the number density of larger floes and an increase in the number of smaller floes. As described in the manuscript, after a floe breaking event $l_{var}$ is set to $\frac{\lambda_w}{2}$ ($\lambda_w$ is determined by the specific wave and floe properties) providing it is larger than $\lambda_{min}$. This means that we increase the number density of all floes below this fragmentation length-scale. Figure 2 illustrates this change.

[Figure]

**Figure 2:** The floe size number distribution in a grid cell before (blue, dashed) and after (red, dotted) a wave break-up where the maximum floe size is reduced from 1000 m to 500 m.

*Lateral melting:* the change in $l_{var}$ due to lateral melting is explicitly linked to the reduction in sea ice concentration due to lateral melting, as described in equation 17. The floe number distribution is also a function of the sea ice concentration (see equation 10 in the manuscript). This means that we must consider both the change to $l_{var}$ and to the sea ice concentration when considering how the model will change in response to lateral melting. Figure 3, below, shows what this looks like in practice.

[Figure]

**Figure 3:** The floe size number distribution in a grid cell before (blue, dashed) and after (red, dotted) after a lateral melting event where the sea ice fraction drops by 75% and $l_{var}$ reduces by 50%, as determined by equation 17 in the manuscript.

Here we see that, even though $l_{var}$ decreases, we see a reduction in the number distribution across all floe sizes. This is because of the reduction in the sea ice fraction. Previous studies, such as Horvat and Tziperman (2017), have shown that lateral melting causes stronger deviation from the power law for smaller floes than larger floes. However lateral melting would also result in floes smaller than $d_{min}$ that will contribute to an even higher lateral melt relative to the floe size. Hence the behaviour of our lateral melt scheme can be considered a compensation between these two behaviours. In an earlier iteration of the model, we did complete tests using the number mean floe size in place of $l_{var}$ in equation 17, however we found negligible impact. The use of the number mean floe size, or alternatively $l_{eff}$, requires the reconstruction of $l_{var}$ using numerical methods as there is no analytical solution possible, hence we decided to avoid introducing this additional complication to

the model. We would also like to point out that when we remove the lateral melt feedback on floe size in sensitivity study (G), the impact on the simulation is negligible. This makes sense considering that even during the peak melt season in the *fsd-stan* simulation, the lateral melt rarely exceeds 1 cm day$^{-1}$. We see this in figure 12 in the manuscript. Even through a careful selection of grid cells where we would expect lateral melting to have the strongest feedback on the floe size distribution, we see $l_{eff}$ only decreases by about 10 %, not enough to see significant feedbacks in the lateral melt rate.

*Floe growth in freezing conditions:* Whereas wave break-up events are independent of the thermodynamic state of the sea ice (i.e. whether it is growing or retreating) and lateral melting is explicitly linked to the sea ice area loss due to lateral melting, the floe growth restoring is related to increases in the sea ice area without being explicitly linked. This is because both new ice growth and floe growth are initiated by the same threshold. Hence, we can have scenarios of floe growth with and without significant increases in the sea ice extent. Figure 4 illustrates how both scenarios will impact the floe size distribution.

[Figure]

**Figure 4:** The floe size number distribution in a grid cell before (blue, dashed) and after (red, dotted) after a floe growth event where (left) the floe size increases from 500 m to 1000 m with an increase in the sea ice area fraction from 0.25 to 0.30 and (right) the floe size increases from 500 m to 1000 m with an increase in the sea ice fraction from 0.25 to 0.50.

The qualitative behaviour we see in figure 4 is consistent with the different floe growth processes. There are three processes that we need to consider during freezing conditions as described by Roach et al. (2018): lateral growth; welding of floes; formation of new floes. When we have freezing conditions without a large increase in the sea ice extent, we expect welding to dominate the evolution of the floe size distribution. This results in the formation of larger floes and the reduction in the number of smaller floes, as we see on the left plot in figure 3. When we have large increase in the sea ice area fraction, we expect both lateral growth and the formation of new floes, hence we expect to see increases in the number distribution across all floe sizes. This is demonstrated by the right plot in figure 4. The primary aim of this study is on the seasonal melt and fragmentation of sea ice rather than the Winter evolution, hence the use of a simple floe growth restoring scheme. However, even with this simple approach, we can capture some of the features of the different floe growth processes.

**My major suggestion is the authors present a justification/derivation/explanation of (10,12,13) to show that the FSD evolution terms are justified. This doesn't need to be very long, but necessary for the reader to evaluate what such equations mean and how they affect the distribution.**

As suggested, a summary of the discussion above has been added to section 2.4 (P9 – P10) to provide an explanation / justification of how the $l_{var}$ parameterisations change the overall FSD.

**It would also be nice (but more work) if the authors convert the parameterization to a new, "untruncated" one characterized by l_{min} and R, where R is the effective floe size. You will (as you already do) have to impose a finite-size cutoff on the floe size to integrate sea ice area for some exponents, but this cutoff would be a fixed property of the system not a parameterized one. Then equations (10-13) are mappable to equations in R instead of l_{max}. When you do that, the equations can be evaluated in a slightly more familiar and appealing context, avoiding the confusing truncated power-law issue altogether.**

When observational studies report a floe size distribution that is fitted to a power law, it is common to report the exponent of the fitted power law. A large range of values has been reported for this exponent between different studies (Stern et al., 2018a). One of the purposes of this study has been to perform sensitivity studies over the range seen in observations to assess how this exponent changes the impact of the FSD on the sea ice mass balance (see section 4.2 in the manuscript). This includes studies where we use simple parameterisations to introduce a variable exponent (section 4.3 in the manuscript).

If we do not use $l_{var}$ to represent variability in the distribution, then within a power law framework over a fixed floe size range the only other component of the system that can change is the exponent. Hence, the exponent in this setup would become an emergent parameter rather than one determined from observations. Understanding the processes that determine the exponent / broader shape of a floe size distribution is necessary and valuable research. However, other models such as the prognostic floe size-thickness distribution model (Roach et al., 2018) are better equipped to answer this question.

Nevertheless, your suggestion is an interesting and important one, and highlights the direction that future studies should proceed in terms of the development of computationally light floe size distribution models.

**Small comment: would it be possible also to include the wave-fracture mode (changing l_{max} to lambda/2) explicitly in Fig 1?**

We have updated figure 1 in the manuscript accordingly.

**References**

[revised manuscript text omitted]

---

## Author Response (AR4)

**Response to editor's comments**

**(Editor comments are shown in black, our response is in blue and changes to the manuscript are shown in red. The revised manuscript with markup is also included in this document. Page references are given as PXLY referring the page X and line Y).**

General comments: We would like to thank the editor for their helpful feedback and supportive guidance through the peer review process. The manuscript has improved substantially in response to the detailed feedback provided by both the editor and reviewers.

**Thank you for your detailed response to the reviewers. I feel the work is sufficient for publication, however I would like to echo the reviewers in that there are areas in the model that could be improved upon in future. I am glad that you are considering the suggested changes to thresholding in your future work and perhaps this should be mentioned in the paper so future readers might be alerted to this.**

We agree with your comments here and have made the following changes in response:

The following section in the discussion:

[revised manuscript text omitted]

These updated sections should clarify to readers the role the WIPoFSD model plays in the wider literature of floe size modelling and outline the direction we, the authors, are working in to further develop floe size modelling.

**I do feel that you should be more transparent as to how your model could potentially misrepresent physics. I agree that an empirical model is useful for assessing feedbacks, but there are some obvious limitations in the model. In particular I feel you need to be more explicit in how the variable l_{var} controls the floe size distribution and that this is perhaps not following known physical processes, as pointed out by reviewer 2.**

We have included the following additional comment at the end of section 2.4 to reiterate the limitations of our modelling approach (P10L26 – P10L30):

*'It is worth commenting here on the limitations of the modelling approach to floe size used in this study. The use of a power law distribution with a fixed exponent to describe the FSD is a valuable simplification to explore the impact of floe size on the Arctic sea ice. The tracer $l_{var}$ is an internal model tool used to enable parameterisations of how individual processes impact the FSD within this constrained framework. The parameterisations described in this section are necessarily approximations of how these processes might impact the FSD and should not be considered exact physical descriptions.'*

As mentioned above, the following lines that have been added to the discussion also reiterate the role of $l_{var}$ and the limitations in presents (P18L20 – P18L25):

*'The use of $l_{var}$ to represent variability within the FSD puts limits on the physical fidelity of the parameterisations of processes that change the FSD in our model. However, if $l_{var}$ is not used to*

*represent variability in the distribution, then within a power law framework over a fixed floe size range the only other component of the system that can change is the exponent. The exponent in such a setup becomes an emergent parameter rather than one determined from observations. An important component of this study is to perform sensitivity studies of the sea ice mass balance to the range of exponents seen in observations.'*

**I am okay with not including figure 1 from your response letter in the manuscript. It is sufficient for this to be in the response to the reviewers, which I believe is archived. I would say however, that you are welcome to include it if you wish. It does help the reader to see where the differences lie and it would be instructive for the reader to see where this aligns to changes in ice concentration. Your text sums this up qualitatively.**

We agree with your suggestions here.

The additional figure has now been included in the manuscript as figure 5, with the following commentary added to the manuscript (P12L9 – P12L13):

*'Figure 5 shows the spatial distribution for the predicted reduction in basal melt from stan-fsd to ref, the actual reduction in basal melt, and the difference between the actual reduction and predicted reduction in basal melt. These map plots are presented as monthly averages for March, June and September averaged over 2007 – 2016.  Figure 5 shows that the predicted basal melt can capture the regional distribution of the expected changes in basal melt from ref to stan-fsd, not just the area-integrated quantity.'*

**Some technical corrections you could consider**

**The MIZ is actually defined as the regions of ice under the influence of waves. Hence lines 10-12 on page 2 are a little awkward to read.**

We have rewritten lines 4 – 12 on page 2 to ensure the definition of the MIZ is clearly stated (P2L3 – P2L12):

*'Similarly the region of the Arctic identified as the marginal ice zone (MIZ), generally defined as the region where ocean waves are able to significantly influence the dynamics of the sea ice (Strong et al., 2017), is projected to increase in extent (Aksenov et al., 2017). An alternative definition of the MIZ, and the one that will be used in the present study, is the region where the concentration of the sea ice extends between 15 and 80 %. This definition of the MIZ is often more practical for modelling and observational studies where sea ice concentration data is more readily available than information about wave behaviour in sea ice.*

*Modelling the MIZ is a significant challenge due to its complexity; it is a region in which there is strong coupling between the sea ice, ocean and atmosphere (Lee et al., 2012; McPhee et al., 1987). The sea ice cover in this region is significantly broken up and fragmented by the waves that define the MIZ (Liu et al., 1992).'*

**line 30 page 4: "was used" not will be. You are reporting on work performed in the past.**

Noted and corrected.

**line 26, page 8: A '+' symbol has been accidentally placed in 'produc+es'.**

Noted and corrected.

[revised manuscript text omitted]